# A participatory approach to determine the use of road cut slope design guidelines in Nepal to lessen landslides

Ellen B. Robson [1], Bhim Kumar Dahal [2], and David G. Toll [1]

[1]Institute of Hazard, Risk, and Resilience, Durham University, Durham, UK
[2]Institute of Engineering, Pulchowk Campus, Tribhuvan University, Kathmandu, Nepal

**Correspondence:** Ellen B. Robson  (ellen.robson@durham.ac.uk)

**Abstract.** Road cut slope failures are a type of landslide process and are extensive across the road network of Nepal. In response to the pressing need for improved road cut slope design guidelines to help prevent these failures, this study employs a participatory approach to assess the use of the current guidelines in Nepal and identify critical areas for enhancement to inform the development of new guidelines. We organized a one-day workshop with 34 participants, conducted six semi-structured interviews and five unstructured interviews, facilitated two one-hour focus groups, and distributed 19 questionnaires. Participants in this research included local and federal government engineers, consultants, and academics. We conducted a thematic analysis of the qualitative data. Our findings reveal significant inconsistency in guideline adherence, attributed to their lack of user-friendliness, inconsistent recommendations, and inadequate training for engineers. We found that engineers at local levels often resort to empirical methods when designing cut slopes due to constraints such as land acquisition difficulties. To address these challenges, we propose the development of contextually appropriate guidelines that prioritize simplicity, accessibility, and practicality for field application. We also suggest that policymakers need to set protocols and standards for key road slope management processes, including land acquisition and compensation, quality assurance checks, and spoil disposal, and provide incentives to encourage high levels of compliance. Through these measures, this study aims to lessen road cut slope failures in Nepal, thereby, enhancing the resilience of the road network.

## 1 Introduction

Over 10% of global rainfall-triggered fatal landslides occur in Nepal, despite having less than 0.4% of the global population (Froude and Petley, 2018). Landslides are widespread throughout Nepal due to a complex interplay of natural and anthropogenic processes (McAdoo et al., 2018; KC et al., 2024). The most significant natural triggers of slope instability in Nepal are tectonic activity (through rock movement and earthquakes), and rainfall (resulting in mass erosion and the reduced strength of rocks and soils) (Shakya and Niraula, 2008; Hearn and Shakya, 2017). The most well-documented anthropogenic activity contributing to slope instability in Nepal is the rapid and haphazard construction of roads (Hearn, 2002; Shakya and Niraula, 2008; Hearn and Shakya, 2017; McAdoo et al., 2018).

Many of the landslides in Nepal occur along the road network and are road cut slope failures (McAdoo et al., 2018; Robson et al., 2024). According to Hearn (2011), 70% of slope failures on mountain roads are cut slope failures rather than larger

'natural' landslides. A road cut slope is a slope that has been excavated adjacent to a road that is often steeper than the surrounding topography. The construction and widening of roads in mountainous regions often results in road cut slopes (Hearn, 2011). When road cut slopes fail (a type of landslide) it can result in substantial economic, environmental, and societal loss through slope debris colliding with pedestrians, vehicles, and infrastructure, and through debris blocking the road (resulting in a delay to people accessing jobs and services, to emergency responders, and to the transport of goods) (Hearn, 2002; Petley et al., 2007). Figure 1 displays examples of such cut slope failures in Nepal.

Paudyal et al. (2023); Robson et al. (2022, 2024) suggest that the extensive cut slope failures on the Nepal road network can be partly blamed on the limitations of current guidelines used by engineers to design the geometry of cut slopes in Nepal. The geometry of a cut slope, specifically its inclination, plays a crucial role in its stability. Generally, the shallower the inclination of a slope, the more stable it is. However, a lower inclination requires more excavation and space and is, therefore, more costly. Design guidelines can be used to establish an optimal cut slope inclination based on different slope characteristics (e.g. the strength of the slope material) to identify a stable inclination that is not overly conservative. The limitations of the current guidelines used in Nepal outlined by Paudyal et al. (2023); Robson et al. (2022, 2024) include not incorporating important ground strength characteristics, strength characterization being too broad, a lack of suitable descriptions, and often being presented in inaccessible formats.

In this paper, we present the methods and outcomes of a participatory study conducted with Nepali engineers to establish their experience and perspective on the current use of road cut slope design guidelines in Nepal and how they can be improved. The outcomes of this paper will be used to inform the development of new guidelines for road cut slope design tailored towards the needs of engineers in Nepal. We decided to conduct this participatory study (research involving the participation of people affected by the issues being researched, Cornish et al. (2023)) with Nepali road engineers to ensure that these guidelines are tailored towards their needs. The new guidelines are to be developed as a collaboration between the Centre for Disaster Studies, Institute of Engineering (IoE) at Tribhuvan University, Nepal, with the Institute of Hazard, Risk and Resilience (IHRR) at Durham University, UK, supported by Mott MacDonald UK, Nepal Geotechnical Society, and the Department for Local Infrastructure (DoLI), and funded by the EPSRC Impact Acceleration Account. It is hoped that the new guidelines, informed by this study, will be used by engineers to design safe cut slopes to lessen road cut slope failures and, therefore, improve the resilience of the Nepal road network. This will contribute to the Sendai Framework for Disaster Risk Reduction 2015-2030 (SFDRR) priority to strengthen disaster risk governance to manage disaster risk through the use of clear guidelines (UNDRR, 2015).

The participatory study utilized a range of qualitative data collection methods, including a one-day workshop, semi-structured interviews, unstructured interviews, focus groups, and questionnaires. The research was undertaken with road engineers working for a range of different agencies and organization types. The qualitative research methods are outlined in Sect. 3, and the outcomes of the research are presented in Sect. 4 and discussed further in Sect. 5. Based on the outcomes of this research, we present general recommendations for cut slope guideline development in Sect. 6.1, and technical recommendations for guideline development specifically in Nepal in Sect. 6.2. These recommendations have been shared with the Department of Roads and the Department of Local Infrastructure at the Government of Nepal. The following section presents background on land-

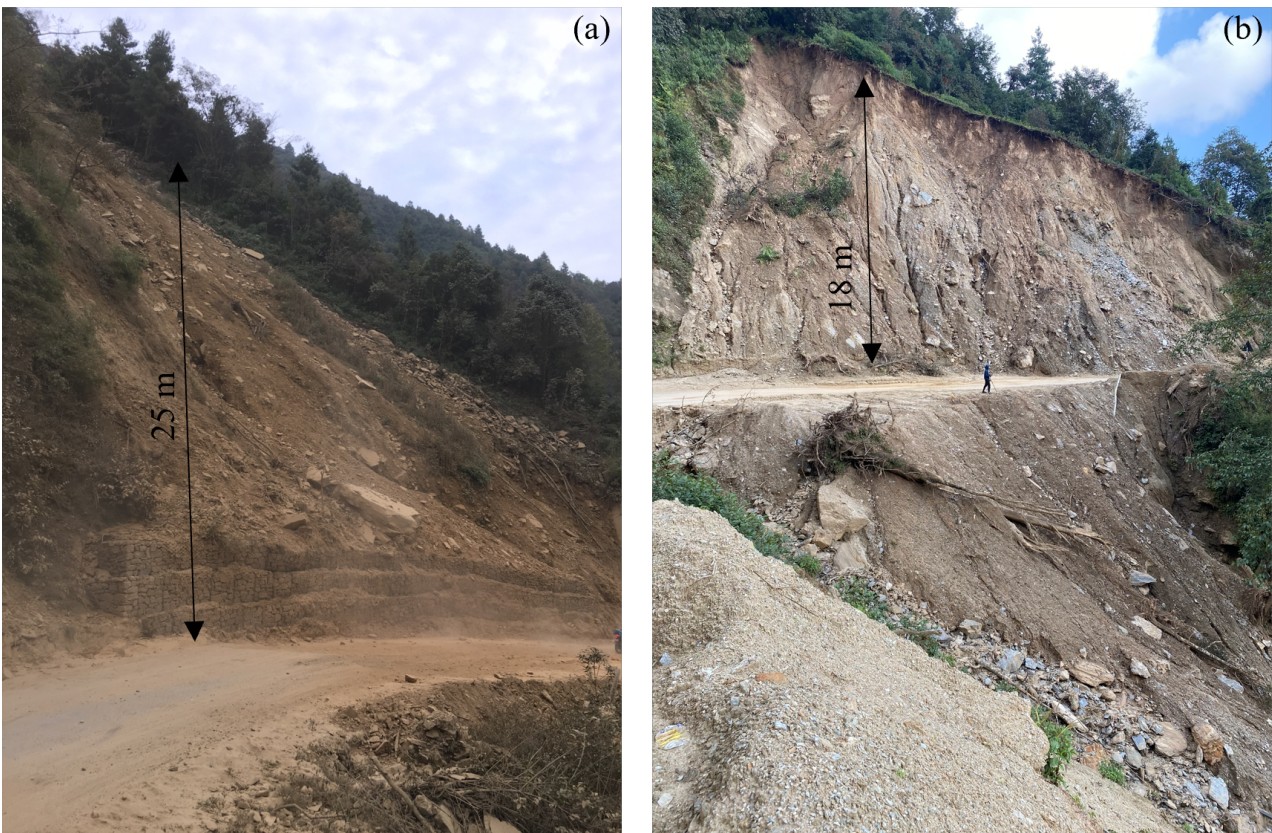

**Figure 1.** Photos of road cut slope failures in Nepal on: (a) the Kulekhani-Pharping feeder road in the Makwanpur District (photo taken by the corresponding author in November 2019); and (b) the Charikhot-Jiri road in the Dolakha District (photo taken by the corresponding author in November 2023).

slide and road construction in Nepal (Sect. 2.1), disaster risk and road slope management (Sect. 2.2), road slope management in Nepal (Sect. 2.3), and the use of participatory studies to improve disaster management (Sect. 2.4).

## 2   Background

### 2.1   Landslides and road construction in Nepal

Earthquakes are one of the main triggers of landslides (typically shallow rock avalanches) in Nepal (Owen, 2018). The other major trigger for landslides in Nepal is rainfall. The annual monsoon season in Nepal runs from June to September during which 80% of Nepal's annual rainfall occurs. 93% of landslides in Nepal occur during this four-month monsoon season (Froude and Petley, 2018). This dominance of landslides occurring during the monsoon was also highlighted by KC et al. (2024) who undertook an analysis of landslides that occurred in Nepal from 2011 to 2020. KC et al. (2024) also found a significant

increasing trend in landslide occurrence in Nepal between 2011 and 2020 (landslide density was 0.85 events/1000 km$^2$ in 2011 and had risen to 3.34 events/1000 km$^2$ by 2020). The authors suggest that this trend is a consequence of the 2015 $M_w$ 7.8 Gorkha earthquake in Nepal which caused ground cracking resulting in a reduction of the ground material strength and, thereby, contributing to landslide occurrence. This suggestion was based on the finding that landslide occurrence within the 14 districts worst affected by the earthquake remains significantly higher than it was before the earthquake. Rosser et al. (2021), who used satellite images to map landslides in the 14 districts worst affected by the Gorkha earthquake up to post-monsoon 2019, found that in 2019 only roughly half of the mapped landslides in this region were triggered by the Gorkha earthquake. They suggest that the cause of remaining landslides should be attributed to the wider social and political context with anthropogenic activity that can exacerbate landslide susceptibility. They include the example of the boom in rural road construction activity that is anecdotally associated with the first 2017 local elections. Rosser et al. (2021) note that the damage resulting from the Gorkha earthquake will have created challenges in conducting sustainable road construction practices.

The density of Nepal's total road network has more than tripled in the last three decades due to significant national and foreign investments aiming to improve economic and social development in Nepal through road construction (Government of Nepal, 2017; Gurung, 2021). Constructing a fixed linear structure of a road on Nepal's dynamic landscape is hugely challenging. However, many scholars have suggested that haphazard practices in road construction in Nepal have aggravated landslide activity (Hearn, 2002; Shakya and Niraula, 2008; Hearn and Shakya, 2017; McAdoo et al., 2018). Robson et al. (2021) aimed to understand the issues around the coordination and protocol of implementing road slope stabilization that may lead to road cut slope failures by conducting qualitative data collection with stakeholders in road construction in Nepal. Key findings of this research were that roads were being haphazardly constructed, that there is poor communication between the key stakeholders, and that slope stabilization is not prioritized in road construction projects. Thus, and as argued by Rosser et al. (2021) and KC et al. (2024), landslides in Nepal are caused by a complex interplay of geophysical and geopolitical processes that are challenging to unpick and address. This contributes to the growing understanding that most disasters worldwide are caused by some interplay of physical and societal processes (Donovan, 2017; McGowran and Donovan, 2021).

### 2.2 Disaster risk and road slope management

Given that most disasters are rooted in a complex interplay of geophysical and geopolitical processes, they require management that addresses both the geophysical and geopolitical processes in a complimentary way (McGowran and Donovan, 2021; Lavell and Maskrey, 2014). This is highlighted by the SFDRR who state that measures to reduce disaster risk should be multisectoral and inclusive (UNDRR, 2015). This report also outlines that '*clear vision, plans, competence, guidance, and coordination within and across sectors, as well as the participation of relevant stakeholders, are needed to strengthen disaster risk governance*' (UNDRR, 2015).

A crucial component of the management of road-related landslide risk is the design and excavation of road cut slopes (Aydin et al., 2018; McAdoo et al., 2018; Hearn, 2002). This component involves the coordination of many stakeholders, including politicians who generally provide the resources, policymakers who set and enforce the standards, the engineers responsible for designing the cut slope, and the engineers responsible for excavating it.

In high income countries (HICs) standard practice for designing road cut slopes involves a detailed site investigation (including in-situ and laboratory testing) to determine the strength of the cut slope and surrounding land, and numerical stability analyses (a process to quantify the stresses of the slope to establish it's stability) of the cut slope to determine the optimal design taking into account the strength of the cut slope geomaterial (soil/rock), as well as spatial and budget constraints. The design will generally incorporate the cut slope geometry (inclination and height), a drainage system (drainage built within, on top of, or next to the cut slope to direct water inside or on top of the slope), and any additional structures (e.g. supporting walls and anchoring systems) implemented to improve stability. Normally, all steps in this design process are conducted in accordance with national or international design standards. For example, British Standards are used in the United Kingdom and these outline that the design of road cut slope stabilization should conform to the Eurocode 7 (European codes for Geotechnical Design) (The British Standards Institution, 2023). Design guidelines (e.g. Geotechnical Engineering Office (2011)) and stability charts (e.g. Wyllie (2017); Li et al. (2008)), used to determine stable cut slope inclinations based on ground strength, are often used in the preliminary design stage. It is important to note that we are generalizing and oversimplifying here by adopting HIC framing, a classification provided by the World Bank based on gross national income per capita (The World Bank, 2023). The design and excavation of road cut slopes in HICs is not perfect, and there will be a vast number of examples where the design is substandard with important steps outlined here missing. However, there is generally more political and economic capacity in HICs to accommodate these steps and adhere to standards. The behavior and relationship of development and disaster management is beyond the scope of this paper, but we suggest Simpson (2022) and Horner and Hulme (2019) for more information on this topic.

In low- and lower-middle-income countries (LIC/LMICs) in mountainous regions, again note that we are generalizing here by adopting this framing, the processes involved in road cut slope design and implementation can be hugely variable in technical rigor, due to having more variability in political and economic capacity. Often major roads have relatively large budgets, and design generally follows a process of geotechnical investigation and numerical analysis (similar to that in HICs), directed somewhat by design guidelines. However, for road projects with a smaller budget, cut slopes will often be designed following design guidelines in government and donor agency manuals (e.g. Department of Public Works and Highways (2007); Slope Engineering Branch (2010)), without additional geotechnical investigation nor numerical analyses (Robson et al., 2022; Hearn, 2002; Robson et al., 2021). However, Robson et al. (2022) documented that current guidelines in Nepal, as well as in other LIC/LMICs, lack technical rigor and usability. Sometimes the design of road cut slopes will be based only on a rule of thumb. 'Rule of thumb' refers to the engineer designing the cut slope based on their experience of what has worked in the surrounding area, and not following any guidelines nor conducting any investigation nor analysis. Hearn and Massey (2009) conducted geotechnical assessments of case studies in Bhutan and Ethiopia and found that very limited geotechnical assessments were carried out before road construction for low-cost roads in Bhutan and Ethiopia. Robson et al. (2021) conducted a series of interviews and discussions with key stakeholders in road slope stability in Nepal and found that local roads in Nepal are often excavated using a bulldozer with no prior slope design.

## 2.3 Road slope management in Nepal

The management of road slopes in Nepal is chiefly governed by the Department of Roads (DoR), local governments, and the Department of Local Infrastructure (DoLI). The DoR, a department in the federal Government of Nepal, is responsible for the management, planning, and maintenance of the strategic road network of Nepal which comprises highways (main trunk road connecting different regions, nationally and internationally) and feeder roads (connecting district roads to the highways). Engineers working for the DoR undertake road and bridge design and maintenance (including slope stabilization). The DoR often hire external consultancies to assist in the design phase of a project and hire contractors for the construction phase of a project. Local governments are responsible for the local road network of Nepal which comprises local roads (connecting settlements) and district roads (connecting local centers to district headquarters). The engineers working for the local governments undertake work on road and bridge design, construction, and maintenance of the local road network. They sometimes hire contractors for construction. The DoLI in the federal government provides technical support to some road projects. Prior to the Nepal federal government decentralization, DoLI was known as the Department of Local Infrastructure Development and Agricultural Roads (DoLIDAR) and provided technical support to local governments. International donor agencies and national development organizations also provide money and technical support for some road projects in Nepal.

The DoR has published multiple sets of standards and guidelines that include recommendations on stable cut slope inclinations (e.g. 'Nepal Road Standards 2070' (Department of Roads, 2013), 'Roadside Geotechnical Problems: A practical guide to their solution' (Department of Roads, 2007), 'Guide to road slope protection works' (Department of Roads, 2003)). Figure A1 in Appendix A displays the tables of recommendations of cut slope inclinations within these guidelines. These are designed to be used by engineers working on road construction projects in Nepal to determine stable cut slope geometries.

Paudyal et al. (2023); Robson et al. (2022) both call for the redesign of road cut slope guidelines for Nepal. They suggest that current guidelines in Nepal do not account for important geotechnical characteristics, can lack suitable descriptions, are often generic, and are often presented in inaccessible formats. This can lead to the mischaracterization of geomaterials and, consequently, the design of unsafe cut slopes. Robson et al. (2022) presented a new methodology to develop road cut slope guidelines based on rigorous geotechnical characterization presented in an accessible format that requires only easily defined parameters to be used. Despite the marked improvement of these new guidelines in terms of geotechnical rigor and usability, we suggest that if new guidelines are to be used by engineers in Nepal, it is important to have a clear understanding for the current use of road cut slope guidelines in Nepal and how new guidelines should be tailored towards the needs of engineers in Nepal.

## 2.4 The use of participatory studies to improve management

As discussed, the SFDRR calls for clear guidance to strengthen disaster risk governance (UNDRR, 2015). In addition, they highlight the importance of the participation of relevant stakeholders for the efficient management of disaster risk. Participatory research approaches are effective means to incorporate relevant stakeholders in disaster risk governance and guidance (Ardaya et al., 2019; Folhes et al., 2015). For example, Ardaya et al. (2019) use participatory methods to ease communication between

the local populations living in flood risk areas of Rio de Janeiro in Brazil and the authorities to aid flood risk management. In this study, we will use participatory research to aid the development of clear guidance on road slope design to reduce the risk of road landslides, by incorporating the experience and opinions of a range of road engineers who will use the guidance.

Robson et al. (2022) discuss a range of LIC/LMIC's road cut slope design guidelines that lack in technical rigor or usability and, therefore, require an upgrade (e.g. the Philippines: Department of Public Works and Highways (2007); Malaysia: Slope Engineering Branch (2010); and Liberia: Ministry of Public Works (2019)). We suggest that a participatory study as presented in this paper should be conducted prior to the design of new guidelines for any of these LIC/LMIC's, to incorporate the experience and opinions of those using the guidelines in their development.

## 3 Material and methods

The qualitative research took place in March 2023. Figure 2 displays a flowchart highlighting the key steps in our methodology. The main component of the qualitative research was a one-day workshop, which was conducted alongside semi-structured interviews, unstructured interviews, focus groups, and questionnaires. The research participants included road engineers working for a range of different agency and organization types in Nepal, with varying levels of experience. Table 1 presents a summary of the qualitative data collection carried out as part of this research, including the type of data collection and who it was conducted with. We employed a range of qualitative data collection methods to enhance the validity and reliability of our findings, capturing the complexities of participants' experiences and perspectives. Each of the qualitative data collection methods are discussed in more detail below (Sects. 3.2 to 3.4). Qualitative data analysis methods are discussed in Sect. 3.5. This study has been conducted with ethical approval from Durham University. All workshop attendees and qualitative data participants were informed about the study's aims and objectives, how their data would be used, and how the data would be stored and protected. All workshop attendees and qualitative data participants signed a consent form after being given this information.

**Table 1.** Qualitative data collection categorized by data collection type and participant job type. Abbreviations: gov. = government.

| | Semi-structured interviews | Unstructured interviews | Focus groups | Questionnaires | Workshop presenters | Workshop attendees |
|---|---|---|---|---|---|---|
| Consultant | 5 | 3 | 0 | 0 | 2 | 6 |
| Local gov. engineer | 0 | 0 | 2 | 9 | 1 | 2 |
| Federal gov. engineer | 1 | 1 | 0 | 10 | 2 | 11 |
| Academic | 0 | 1 | 0 | 0 | 2 | 15 |
| Total | 6 | 5 | 2 | 19 | 7 | 34 |

### 3.1 Workshop

The one-day workshop took place on $28^{th}$ March 2023 at the Centre for Energy Studies, Pulchowk Campus, Tribhuvan University organized by the authors of this paper. The workshop was conducted to bring together engineers and academics working

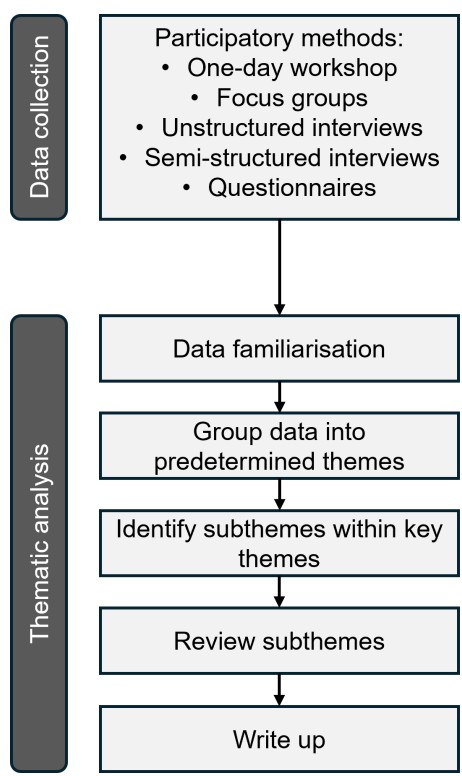

**Figure 2.** A flowchart highlighting the key steps of the method for this participatory study.

on road slope stability, but working for a range of different agencies and organizations, to allow rich discussion and scrutiny of certain topics, as well as knowledge-sharing. There were 34 participants at this workshop including six engineers working for consultancies, two engineers working for the local government, 11 engineers working for the federal government, and 15 engineers working for different academic institutions. Seven 20-minute presentations took place, each followed by 10 minutes of discussion. Two presentations were given by academics, three by engineers in the federal government, and two by consultants. The presenters were in leadership roles in a range of key agencies and organizations and were able to give an overview and insight into the use of guidelines by the agencies and organizations that they represented. Presentations were mostly in English, with discussions occurring in a mixture of English and Nepali. Workshop minutes were typed in Nepali and subsequently translated into English by a student at Tribhuvan University. Further details on the workshop presentations and discussions is presented in Appendix B.

### 3.2 Focus groups

Two focus groups took place, both with groups of engineers at two different local government units in Nepal. There are 753 local government units (main offices) in Nepal. The focus groups were conducted to gather rich insights into the experiences of local government engineers. One focus group had 5 participants, and the other had 11. Gill et al. (2008) recommends

a maximum focus group size of 14. We invited all engineers who specialized in road design and construction at that local government unit who were available to attend. Participants had a range of experience and were employed at different levels within the local government at that unit. The focus groups were about an hour long and were conducted mainly in Nepali (handwritten minutes from the meeting were subsequently translated into English).

## 3.3 Interviews

An interview is a qualitative research method used to gather an in-depth account of the participants' experiences (Gill et al., 2008; Flick, 2018). We conducted interviews with Nepali road engineers to gather in-depth accounts of their experiences of road slope stability in Nepal. Six semi-structured interviews (guided by a list of predetermined questions) were conducted in English with: two consultants at a consulting firm specializing in road construction; a government engineer; and three consultants at a consulting firm specializing in all forms of civil engineering. Semi-structured interviews were employed since their flexibility allows for the interviewee to adapt to responses by asking additional questions to pursue further details (Bryman, 2016). The questions that guided these semi-structured interviews were written by the authors of this paper to address the predetermined themes, and are provided in the Appendix C. The majority of the questions are open-ended, and where they are not, they include an open-ended follow-up question. As suggested by Gill et al. (2008), the interview questions start with questions that are easy for the participant to answer. The interviews generally lasted around 30 minutes and were recorded with permission using a Dictaphone. They all took place in the participants' offices in Kathmandu.

Five unstructured interviews (not guided by a list of predetermined questions) occurred in English: one with an academic; one with a federal government engineer; and three with consultants. Unstructured interviews were employed where participants seemed more at ease with free-flowing discussion. These interviews all took place in the participants' offices in Kathmandu, and the minutes of the interviews were handwritten.

## 3.4 Questionnaires

Questionnaires incorporate a list of multiple-choice questions and are designed to efficiently collect information (Slattery et al., 2011). Nineteen questionnaires were conducted in this study: nine with local government engineers and 10 with federal government engineers. The questionnaires included 17 multiple-choice closed questions (outlined in Appendix D) and were in English. The questions were written by the authors of this paper, and designed to address the predetermined themes. 14 out of 17 questions allowed the respondent to include an answer not offered on the form. Around half of the questionnaires were completed at a conference in Kathmandu (conference attendees were selected at random), whilst the other half were completed at local government offices. All questionnaires were completed individually and they took around five minutes to complete on average.

## 3.5 Data analysis

All semi-structured interviews were manually transcribed. Handwritten minutes from focus groups and unstructured interviews were typed up. The questionnaire answers were recorded in a Microsoft Excel spreadsheet. The workshop minutes were already typed up and a summary of these was sent to workshop attendees in the days following the workshop.

Thematic analysis is a commonly used method to identify and analyze themes within qualitative data (Braun and Clarke, 2006). We followed five key steps for thematic analysis that were adapted from Nowell et al. (2017): (a) familiarising ourselves with the data; (b) grouping the data into the predetermined themes (deductive); (c) identifying initial subthemes within the main themes (reductive); (d) reviewing these subthemes; and (e) writing up the findings. We utilized a deductive approach to initially group our data as we had clear objectives for the use of the data (i.e. to inform the development of new guidelines that are suited to Nepali road engineers). Therefore, the data was grouped into predetermined themes that would help us reach this objective. This was done by placing the data into the theme it best represented. The subthemes were then established by grouping the data within these themes based on their specific topics.

The four predetermined key themes are:

1. Guideline use (outcomes presented in Sect. 4.1 with further discussion in Sect. 5.1);

2. General slope stability practice (outcomes presented in Sect. 4.2 with further discussion in Sect. 5.2);

3. Opinions on the guidelines (outcomes presented in Sect. 4.3, with further discussion, combined with suggested improvements, in Sect. 5.3);

4. Suggestions for improvements to the guidelines (outcomes presented in Sect. 4.4, with further discussion in 5.3)

The subthemes are discussed as separate paragraphs within each of the main thematic outcome sections.

## 3.6 Limitations of the participatory study

A participatory study of any kind is subject to biases introduced by the involvement of participants, with the data and findings being skewed towards the participants' perspectives (Burgess, 2002). In our study, we tried to include participants working for a range of different organization and agency types with different levels of experience so that the data was not biased toward a particular group of road engineers. In addition, we used a range of data collection methods to ensure that we gathered perspectives that may have been alienated if only one data collection method was chosen. In doing so, we also reduced self-selection bias, the biases introduced when a study solicits participation from people, and those that take part are likely to differ from those that do not (Bermingham, 2020).

A key limitation of this study is that the majority of data collection activities (other than the focus group discussions and around half of the questionnaires) took place in Kathmandu. This occurred as Kathmandu acted as a convenient federal location for the workshop, with many engineers (particularly federal government engineers and consultancies) being based in or near Kathmandu. This means our findings are biased toward engineers who are based in Kathmandu. However, many of these engineers (the research participants) have experience working in many different regions across Nepal.

## 4 Qualitative research outcomes

Table 2 presents an overview of the main findings of the participatory study, categorized by the key themes. These findings are discussed in more detail in the following sections.

### 4.1 Theme 1: Guideline use

The questionnaire included a question asking if the respondents used any guidelines or manuals to design road cut slopes (Question 6, Appendix D). All 19 questionnaire respondents answered this question. The questionnaire responses to this question are displayed as a dendrogram in Figure 3, categorized by respondents' job type. Based on the responses, it can be suggested that federal government engineers mostly use guidelines for designing road cut slopes, whereas local government engineers are more inconsistent in their use of guidelines.

However, five of the presenters at the workshop suggested that the guidelines published by the DoR were not followed by Nepali road engineers, with road cut slopes being excavated to a steeper inclination than advised, and subsequently left unprotected.

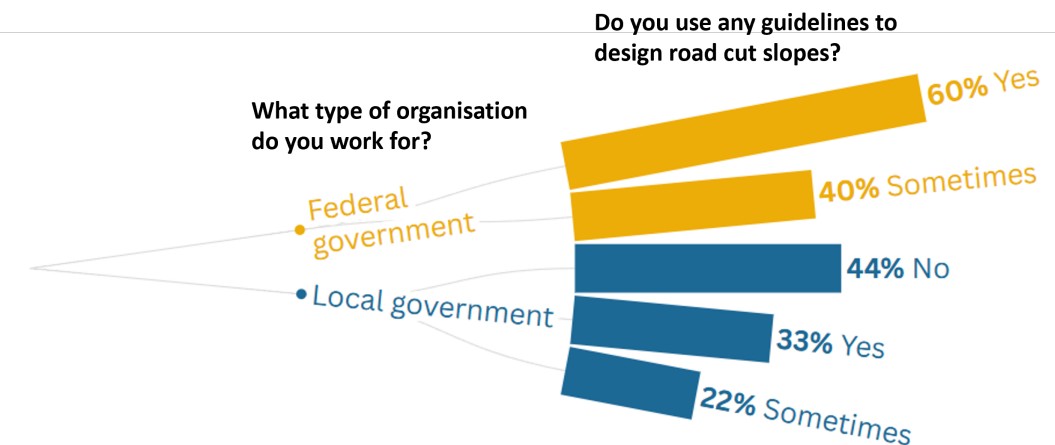

**Figure 3.** Dendrogram of questionnaire results for question on guidelines use. First level: What type of organization do you work for? (Question 1, Appendix D); Second level: Do you use any guidelines to design road cut slopes? (Question 6, Appendix D).

If questionnaire respondents answered 'No' or 'Sometimes' to whether they use guidelines, they were asked how they decide on the cut slope inclination (Question 10, Appendix D). The options for answers included 'Rule of thumb' (design based on experience rather than using guidelines), 'Numerical modeling', 'Stability chart', and 'Other, please specify'. Out of the six respondents that selected that they 'Sometimes' use guidelines, three selected that they use a 'Rule of thumb' approach to determine cut slope inclinations, whilst two selected 'Other, please specify' and specified that they use 'Field judgment', and one did not respond. It is thought that the 'Field judgment' approach is similar to 'Rule of thumb', in that it is based on

**Table 2.** A table highlighting the main findings of the participatory study categorized into the key themes.

| Guidelines use | General slope stability practice | Opinions on the guidelines | Suggested improvements |
|---|---|---|---|
| Federal government engineers are more likely to use the guidelines than local government engineers. | Geotechnical investigation is not conducted at a sufficient level of detail. | Guidelines are hard to use in a field setting. | Guidelines should include the most critical aspects of slope engineering (e.g groundwater, run-off, vegetation, and geomaterial layers). |
| Local engineers often use a rule of thumb approach to design cut slopes. | Groundwater is not always accounted for in the cut slope design. | There are inconsistencies between different guidelines. | Guidelines should be developed for the specific geological, physiographic, and meteorological conditions of Nepal. |
| Local engineers face constraints in their slope designs due to land acquisition problems. | If groundwater is accounted for, a borehole, trial pit, spring lines, or Electrical Resistivity Tomography are used to estimate the groundwater table. | There is a lack of training on the use of the guidelines. | Guidelines should account for conditions upslope and downslope of the cut slope itself. |
| Guidelines are mostly used to design the geometry of cut slopes. | Mohr-Coulomb is mostly use to characterize the strength of soil, while Rock Mass Rating is used for rocks. | There is a lack of advocacy to use the guidelines. | All road engineers, including contractors and consultants, should be provided with training on the use of new guidelines. |
| Guidelines are used in both field and desk settings. | Geomaterials are incorrectly identified by engineers. | Guidelines do not address spoil disposal. | There needs to be improved advocacy for using the guidelines. |
| 'Nepal Road Standards 2070' (Department of Roads, 2013) seems to be the most commonly used guideline. | Contractors sometimes cut corners in the construction to save money. | Guidelines do not incorporate effect of groundwater. | Guidelines should be simple to follow and user-friendly. |
| | The government prioritises funds for the construction and widening of a road, rather than slope stabilisation. | There are no guidelines for slope benching. | There should be multiple sets of guidelines for different types of engineers to reflect their varying needs and resources. |
| | The government sometimes use road construction as a political bargaining tool. | | Guidelines should address land acquisition problems. |
| | | | Guidelines should specify the need for quality assurance checks. |
| | | | Guidelines should incorporate protocol on spoil disposal. |

experience without the use of guidelines. However, 'Field judgment' emphasizes the importance of using site-specific field observations. All four respondents who selected that they did not use guidelines selected that they use a 'Rule of thumb' approach to determine road cut slope inclinations.

In both focus group discussions with local government engineers, it was revealed that they use a rule of thumb approach to design cut slopes. However, they outlined that cut slope inclinations are often dictated by land acquisition problems. They

have to offer financial compensation if they need to excavate into privately owned land. However, they do not have enough money to offer compensation in all projects. This results in initial safe cut slope designs being compromised, with steeper (more unstable) cut slopes that save space being implemented. This was also highlighted as a major constraint in the design of safe cut slopes at the workshop.

The questionnaire respondents that answered 'Yes' or 'Sometimes' to whether they use guidelines were asked what aspects of

295 slope design they use guidelines or manuals for (Question 9, Appendix D). Options for answers included 'Cut slope inclination', 'Retaining walls', 'Anchoring systems', 'Drainage', and 'Other, please specify', and they could select multiple answers. 14 respondents answered this question. Out of those 14, 13 respondents selected 'Cut slope inclination', nine selected 'Retaining walls', six selected 'Anchoring systems', seven selected 'Drainage', and no respondents selected 'Other, please specify'. This implies that when engineers use guidelines, it is mainly for road cut slope inclination design.

The questionnaire respondents that answered 'Yes' or 'Sometimes' to whether they use guidelines were also asked where they use the guidelines (Question 8, Appendix D). Five respondents selected that they used the guidelines in the field, five selected that they used the guidelines at a desk, and four selected that they used the guidelines both in the field and at a desk.

The questionnaire respondents that answered 'Yes' or 'Sometimes' to whether they use guidelines were asked to select which guidelines they use (Question 7, Appendix D). There were five options of guidelines to choose from including:

1. 'Nepal Road Standards 2070' (Department of Roads, 2013)

2. 'Roadside Geotechnical Problems: A practical guide to their solution' (Department of Roads, 2007)

3. 'Guide to road slope protection works' (Department of Roads, 2003)

4. 'Mountain Risk Engineering Handbook' (ICIMOD, 1991)

5. Other, please specify

Respondents could select multiple answers. Ten respondents only selected the 'Nepal Road Standards 2070' (Department of Roads, 2013). The other three selected more than one manual out of the prescribed answers. No respondents selected 'Other, please specify'.

In both focus group discussions with local government engineers, they said that if they were to use guidelines (rather than rule of thumb), they would use those published by the DoR, in particular the 'Nepal Road Standards 2070' (Department

of Roads, 2013). In a semi-structured interview with two consultants, the consultants said that they use the 'Mountain Risk Engineering Handbook' (ICIMOD, 1991) as well as 'Nepal Road Standards 2070' (Department of Roads, 2013) for road cut

slope design and geotechnical investigation. In a semi-structured interview with three consultants, they outlined that they use the Indian Standards (IS) for cut slope design. In a semi-structured interview with a federal government engineer, they outlined that they use 'Nepal Road Standards 2070' (Department of Roads, 2013) to design cut slopes, however, they noted that:

320   "*...although we are using the guidelines, we are facing several in several stability problems again and again. So we have to revise it depending upon the practical experiences from the construction site.*"

Although no international consultants were interviewed, it was highlighted in a semi-structured interview with three consultants at a private firm that international experts working for international donor agencies or international consulting firms use their own company/organization's guidelines and standards, rather than those developed in Nepal. They suggested that these

325   international guidelines and standards have a higher safety factor than the DoR guidelines as:

"*they don't want to see their designs being failed. Otherwise, they will always be questioned by those agencies to the international consultant, their reputation goes linked.*"

## 4.2   Theme 2: General slope stability practice

The focus group discussion with local government engineers revealed that, in general, they do not conduct any form of geotech-

330   nical investigation due to financial and time constraints. In addition, they do not conduct numerical stability analyses. During a focus group at one local government unit, it was revealed that they receive more than 100 applications from residents in their municipality requesting road improvements (mainly slope-related) in one monsoon season, but they do not have the funds nor time to complete all of these, so they prioritize requests based on what has the highest demand. In a semi-structured interview with two consultants from a private firm, they suggested that geotechnical investigation is not conducted at a sufficient level of

335   detail in Nepal, resulting in incorrect or poor geotechnical data. They thought insufficiently detailed geotechnical investigation occurs due to a lack of funding.

Groundwater has a significant influence on the stability of a slope. When questionnaire respondents were asked if they consider groundwater in their slope mitigation design, four answered 'Yes, every time', seven answered 'Yes, sometimes', six answered 'No', and two did not answer (Question 12, Appendix D). See Figure 4 for these answers categorized by respondents'

340   job type. If the respondents' answers included 'yes' to whether they consider the groundwater table in their design, they were asked what method they use to determine the water table height (Question 13, Appendix D). Seven answered 'Borehole', three answered 'Trial pit', one answered 'Assess geomorphology' and one answered 'Other' with no further detail. If respondents answered 'No' to whether they consider groundwater in their design, they were then asked why they did not. Answers included lack of equipment, too costly, lack of manpower, and not enough time, with lack of equipment being the most popular answer

345   (in five out of seven responses). Focus groups with local government engineers revealed that they used spring lines to identify the groundwater table as they did not have sufficient funds to dig boreholes.

Two consultants in a semi-structured interview said that they regularly use Electrical Resistivity Tomography (ERT), a geophysical method to determine the subsurface resistivity distribution, to identify the water table. They highlighted that ERT is a relatively affordable method in Nepal. They then include the water table in numerical stability analyses. In another semi-

350   structured interview with consultants at a different firm, it was revealed that they just used an estimate of the groundwater table

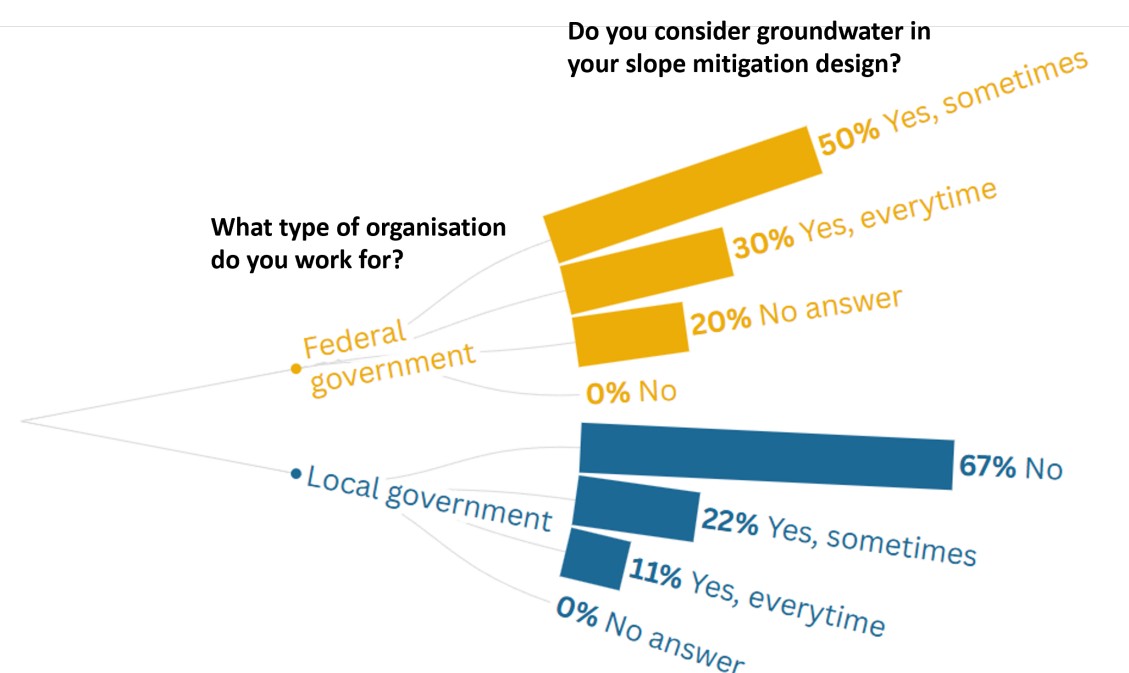

**Figure 4.** Dendrogram of questionnaire results on how groundwater is included in slope stability design. First level: What type of organization do you work for? (Question 1, Appendix D); Second level: Do you consider groundwater in your slope mitigation design? (Question 12, Appendix D).

**Table 3.** Questionnaire results for the question 'How do you characterize the strength of soil and rock?' (Questions 4 & 5, Appendix D).

| | How do you characterize the strength of soil and rock? | | | |
|---|---|---|---|---|
| | Mohr-Coulomb | Generalized-Hoek-Brown | Rock Mass Rating | Other |
| Soil | 15 | 1 | 0 | 3 |
| Rock | 6 | 1 | 10 | 2 |

established by locating the position of springs. Based on the qualitative data collection specific to groundwater, it seems that the handling of groundwater in the investigation and design of road cut slopes hugely varies in Nepal.

The questionnaire respondents were asked how they characterize the strength of soil and how they characterize the strength of rock (Questions 4 & 5, Appendix D). The multiple-choice options listed for this question included well-known failure criteria for rocks and soils. Failure criteria are used to quantify the stresses of a rock or soil mass at failure. Table 3 displays the results of these questions. Two respondents selected 'Other, please specify' for both questions, but did not give extra detail on what method they used. One respondent who selected 'Other, please specify' for the characterization of soil specified that they characterized soil with visual inspection.

Local government engineers in the focus groups said that they do not use any failure criterion in the design of road cut slopes. The consultants interviewed said that they use the Mohr-Coulomb failure criterion to characterize soil, and the Generalized-Hoek-Brown failure criterion or the Rock Mass Rating system to characterize rocks.

It was discussed at the workshop that geomaterials are often incorrectly identified by practitioners in the field. This was blamed on a lack of geological knowledge of the engineers, as well as insufficient geotechnical investigation being conducted prior to the design. This was also highlighted in a semi-structured interview with three consultants at a private firm, who said:

"*...Department of Roads, do not have any geologist, they have many geotechnical engineers, but the geotechnical engineers are not always capable of identifying the potential kind of slope problems*".

However, it was emphasized in a different semi-structured interview with consultants and in two unstructured interviews with consultants that Nepali geotechnical engineers have a good understanding of the road slopes in Nepal, but a lack of resources or time leads to incorrect data and/or design. In an interview with two consultants, it was said:

"*please don't think these Nepalese engineers, or these Nepalese engineering geologists or geotechnical engineers, they don't know how to work with the slope. They know very well. Because this is their own terrain.*

Questionnaire respondents were asked whether the cut slope inclination design is adhered to by construction workers who are managed by contractors (Question 11, Appendix D). 16 questionnaire respondents selected 'Yes, but not very accurately', whilst two selected 'Yes, very accurately' and one selected 'No'. In a semi-structured interview with two consultants and in two presentations at the workshop, it was suggested that contractors try to cut corners to save money on projects. In a semi-structured interview, a government official stated:

"*contractors will think about the money only, they want to save money. And when going for saving money, they will do the things in an unsystematic manner so that the failure may occur*".

A key point highlighted in an unstructured interview with a consultant, an unstructured interview with an academic, and two semi-structured interviews with consultants at private firms was that the Nepali government's priority is to construct and widen roads and that not enough time and resources are given to improving the stability of cut slopes. In a semi-structured interview, one consultant said:

"*Department of Roads or the Ministry of Physical Planning, whatever the case is, they don't give priority to the cut slope studies and stabilization, they just care about the pavement. Okay, pavement, how wide it is. That's all. That is the problem.*"

At a focus group with local government engineers, it was highlighted that the government often provides a lump sum of money to consultants or contractors for a project, but does not specify what it should be spent on. This results in underfunding of slope stability works, with the priority in spending on the lengthening and widening of roads (excavation and construction).

Furthermore, one of the presenters at the workshop, as well as consultants in interviews, suggested that the government uses road construction as a political bargaining tool. Electoral candidates promise to construct roads to win votes in elections, but often only provide sufficient money to excavate the road, and not to stabilize the road cut slopes adequately.

| What are the current limitations of road cut slope guidelines in Nepal? | | | |
| --- | --- | --- | --- |
| | Do not include rock/soil descriptions | Hard to use in the field | Not accurate |
| Single answer | 2 | 10 | 1 |
| Multiple answers | 4 | 5 | 4 |

**Table 4.** Questionnaire results for the question 'What are the current limitations of road cut slope guidelines in Nepal?' (Question 15, Appendix D). Thirteen questionnaire participants selected only a single answer, five participants selected multiple answers, and one did not answer.

## 4.3 Theme 3: Opinions on the guidelines

The questionnaire included a multiple choice question on what the current limitations of the Nepali road cut slope design guidelines are (Question 15, Appendix D). The breakdown of questionnaire responses to this question are displayed in Table 4. Thirteen questionnaire participants selected only a single answer, five participants selected multiple answers, and one did

not answer. All questionnaire participants selected from the answers that were provided, rather than specifying an alternative answer under the option 'Other, please specify'. The most common answer was that the guidelines are hard to use in a field setting.

Four of the presenters at the workshop outlined that current guidelines are not adequate for the design of safe cut slopes. Workshop presentations and discussions highlighted that there are inconsistencies between the design guidelines published by

the DoR on slope inclinations. For example, 'Guide to road slope protection works' (Department of Roads, 2003) outlines that cut slope inclinations for soft rocks should be between $40^o$ and $63^o$, whilst 'Nepal Road Standards 2070' (Department of Roads, 2007) recommends that inclination for 'highly weathered rock' (considered as a soft rock) can be as low as $35^o$. 'Guide to road slope protection works' (Department of Roads, 2003) and 'Roadside Geotechnical Problems: A practical guide to their solution' (Department of Roads, 2007) outline that the maximum cut slope inclination for any soil cut slopes should be $51^o$,

while 'Nepal Road Standards 2070' (Department of Roads, 2013) outlines a maximum of $45^o$.

A lack of training on the use of the guidelines currently used in Nepal was highlighted as a key limitation by local government engineers in the focus groups. This point also featured in presentations at the workshop. In the workshop, it was discussed by participants that there is a lack of advocacy in the use of the guidelines by engineers working at all government departments and by governing officials.

At the workshop, four of the presenters outlined that the guidelines do not address spoil disposal adequately. Spoil disposal is the disposal of soil or rock material excavated and not reused onsite. It was outlined that often excavated material is often deposited downslope of the cut slope site. Depositing material in this way can cause additional instability to the entirety of the hillslope (Hearn et al., 2003).

It was also discussed at the workshop that current guidelines do not consider groundwater. Groundwater has a huge influence

on the stability of a slope. This is especially important in Nepal due to the annual monsoon season, during which 80% of Nepal's

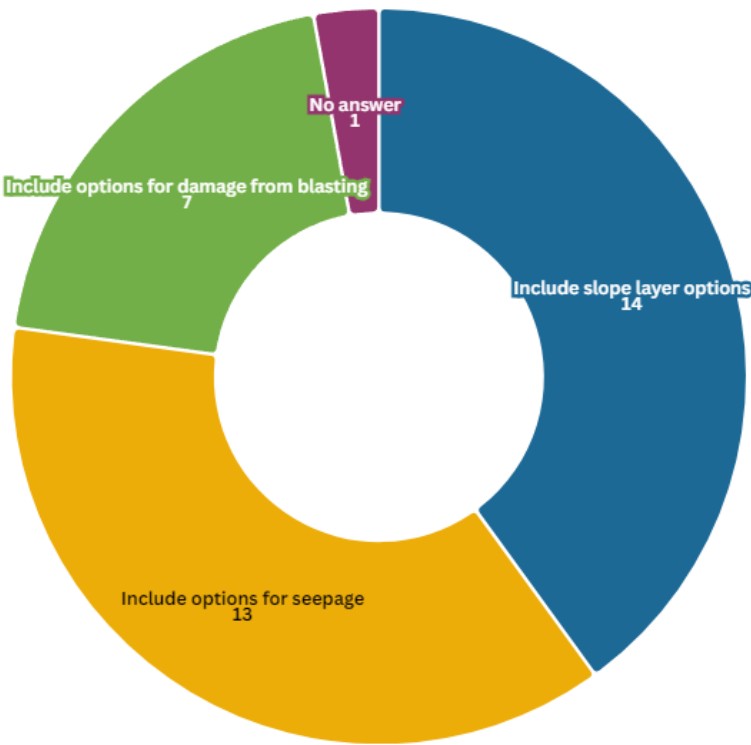

**Figure 5.** Questionnaire responses on the accuracy of the guidelines (Question 17, Appendix D). 18 respondents answered with one or more answers.

annual rainfall occurs during four months of the year (Shakya and Niraula, 2008). This heavy rainfall can drastically change the height of the groundwater table in the slope and, therefore, its stability.

In addition, it was also discussed at the workshop that current guidelines do not include standards for slope benching. Benching is a method of low-cost slope stabilization used globally where the cut slope is divided into a series of horizontal 420 steps, with near-vertical surfaces between steps.

### 4.4 Theme 4: Suggested improvements

The questionnaires included a question on how the accuracy of the guidelines could be improved (Question 17, Appendix D). Six respondents gave just one answer, 12 gave multiple answers, and one did not answer (see Figure 5). No respondents selected 'Other, please specify'. The responses suggest that engineers think the guidelines should include options for layers of 425 different geomaterial in the slope, as well as options for groundwater.

It was outlined in the focus group with local government engineers and at the workshop, that guidelines must include the most critical aspects of slope engineering and must be developed specifically for the geological, physiographic, and meteorological

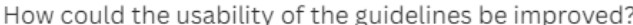

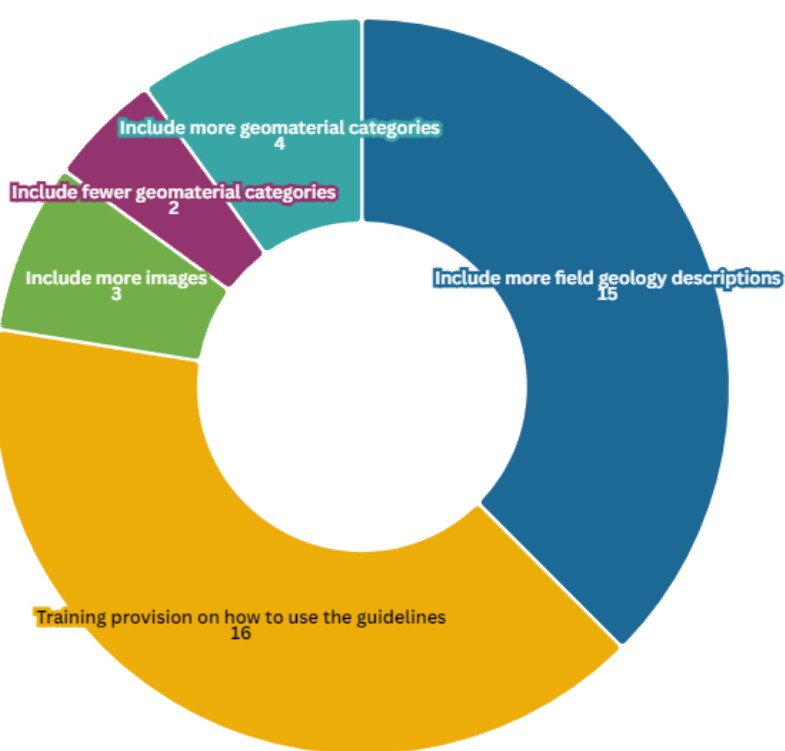

**Figure 6.** Questionnaire responses on the usability of the guidelines (Question 16, Appendix D) 19 questionnaire respondents answered with one or more answers.

conditions of Nepal. Critically, workshop participants outlined that groundwater needs to be incorporated into new road cut slope design guidelines, given the importance of pore water pressure on slope stability. Local engineers at the focus groups
also thought that guidelines should include details on vegetation and run-off. Workshop participants discussed the need for the guidelines to be developed in a way that accounts for the geology, geomorphology, and hydrology of the areas upslope and downslope of the cut slope itself, as the characteristics and behavior of these areas can affect the stability of the cut slope.

Questionnaire respondents were also asked how the usability of the guidelines could be improved (Question 16, Appendix D). All respondents answered (see Figure 6). Seven respondents gave only one answer, while 12 gave multiple answers.
No respondents selected 'Other, please specify'. The responses suggest that engineers want more training on the use of the guidelines, as well as for the guidelines to include more descriptions to characterize the geology.

The local government engineers in the focus groups, consultants interviewed and workshop participants agreed that a program of training should be rolled out with the publication of new guidelines and that this should involve all engineers involved in road construction in Nepal, including contractors and construction workers. The local government engineers in the focus
groups and consultants in a semi-structured interview suggest that training on slope stability in general should be offered to

construction workers operating on road projects. In addition, the consultants interviewed and workshop participants emphasized that there needs to be improved advocacy for the use of the guidelines by engineers, as well as by governing officials working for federal government departments.

It was emphasized in focus groups with local engineers, in interviews with consultants and, at the workshop that new guidelines need to be simple to follow and user-friendly for all engineers in Nepal. In a semi-structured interview with an engineer working for the federal government, he said:

"*if the guideline is friendly to the contractors, construction engineers, then they will follow otherwise they may skip it. If it is time taking any money, more money, investment things then they will skip it.*"

Some participants at the workshop suggested that there should be multiple sets of design guidelines developed, each designed specifically to suit engineers working for different government divisions (i.e. guidelines specifically made for local government engineers and guidelines specifically made for federal government engineers), to reflect their varying needs, resources and challenges.

At the workshop, participants discussed the need to address the problem of land acquisition in new design guidelines. Local governments often cannot afford to compensate land-owners where cut slopes need to be excavated into their land for stability, resulting in steeper (more unstable) cut slopes being made. It was suggested that private land-owners should be made more aware of landslide risk reduction. It was also suggested that guidelines should specify the need for quality assurance checks to be carried out by geotechnical experts.

Another suggestion discussed by workshop participants and in four of the presentations was for the new guidelines to clearly address spoil disposal. It was suggested that a diagram should be included to show that excavated material should not be thrown down slope, but instead be deposited on stable ground or used in other construction projects. It was discussed that contractors should receive a clear plan for spoil disposal ahead of works.

Workshop participants also discussed the need for new road cut slope guidelines specifically for road benching.

## 5  Discussion

The participatory research presented here aims to establish the current use of road cut slope inclination geometry guidelines in Nepal to inform the development of new guidelines tailored towards the needs of engineers in Nepal.

### 5.1  Theme 1: Guideline use

A key finding in this qualitative data collection was that cut slope design guidelines are more likely to be used by federal government engineers and consultants than by local government engineers (from questionnaire responses and interviews with consultants). 'Nepal Road Standards 2070' by Department of Roads (2013) was found to be the most widely used. Questionnaire responses found that the guidelines are mainly used to help the design of cut slope inclinations. Local government engineers often use a rule of thumb approach to design road cut slopes (based on questionnaire responses, as well as discussions at the focus groups). As previously discussed, the rule of thumb approach refers to the engineer designing the cut slope

based on their experience on what has worked in the surrounding area. The area surrounding a cut slope can certainly give important clues to the stability of that cut slope (e.g. the geological and hydrological context), however, this information alone

is insufficient to determine the design of a cut slope. In addition, engineers' experiences can differ substantially, meaning a rule of thumb approach is highly subjective and can result in the design of unstable slopes.

Importantly, it was found (through the focus groups with local government engineers) that the design and implementation of road cut slopes by local government engineers is heavily constrained by problems with land acquisition. Road cut slopes end up being steeper than an initial stable (less steep) design due to not having the funds to acquire and excavate into private property.

This is specifically a problem for local governments, as the DoR has a larger budget for land acquisition and compensation. A suggestion to improve this issue is for resources to be made available to local governments so that they can acquire the land necessary to excavate shallower cut slopes. The SFDRR outlines the importance of *'empower local authorities, as appropriate, through regulatory and financial means to work and coordinate with civil society, communities, and indigenous peoples and migrants in disaster risk management at the local level'* (UNDRR, 2015, p. 18). 'Overseas Road Note 16' by Hearn and

Lawrence (1997) states that compensation for land acquired is usually equated to the expected yield of the land during the design life of the scheme. However, Hearn et al. (2003) discusses the complexities of land acquisition noting that even if landowners are compensated, they could continue to use the land above the slope, which could result in destabilization (e.g. crop irrigation). Hearn et al. (2003) also highlights the need for the road authority to *'take an active role in the management of land inside and outside the right of way through discussions with farmers, landowners, and the local authorities'* (Sect. 4.3.2.8).

The DoR need to consider all the aforementioned points to define and clarify a protocol for land acquisition, specifically for the local government engineers.

In the workshop, it was discussed that cut slopes are often steeper than recommended by the current guidelines (Sect. 4.1). This was also a key finding of Robson et al. (2024) in evaluating the inclination of road cut slopes in Nepal against the advised cut slope inclinations in DoR guidelines. This may be due to contractors trying to cut corners in projects to save money

as highlighted in focus groups and interviews (Sect. 4.2). It could also be due to incorrect identification of geomaterials by engineers (Sect. 4.2), which suggests that they have not received sufficient training on how to use the guidelines (this key point was commonly highlighted in the workshop), and also suggests that there may be insufficient training on geomaterial identification in general.

Incorrect identification of geomaterials may also be due to guidelines being difficult to use in the field setting, which was

indicated by the questionnaire results and discussed at the workshop (see Sect. 4.2). This is an important finding since our data collection also reveals that engineers use the guidelines both as part of the desk and field study (see Sect. 4.1). The existing guidelines do not specify whether they are intended to be used in a field or desk setting, however, 'Roadside Geotechnical Problems: A practical guide to their solution' by Department of Roads (2007) provides field assessment forms, indicating that it was developed to be used in the field. We suggest that the guidelines provided by the DoR are hard to use in a field setting as they

present tables in different formats, the geomaterial descriptions are not always clear and key information about what cut slopes they are applicable to is missing. As highlighted by Robson et al. (2022), the original source of Table C3.6 (p. 12) in 'Roadside Geotechnical Problems: A practical guide to their solution' (Department of Roads, 2007) is Hearn (2011) and 'Roadside

Geotechnical Problems: A practical guide to their solution' (Department of Roads, 2007) misses out the key information that this table of recommendations is only for cut slopes up to 10 m in height to achieve a FoS of 1. Incorrect identification of geomaterials could also suggest that engineers have not received sufficient training on how to use the guidelines, or that the guidelines are difficult to use in the field setting, which was indicated by the questionnaire results. This highlights that the current guidelines lack clarity and that some engineers lack the competence to use them, meaning that the current guidelines do not currently conform to the priorities set out by the SFDRR.

## 5.2 Theme 2: General slope stability practice

In the qualitative data collection focusing on general slope stability practice, it was found that geotechnical investigation is not common practice on local roads, and where it is conducted on other roads in Nepal, it is often not conducted at a sufficient level of detail, resulting in inaccurate data. A lack of funding for geotechnical investigation, and for road slope stabilization in general, may be due to under-funding by governments as they prioritize opening roads, rather than stabilizing road cut slopes along these roads (as found in the data collection and discussed in Sect. 4.2). Based on this finding, guidelines for the design of cut slopes up to a certain height should be designed with input parameters that do not need to be identified through in-depth geotechnical investigation. Beyond a certain cut slope height, the guidelines should state that a thorough geotechnical investigation is required.

Groundwater is not always considered in road cut slope design in Nepal, particularly not by local government engineers due to a lack of equipment and budget. Where groundwater testing is carried out, engineers use boreholes or ERT.

Where failure criteria are used to characterize the strength of the cut slope geomaterial, we find that the Mohr-Coulomb failure criterion is mostly used to characterize soil, whilst the Rock Mass Rating system is mostly used to characterize rock behavior. Therefore, new guidelines should be based on these criteria to make them user-friendly for engineers in Nepal. New guidelines should also be accessible and easy to use in a field setting, including clear advice on how to characterize the geomaterial and groundwater.

This study also highlighted how the Government of Nepal has potentially mismanaged road construction at times, resulting in road slope failures. It was highlighted in Sect. 4.2 that the government prioritizes rapidly expanding road lengths (and widths) over constructing well-designed roads with safe road cut slopes. This was also a key finding of Robson et al. (2021). Furthermore, it was suggested that politicians use roads as a political bargaining tool. This is also highlighted by Gurung (2021) in challenging infrastructural orthodoxies in the context of Himalayan roads, specifically in the Karnali Province of Nepal. We suggest that politicians in Nepal need to recognize the importance of road slope management in their disaster risk management and coordinate more effectively with other stakeholders in road slope management to build resilient roads. We also suggest that further investigation should be carried out on the political influences of road construction in Nepal and the impact that they have.

### 5.3 Theme 3 & 4: Opinions on the guidelines and suggestions for improvements to the guidelines

Participants of the study highlighted that guidelines should be developed with careful engineering consideration, whilst also being user-friendly to engineers with varying backgrounds. It was suggested that current guidelines do not adequately account for groundwater conditions and that new guidelines should do a better job of this. As in-situ testing of the groundwater is not common practice, if new guidelines are to include the effect of groundwater, this needs to be done in a way that does not require practitioners to conduct costly groundwater testing. It was also suggested that the guidelines should account for the
characteristics of the area upslope and downslope of the cut slope itself.

Engineers highlighted the need for new guidelines to include standards for benching. Globally, standards for bench heights in rock vary between 7 and 10 m in height. However, bench width standards are more variable depending on country specifics (Hearn, 2011). Benching standards specific to Nepal should outline bench width recommendations for cut slopes of typical geomaterial types in Nepal.

In addition, engineers at the workshop called for clearer advice in terms of spoil disposal. They suggested that a diagram for spoil disposal could be added to the guidelines. 'Landslide risk assessment in the rural sector: guidelines on best practice' by Hearn et al. (2003) specify that considerations to spoil disposal must be done during the feasibility stage of a construction project, with potential spoil areas identified and costing (quantities estimated) carried out. Protocol for spoil disposal needs to be clarified by the federal government.

The federal Government of Nepal also needs to establish a protocol to ensure that contractors do not cut corners in road construction works and that the work is carried out as per the agreed design. This can be dealt with by conducting quality assurance checks. Therefore, the protocol for quality assurance checks needs to be clarified by the DoR and highlighted in new guidelines, as well as during a training program. The SFDRR outlines that necessary mechanisms and incentives should be employed to ensure high levels of compliance with the existing safety-enhancing provisions of sectoral laws and regulations
(UNDRR, 2015). This protocol could be written into policy as a local law. For example, policy-makers in British Columbia, Canada, have enforced byelaws for the direct supervision of landslide assessments, as well as internal and external peer review of the landslide assessment (APEGBC, 2010). They state that direct supervision can *'typically take the form of specific instructions on what to observe, check, confirm, test, record and report back to the Qualified Professional'* (p. 30). They discuss that the internal review should be carried out by another qualified professional in the same firm, and the external review should
be carried out by someone independent. Another approach to encourage contractors not to cut corners is to provide them with training to improve their understanding of the importance of following a safe cut slope design.

It was suggested that if new guidelines are published there should be a program of training rolled out on their use, to engineers working for all agency and organization types across Nepal, as well as to contractors and construction workers. An example of a successful road slope stabilization training scheme was part of the South East Asia Community Access Project
(SEACAP 21) (Scott Wilson, 2009; Hearn et al., 2021). This project was conducted by the UK Department for International Development (DfID), now replaced by the Foreign, Commonwealth & Development Office (FCDO). It was a three-year project in Laos, commencing in 2006 aiming to improve road slope management practices (Hearn et al., 2021). They selected specific

problematic cut slope sites in Laos to implement low-cost engineering mitigation methods. The training included a short presentation and a seminar on the work undertaken by SEACAP 21 with postgraduate students from the National University of Laos (NUoL). They also conducted an assessment of courses offered at NUoL and provided recommendations for additional course content, as well as thesis topics. SEACAP 21 also provided field training to students and lecturers from NUoL, as well as to staff from the Department of Public Works and Transport. As part of the field training, participants visited the SEACAP 21 field sites where 'experts' explained their approach to the site assessment. A training course for road maintenance engineers in Laos took place in 2019, revisiting and evaluating the success of the SEACAP 21 sites. If training is going to be rolled out with the publication of new guidelines in Nepal, it should be integrated into the university curriculum for the geotechnical engineering master's program (currently only offered at the Institute of Engineering on Pulchowk Campus at Tribhuvan University), as well as through conducting field training with practicing engineers.

A key limitation of the existing guidelines highlighted during the data collection is the inconsistencies in the recommended cut slope inclinations between existing DoR guidelines. This is problematic as engineers are using different guidelines, as highlighted in the questionnaire (see Sect. 4.1). If new guidelines are to be published, the DoR needs to advocate for the use of the new guidelines only. It was also suggested that generally, advocacy in the use of road cut slope design guidelines needs to improve from engineers themselves, as well as from governing officials. This could be encouraged during training, by emphasizing and explaining the economic and social benefits of using the new guidelines.

Engineers at the workshop suggested that different guidelines could be made specifically for engineers working for different agency and organization types. In doing so, they would be tailored towards the resources they have available to them and to their specific problems (i.e. protocol on land acquisition problems in guidelines for local government engineers). This should include developing guidelines that can be used by the excavators who lack geotechnical training, but are the people ultimately responsible for the cut slope geometry.

### 5.4 General discussion

As outlined in the introduction, Nepal's landscape is naturally dynamic (tectonically, meteorologically, and topographically), and excavating into this landscape to construct a road is a tremendously difficult challenge. However, new road construction and widening projects are currently widespread across Nepal. New guidelines are required to reduce the contribution that these road construction projects have to slope instability.

Given the challenging landscape and constrained resources (particularly in the case of local engineers), the guidelines should reflect the capabilities to excavate a stable slope. In some cases, it may be near impossible to excavate a completely stable slope (with a high Factor of Safety) without implementing a stabilization measure (e.g. an anchoring system) over the entire slope, which may be unrealistic. Therefore, there needs to be some degree of acceptance, which may come in the form of a risk assessment within the guidelines. Where there is potential for higher societal and economic loss (i.e. on a road with heavy traffic or next to infrastructure), a more conservative design should be implemented. This should be integrated into the guidelines.

Careful thought should also be given to the communication strategy used to publicize the new guidelines to engineers and the public. The reason for the development of new guidelines and the method employed in developing them should be communicated to engineers. The communication could hugely influence the take-up of new guidelines.

### 5.5 Next steps for guideline development

Based on this study, it has been decided that the guidelines developed by the collaboration between IoE and the IHRR will be specifically for local government engineers in Nepal given that these engineers are most in need of guidelines that are suitable for their specific level of training and limited resources. The guidelines will be developed in line with the key technical recommendations highlighted in Sect. 6.2.

The guideline development plan has five key stages: (1) desk study to define geographical zones of Nepal based on phys-
iographical, meteorological, and geological data; (2) fieldwork conducted across Nepal to evaluate the stability of cut slope scenarios in each of the geographic zones; (3) numerical stability analyses conducted on the cut slope scenarios to assess the stability of slopes and to extrapolate their physical conditions; (4) guideline development based on the output of the first three stages; and (5) dissemination of guidelines to engineers and improvements based on feedback. By gathering data on cut slopes in different geographic zones, we will account for interactions between geomorphic, tectonic, and climate processes and topog-
raphy on slope stability (Owen, 2018). The guidelines will be structured as a set of simple questions and corresponding simple field tests (with diagrams and photos) to help the engineers characterize the slope material and determine the most appropriate slope inclination. These guidelines will be made available in hard and soft copy. Where the suggested safe inclination cannot be achieved (e.g. due to land acquisition constraints), engineers will know that additional stability measures are required to prevent slope failure. The guidelines are being developed in consultation with the Department for Local Infrastructure (DoLI)
to ensure they are aligned with their needs. Once the four key stages of the guideline development are complete, a series of training sessions will be conducted with local government engineers.

## 6 Conclusions

### 6.1 Concluding remarks and general recommendations

The SFDRR outlines strengthening disaster risk governance as a key priority in managing disaster risk (UNDRR, 2015). They
highlight clear guidance as a crucial part of this priority. Robson et al. (2022) and Paudyal et al. (2023) suggest that the guidance for the design of road cut slopes in Nepal is not fit for purpose, and partly contributes to the risk of road-related landslides in Nepal. They call for a new set of guidelines to be developed to replace those currently used, to reduce the risk of road-related landslides in Nepal.

The Centre for Disaster Studies, Institute of Engineering (IoE) at Tribhuvan University, Nepal, is collaborating with the
Institute of Hazard, Risk and Resilience (IHRR) at Durham University, UK, to develop new design guidelines for road cut slopes in Nepal. This work is being supported by Mott MacDonald UK, the Nepal Geotechnical Society, and the Department

for Local Infrastructure (DoLI), and funded by the EPSRC Impact Acceleration Account. To ensure these guidelines respond to the needs of the stakeholders using them, this paper presents a participatory study to gather the experiences and perspectives of Nepali road engineers on the use of the current guidelines and how they can be improved.

The participatory research approaches included questionnaires, semi-structured interviews, unstructured interviews, focus groups, and a workshop with road engineers working for different agencies and organizations in Nepal. We found that federal government engineers are more likely to use the 'Nepal Road Standards 2070' DoR guidelines to design cut slopes, while local government engineers often resort to using a rule-of-thumb approach. Inconsistency in the use of guidelines can be blamed on their lack of user-friendliness (especially in a field context), inconsistencies between guidelines, and a lack of training on the use of the guidelines. In addition, it was found that local engineers are often constrained in their design as they do not have the budget to provide compensation to acquire the land required. We also found a lack of comprehensive geotechnical investigation by local government engineers further exacerbates the unreliability of slope designs.

This study highlights the roles and responsibilities that key stakeholders have in road slope management and improvements that these stakeholder groups can make to reduce the risk of road-related landslides. These improvements are relevant to other LIC/LMICs that need to improve the management of road-related landslide risk in line with the SFDRR (e.g. Bhutan and Ethiopia - Hearn and Massey (2009), the mountainous regions of India - Sana et al. (2024), Indonesia - Diara et al. (2022), Malaysia - Rahman and Mapjabil (2017)), and include:

1. Policymakers need to set standards and laws for road slope management processes, and *'encourage the establishment of necessary mechanisms and incentives to ensure high levels of compliance'* with these standards and laws (UNDRR, 2015, p. 17).

2. Policymakers need to define and clarify a protocol for land acquisition and compensation, quality assurance checks, and spoil disposal, and provide incentives to encourage compliance with this protocol.

3. Policymakers should also define the protocol and provide incentives for the uptake of clear guidelines.

4. We found that politicians can have a negative impact on landslide risk by prioritizing rapidly expanding road lengths (and widths) to gain popularity, instead of constructing well-designed roads with safe road cut slopes. We suggest that politicians can improve their priorities in road construction by coordinating more effectively with other stakeholders in road slope management and road users, and recognize road slope management as a key component in their disaster risk management protocol to commit to the SFDRR.

5. Engineers and technical specialists have a crucial responsibility in designing and excavating road slopes so that they do not contribute to landslide risk. The responsibility that they have in disaster risk reduction should be conveyed to them more clearly in their training.

The coordination of these key stakeholder groups is crucial to ensure that road slope management is effective in reducing the risk of road-related landslides.

## 6.2 Technical recommendations for guideline development

Based on the findings of the participatory study, we present the following key recommendations for the development of new road cut slope design guidelines for Nepal:

1. Guidelines should be produced using a rigorous geotechnical field and/or numerical analysis method, and be developed according to Nepal's geological, physiographic, and meteorological conditions.

2. Guidelines should be presented in a user-friendly format, and not be overly complicated.

3. Guidelines should incorporate the effect of infiltration and groundwater. However, they should recognize that detailed groundwater investigations are unlikely to be used.

4. Guidelines (and training) should include advice on spoil disposal that is clarified by the DoR.

5. Guidelines should state that an additional stabilization measure will need to be implemented where land cannot be acquired.

6. When guidelines are published, training of the use of the guidelines should be provided to engineers specialized in road slope stability working for all types of agencies and organizations across Nepal, including contractors and construction workers.

## 6.3 Future research recommendations

This study points out that politicians in Nepal use roads as a political bargaining tool. We suggest that further research should be 685 conducted to investigate how political influence in road construction can contribute to landslide risk. We have two suggestions for the main lines of investigation into this topic: (1) research conducted to understand how road construction varies over time within an election cycle, so that the impacts following an election can be anticipated; and (2) how the link between political concerns, road construction, and road failure varies across different parts of the country. As a starting point, we need to better understand the distribution of roads, road construction, and road cut slope failures in space and time.

We suggest that further research is needed on effective coordination and communication between stakeholders in road slope management.

This study also underscores the challenges of interdisciplinary work within disaster risk reduction (Donovan et al., 2023). We suggest that there is a need to develop vocabularies and best practices for interdisciplinary research and action at the intersection of roads, risks, and resilience.

Finally, we believe that this participatory study has successfully gathered the experiences and perspectives of Nepali road engineers on the use of the current guidelines and how they can be improved. However, as stated in the limitations section, this study (and participatory study of any kind) is subject to biases introduced by the involvement of participants. Despite this, we recommend that a participatory study of this kind can be replicated in other LIC/LMICs that need to improve the management of road-related landslide risk, to ensure that improvements are made in line with the needs of road management stakeholders.

 **Appendix A:  Nepali cut slope guidelines**

Figure A1 displays the tables of recommendations for cut slope inclinations within guidelines published by the Department of Roads (DoR) in the Government of Nepal. These are designed to be used by engineers working on road construction projects in Nepal to determine stable cut slope geometries.

**Table 3.1 Recommended Standard Slope Gradient for Cut Slopes** (a)

| Soil classification | | Cutting Height (m) | Slope Gradient (V:H) |
|---|---|---|---|
| Hard rock | | | 1:0.3 ~ 1:0.8 |
| Soft rock | | | 1:0.5 ~ 1:1.2 |
| Sand | Not dense (loose), poorly graded | | 1:1.5 ~ |
| Sandy soil | Dense, or well graded | Less than 5 m | 1:0.8 ~ 1:1.0 |
| | | 5~10 m | 1:1.0 ~ 1:1.2 |
| | Not dense (loose) | Less than 5 m | 1:1.0 ~ 1:12 |
| | | 5~10 m | 1:1.2 ~ 1:15 |
| Sandy soil mixed with gravel or rock mass | Dense, well graded | Less than 10 m | 1:0.8 ~ 1:1.0 |
| | | 10~15 m | 1:1.0 ~ 1:1.2 |
| | Not dense (loose), or poorly graded | Less than 10 m | 1:1.0 ~ 1:1.2 |
| | | 10~15 m | 1:1.2 ~ 1:1.5 |
| Cohesive soil | | Less than 10 m | 1:0.8 ~ 1:12 |
| Cohesive soil mixed with rock masses or cobble stones | | Less than 5 m | 1:1.0 ~ 1:1.2 |
| | | 5~10 m | 1:1.2 ~ 1:1.5 |

Note1: Recommended standard gradient is only indicative and detailed assessment and design of cut slopes should be carried out by an engineer. Silt is to be classified as cohesive soil.

**Table 11-5 Cuttings side slopes** (b)

| Soil type | Side Slope(vertical:horizontal) |
|---|---|
| Ordinary Soil | 1:2 to 1:1 |
| Disintegrated rock or conglomerate | 1:$^1/_2$ to 1:$^1/_4$ |
| Soft rock, shale | 1:$^1/_4$ to 1:$^1/_8$ |
| Medium Rock | 1:$^1/_{12}$ to 1:$^1/_{16}$ |
| Hard Rock | Almost vertical |

(c)

**Table C3.2 Preliminary Cut Slope Gradients (V:H) for cut height < 15 m**

| Soil classification | | Cut height (m) | | |
|---|---|---|---|---|
| | | < 5 m | 5-10 m | 10-15 m |
| Hard rock | | 1:0.3 – 1:0.8 | | |
| Soft rock | | 1:0.5 – 1:1.2 | | |
| Sand | Loose, poorly graded | 1:1.5 | | |
| Sandy soil | Dense or well graded | 1:0.8 – 1:1.0 | 1:1.0 – 1:1.2 | - |
| | Loose | 1:1.0 – 1:1.2 | 1:1.2 – 1:1.5 | - |
| Sandy soil, mixed with gravel or rock | Dense, well graded | 1:0.8 – 1:1.2 | | 1:1.0 – 1:1.2 |
| | Loose, poorly graded | 1:1.0 – 1:1.2 | | 1:1.2 – 1:1.5 |
| Cohesive soil | | 1:0.8 – 1:1.2 | | - |
| Cohesive soil, Mixed with rock or cobbles | | 1:1.0 – 1:1.2 | 1:1.2 – 1:1.5 | - |

Source: Guide to Slope Protection

**Figure A1.** Tables of recommendations of cut slope inclinations taken from the following guidelines published by the Department of Roads (DoR): (a) 'Guide to road slope protection works' (Department of Roads, 2003)); (b) 'Nepal Road Standards 2070' (Department of Roads, 2013); and (c) 'Roadside Geotechnical Problems: A practical guide to their solution' (Department of Roads, 2007).

## Appendix B: Further details of the one-day workshop

The one-day workshop was held on $28^{th}$ March 2023 from 10 am to 5 pm at the Centre for Energy Studies, Pulchowk Campus, Tribhuvan University in Kathmandu, Nepal. There were 34 participants at this workshop including six engineers working for consultancies, two engineers working for the local government, 11 engineers working for the federal government, and 15 engineers working for different academic institutions.

The workshop began with a welcome ceremony led by representatives from the Centre for Disaster Studies, Institute of
Engineering, Pulchowk Campus, Tribhuvan University. Following on from this were seven 20-minute presentations, each followed by 10 minutes of discussion with workshop participants. Two presentations were given by academics, three by engineers in the federal government, and two by consultants. The presenters were in leadership roles in a range of key agencies and organizations and were able to give an overview and insight into the use of guidelines by the agencies and organizations that they represented. The presenters working for the federal government, all represented different specialty branches within the
federal government. Table D1 presents the key points from the workshop presentations and the following discussions.

Table B1:  A table presenting an overview of the key points in the workshop presentations and discussions.

| Presenter | Key points of presentation | Key discussion points |
|---|---|---|
| Academic | a) After 2017, there was a dramatic increase in rural road development.<br>b) Slopes are often steeper than current guidelines advise, so guidelines are being ignored.<br>c) Research using numerical analyses to assess the stability of road cut slopes found that the stability of cut slopes is affected by rainfall, run-off, and vegetation. | a) As most landslides occur during the monsoon season, the guidelines must account for the effect of rainfall.<br>b) Drainage is crucial in slope stabilization.<br>c) Nepal should shift it's focus from constructing new roads to stabilizing existing slopes. |
| Consultant | a) Common issues include wrong classification of soil types by engineers in the field, not accounting for ground conditions, spoil disposal downslope, and lack of drainage.<br>b) Engineers are not designing according to the guidelines.<br>c) There is a need to motivate engineers to use guidelines.<br>d) Contractors want to complete the work quickly.<br>e) Site supervisors need improved training. | a) Need to have a discussion with contractors at the start of the project to stop downslope spoil disposal.<br>b) Guidelines should include a diagram for spoil disposal. |

| | | |
|---|---|---|
| Federal government engineer | a) Existing guidelines do not account for groundwater.<br>b) Cut slopes are steeper than guidelines suggest.<br>c) Example of the 'Fast Track' road where guidelines said slopes should be cut at 45 degrees, but they were cut at 60, and then failure occurred.<br>d) Guidelines are not appropriate for the Nepal setting. | a) Contractors try to save money in the construction phase, with the resulting excavation being different to the design. |
| Federal government engineer | a) The guidelines provided by the DoR are not correctly implemented.<br>b) Benching should be used more widely and guidelines specific to Nepal are needed<br>c) Common issues include there being a large time gap between cutting and support, guidelines not followed, over-excavation, and spoil disposal downslope | a) Land ownership may be a problem for benching.<br>b) Main focus of Nepalese roads is on constructing a road with a black top, rather than the slope stability. |
| Federal government engineer | a) Main issues in rural road construction includes there being a gap in holistic planning, the road network is increasing without considering road asset management, poor decision-making, and planning is based on political bargaining.<br>b) Land acquisition problems prevent safe cut slope geometries.<br>c) There is a lack of communication between the tiers of road network governance.<br>d) Local-level engineers are not following guidelines.<br>e) Spoil disposal downslope.<br>f) Guidelines should be easy to use without engineering calculation.<br>g) Need to improve advocacy in the use of guidelines.<br>h) No quality assurance checks are carried out at the local level. | a) Guidelines should be simple to use, but incorporate local geomaterials.<br>b) Needs to address land compensation.<br>c) Different guidelines should exist for different levels of engineer. |
| Consultant | a) Key issues include downslope spoil disposal, surface water, and adverse geological conditions.<br>b) The exact design of the cut slope is not carried out due to local influence.<br>c) Cut slope geometry is governed by land acquisition. | a) Need guidelines are needed.<br>b) Use of new guidelines should be mandatory. |

Table B1: A table presenting an overview of the key points in the workshop presentations and discussions. (Continued)

| Academic | a) Poor communication between stakeholders in road slope stability. | a) There needs to be regular training on the use of new guidelines. |
| | b) Cut slopes are steeper than current guidelines suggest. | |
| | c) The effect of groundwater needs to be incorporated in the guidelines. | |

 **Appendix C: Semi-structured interview questions**

1. What type of organisation do you work for?

2. What is the role of your organisation in road construction/remediation projects?

3. Please describe your role in the organisation

4. Please describe your training/education for this role

 5. What road slope stabilisation technique do you most frequently implement?

6. What do you think are the key reasons for slope stabilisation failure (the failure of engineering solutions) in Nepal?

7. What are the biggest challenges facing engineers in to stabilise road cut slopes in Nepal?

8. What strength criterion do you use to characterise rocks?

9. What strength criterion do you use to characterise soils?

 10. Do you consider the groundwater in your slope mitigation design?

11. If **yes**, how do you measure the groundwater table height?

12. If **yes**, how do include groundwater in your design?

13. If **no**, why not?

14. Do you use a manual or guidelines to design/stabilise a road cut slope?

 15. If **yes**, what aspect of road slope design is it most useful for?

16. If **yes**, which one and why this? Where do you use it?

17. If **no**, why not?

18. How do you decide on the cut slope inclination?

19. What are the problems in using current guidelines/manuals?

 20. How can the usability of guidelines be improved?

21. How can the accuracy of guidelines be improved?

22. How well do construction workers follow a design for the excavation?

23. What do you think could improve the design/construction of slope stabilisation in Nepal?

24. Is there anything else you would like to add?

25. Do you have any contacts that you recommend I speak with or interview?

**Appendix D: Questionnaire**

1. What type of organisation do you work for?

   ○ National government

   ○ Local government

○ Consultancy

   ○ Contractor

   ○ University

   ○ Other, please specify

2. What do you do in your role? (tick all that apply)

○ Field investigation

   ○ Design slope engineering interventions

   ○ Manage construction works

   ○ Research

   ○ Other, please specify

3. What is your academic background?

   ○ Master's degree

   ○ Undergraduate degree

   ○ School

   ○ Other, please specify

4. How do you characterise the strength of rock?

   ○ Mohr-Coulomb (M-C) failure criterion

   ○ Generalised-Hoek-Brown (G-H-B) failure criterion

   ○ Rock mass rating (RMR) system

   ○ Other, please specify

5. How do you characterise the strength of soil?

   ○ Mohr-Coulomb (M-C) failure criterion

   ○ Generalised-Hoek-Brown (G-H-B) failure criterion

- ○ Rock mass rating (RMR) system

- ○ Other, please specify

6. Do you use guidelines/manuals to design road cut slopes?

- ○ Yes (go to question 7, 8, 9)

- ○ No (go to question 10)

- ○ Sometimes (go to question 7, 8, 9, 10)

7. If **yes/sometimes**, what guidelines/manual do you use? (tick all that apply)

○ Nepal Road Standards 2070

- ○ Roadside Geotechnical Problems: A practical guide to their solution

- ○ Guide to road slope protection works

- ○ Mountain risk engineering handbook

- ○ Other, please specify

8. If **yes/sometimes**, where do you use the guidelines? (tick all that apply)

- ○ Field

- ○ Desk

- ○ Other, please specify

9. If **yes/sometimes**, what aspects of slope design do you use guidelines/manuals for? (tick all that apply)

○ Cut slope inclination

- ○ Retaining walls

- ○ Anchoring systems

- ○ Drainage

- ○ Other, please specify

10. If **no/sometimes**, how do you decide on the cut slope inclination?

- ○ Rule of thumb

- ○ Numerical modelling

- ○ Stability chart

- ○ Other, please specify

11. Is the cut slope inclination design followed by the construction workers?

      ○ Yes very accurately

      ○ Yes, but not very accurately

      ○ No

12. Do you consider groundwater in your slope mitigation design?

○ Yes, every time (go to question 13)

      ○ Yes, sometimes (go to question 13)

      ○ No (go to question 14)

13. If **yes**, what method do you use to measure the water table? (tick all that apply)

      ○ Borehole

○ Trial pit

      ○ Assess geomorphology of the site

      ○ Other, please specify

14. If **no**, why not? (tick all that apply)

      ○ Too costly

○ Not enough time

      ○ Don't have the equipment

      ○ Not important

      ○ Other, please specify

15. What are the current limitations of road slope design guidelines in Nepal? (tick all that apply)

○ Do not include descriptions about rock/soil types

      ○ Hard to use in the field

      ○ Not accurate

      ○ Other, please specify

16. How could the usability of the guidelines be improved? (tick all that apply)

○ More field geology descriptions

      ○ More training on how to use them

- More images

- Fewer rock categories

- More rock categories

- Other, please specify

17. How could the accuracy of the guidelines be improved? (tick all that apply)

- Include slope layer options

- Include groundwater options

- Include options for damage from blasting

- Other, please specify

*Author contributions.* EBR and BKD formulated the overarching research aims. They also designed the methodology and conducted the data collection. DGT provided supervision for the project. EBR wrote the original draft, whilst the whole team reviewed and edited it.

*Competing interests.* The authors declare that they have no conflict of interest

*Acknowledgements.* We would like to acknowledge Prof. Bruce Malamud, Dr Hanna Ruszczyk and Professor Alex Densmore at Durham University for offering their helpful comments on this paper. Funding for this research came from Durham University's Research Impact Fund.

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
