# Peer review of "A participatory approach to determine the use of road cut slope design guidelines in Nepal to lessen landslides"

_EGUsphere, 2024_

## Referee Comment (RC1)

**Overall comments**

This article presents the findings of recent qualitative research into the use of rod cut slope design guidelines in Nepal. It is a topical piece responding to a concerning rise in the number of landslides made possible by road construction in Nepal and more widely across The Himalayas. It is well written, logically structured, and accessible to specialists and non-specialists alike. The methodology is clearly set out and the results well presented. There are a good set of recommendations provided at the end of the piece, and it is clear that this work will support further work on this issue going forward. The manuscript is novel in that it really tries to get to grips with the issues as they arise "on the ground" with people involved in the process of road construction and, arguably, "disaster risk creation". It is of obvious relevance to the case study in question but does well to highlight that the findings and implications could be applied, in a broad sense, to other LIC contexts. The manuscript could be improved if it linked to some wider questions and issues relating to disaster risk management and reduction, and reflected further on the limitations of the largely technical recommendations provided. The manuscript would probably benefit from including a literature/background section rather than what is a lengthy and dense introduction. Overall, whilst this is a *natural hazards* journal I think there is scope to bring in some perspectives from critical social scientific views of landslides and disasters: not least because this is a qualitative study focussing on the links between infrastructural development and risk.

**Specific comments**

Below I will set out some specific suggestions in line with the points made above and in relation to a few specific points made in the manuscript which need revision and/or further elaboration.

**Introduction > introduction and background section**

- In the introduction, could you include some more numbers/statistics on numbers of landslides (and landslides related to road construction), casualties, economic impacts, etc? Or maybe link to specific events/landslides etc. Something to grab the reader's interest.
- I think the manuscript would benefit from splitting the current introduction into two sections. For the new introduction section, I would recommend moving the final paragraph of section 1 (109-119) to roughly line 30. This would probably mean you need to slightly rewrite the current final sentence to lead properly into a background section (lines 27-30). The paragraph lines 31-34 seems out of place and does not really add much to the manuscript. I would remove it. What remains would be the new introduction section with the new Background section becoming "In HICs" (line 35) onwards.

**New Background section**

- You cover most of what is required here to set the context for the rest of the paper but some areas could be tightened and it could flow a little more logically and respond to wider theoretical/policy debates. In short, I think you need to emphasise more the importance of roads to landslide causation and then more explicitly set out how your research responds to this challenge. More specific suggestions below:
  - From around line 56 onwards you review the literature on trends of landslide causation in Nepal. Your overview of the range of physical processes which make Nepal landslide prone is solid. For instance, you cite KC et. al (2024) who find an uptick in landslide occurrence since 2011 and attribute this to changes in rainfall patterns and the 2015 Gorkha Earthquake. You then allude to physical factors not fully explaining landslide causation in Nepal from page 4 onwards. However, and

particularly in relation to the KC et. al paper, I think there is scope to expand on the reasons for this recent uptick in landslide activity and the centrality of roads to it. For example, you could cite Rosser *et al.'s* (2021) scientific study which clearly shows that the 2015 EQ can only be attributed to roughly half of the increase in landslide activity since 2015 (page 11). Instead, they suggest the signing of the 2015 constitution, 2017 elections, and ensuing investments into road infrastructure may well explain the disconnect between the expected number of landslides in their co-seismic modelling and actual landslide numbers. This also correlates with Petley *et al.*'s (2007) foundational paper on landslide causation in Nepal. Given the focus of your paper, it would seem important to be explicit about the centrality of roads to these issues and the scientific evidence which backs this claim up. At a theoretical level, this also helps link your analysis to the idea that disasters from complex interplays of processes which escape easy categorisations between the geophysical and the geopolitical (Donovan, 2017).

- o The current paragraph from line 78 onwards disrupts the flow of the argument slightly. Some of the points here are useful but they are not well linked to the overall manuscript. To integrate it more, could you highlight the trade-off between the recent uptick in landslides and the fact that, as you say, "*the density of the total road network has more than tripled in the last three decades due to significant national and foreign investments aiming to improve economic and social development in Nepal through road construction*". This seems like the kind of point you could use to highlight the policy problems your paper is responding to. It also situates the paper more firmly in the wider literature on disasters and development (Collins, 2009), and disaster risk management (Lavell and Maskrey, 2014; McGowran and Donovan, 2021).

- A related point here is that your current framing of HIC/LIC is oversimplified (even beyond the wider question of approaching questions of development through the unit of the nation state (see Horner and Hulme, 2019; Horner, 2020). I understand this is not a paper about "development" as such but given you are adopting this HIC/LIC framing it seems important to acknowledge there is a question of inequality here and the uptake of guidelines is ultimately tied into questions of power and resources. It may also be worth caveating that the uptake of guidelines is not perfect in "HICs" but that at base there is more capacity to accommodate the extra costs adhering to guidelines incurs. An example which springs to mind in terms of the complicated relationship between economic development, disasters, and adherence to building regulations would be Turkey. One or two sentences which acknowledge that these issues are tied into political and economic processes and questions which are beyond the scope of the paper to address in depth would be sufficient. Maybe you could signpost Ed Simpson's 2021 book as an example of a text which engages with these questions more explicitly? Gurung's *Geoforum* paper is already cited but is indicative of the kinds of questions I think you could reflect more on. Dinesh Paudel's work on disaster reconstruction in Nepal also seems relevant (e.g. Paudel and Le Billon, 2020).

- Ultimately, I think this new background section needs to integrate the discussion of landslide causation, road construction, development, and the uptake of guidelines to highlight the importance of your study (which is novel in its focus on those actually involved in the construction process).

**Results and discussion**

Assuming the literature review incorporates the above points, the results and discussion could do more to respond to these more fundamental and wide-ranging questions the manuscript raises.

Beyond the technical recommendations you make, are there more policy-focussed questions your study raises? For example, would there be scope to tie the guidelines and their implementation into Nepal's rapidly developing and increasingly important commitments and frameworks relating to Disaster Risk Reduction/ the Sendai Framework, etc? You allude to questions of politics and policy in section 4.2 but I think there is more to say here, maybe in the overall conclusion itself?

**Technical comments**

Line 39 – Add a comma after "Normally"**,**

**References**

Collins, A. (2009) *Disaster and development*. Routledge.

Donovan, A. (2017) 'Geopower: Reflections on the critical geography of disasters', *Progress in Human Geography*, 41(1), pp. 44–67. Available at: https://doi.org/10.1177/0309132515627020.

Horner, R. (2020) 'Towards a new paradigm of global development? Beyond the limits of international development', *Progress in Human Geography*, 44(3), pp. 415–436. Available at: https://doi.org/10.1177/0309132519836158.

Horner, R. and Hulme, D. (2019) 'From International to Global Development: New Geographies of 21st Century Development', *Development and Change*, 50(2), pp. 347–378. Available at: https://doi.org/10.1111/dech.12379.

Lavell, A. and Maskrey, A. (2014) 'The future of disaster risk management', *Environmental Hazards*, 13(4), pp. 267–280. Available at: https://doi.org/10.1080/17477891.2014.935282.

McGowran, P. and Donovan, A. (2021) 'Assemblage Theory and Disaster Risk Management', *Progress in Human Geography*, 45(6), pp. 1601–1624.

Paudel, D. and Le Billon, P. (2020) 'Geo-Logics of Power: Disaster Capitalism, Himalayan Materialities, and the Geopolitical Economy of Reconstruction in Post-Earthquake Nepal', *Geopolitics*, 25(4), pp. 838–866. Available at: https://doi.org/10.1080/14650045.2018.1533818.

Petley, D.N. *et al.* (2007) 'Trends in landslide occurrence in Nepal', *Natural hazards*, 43(1), pp. 23–44.

Rosser, N. *et al.* (2021) 'Changing significance of landslide Hazard and risk after the 2015 Mw 7.8 Gorkha, Nepal Earthquake', *Progress in Disaster Science*, 10, p. 100159. Available at: https://doi.org/10.1016/j.pdisas.2021.100159.

---

## Referee Comment (RC2)

**Peer review report 1 on "A participatory approach to determine the use of road cut slope design guidelines in Nepal to lessen landslides"**

**1. General Comments**

After careful reading the pre-print, it is recommended to revise Manuscript egusphere-2024-1300 from the NHESS section. The manuscript showcases the application of various participatory methods to evaluate the usability of different road cut slope design guidelines in Nepal in the context of landslides prevention, which is of relevance to the field of natural hazards. The study addresses issues in existing design guidelines in relation to road cut slope failures, which are prevalent in Nepal due to a combination of natural factors and inadequacies. It involves engineers from various governmental levels, consultants, and academics to get a comprehensive understanding of the on-ground challenges and the applicability of the current guidelines. While the article provides valuable insights for the context of Nepal, it would be important to reflect on the discussion how participatory approaches can enhance other existing standards by comparing other similar studies.

The manuscript contributes to the understanding of landslide prevention in Nepal by highlighting the gaps in current practices and proposing ways to address them from the participants' perspectives. The scientific quality of the manuscript is fair, with the methods and data collection being well-explained and appropriate for the research questions posed. However, the manuscript could benefit from a more detailed explanation of the thematic analysis of qualitative data, e.g., how themes were derived from the data. A clearer outline of the limitations of the applied participatory methods would strengthen the manuscript too.

The presentation quality is acceptable, with the manuscript being susceptive to major improvements in the structured (content and flow), the use of tables and figures, and a better presentation of results. The manuscript is generally clear but some sections, particularly those dealing with technical aspects, could be simplified to make them more accessible to a broader audience. This is particularly important given the diverse audience of NHESS. The description of the methodology could be enhanced by adding a figure of how all these different methods were integrated and results analysed. Also, a table or figures summarizing clearly the results described in text in the section could significantly enhance the readability of the article. The manuscript could benefit from a more robust discussion on, for instance, how these findings could be generalized to other low-income countries facing similar challenges, or by reflecting on the relevance of the outcomes to the disaster risk management field (academia, policymaking, or practice), or examining the implications and correlations in the landslide's prevention in Nepal. For this, it might benefit from the inclusion of more recent studies on landslide prevention and road construction in other low-income countries, to provide a broader context for the findings. The conclusions are substantial and directly tied to the findings of the study. The main conclusion, development of new guidelines, needs however more elaborating and supporting information. Additionally, the key recommendations of the study should be more general recommendations to the field rather than specific recommendations for new guidelines. Likewise, for the next steps, it would be good that the suggested outlook is framed around the implementation of participatory approaches for

lessen landslides, or any other scientific relevant gap, rather than next steps for developing specific guidelines.

More specific suggestions are detailed in the section below.

| Principal criteria | Excellent (1) | Good (2) | Fair (3) | Poor (4) |
|---|---|---|---|---|
| **Scientific Significance:** Does the manuscript represent a substantial contribution to the understanding of natural hazards and their consequences (new concepts, ideas, methods, or data)? | | | X | |
| **Scientific Quality:** Are the scientific and/or technical approaches and the applied methods valid? Are the results discussed in an appropriate and balanced way (clarity of concepts and discussion, consideration of related work, including appropriate references)? | | | X | |
| **Presentation Quality:** Are the scientific data, results and conclusions presented in a clear, concise, and well-structured way (number and quality of figures/tables, appropriate use of technical and English language, simplicity of the language)? | | | X | |

**2. Specific comments**

*ABSTRACT*

Line 12: the suggestion of having a training program with "all" Nepali road engineers might be too ambitious.

*INTRODUCTION*

Line 16: need to include the context of the construction and widening of roads, since the statement is not true for every topography. For example, saying "in mountainous regions".

Line 24: when talking about limitations, it would be beneficial for the reader to include some examples of these limitations mentioned by Paudyal et al and Robson et al.

Line 25: it is important to enhance the justification and background of participatory studies in relation with the study's scope. Add why and how participatory approaches can help with as referred in the abstract: "… improved road cut slope designs to prevent these failures…. And assess the efficacy of the current guidelines…"

Lines 35-44: After reading the article, this paragraph seems irrelevant for the content and discussion, and cuts the flow of the reading. I would suggest removing it.

Lines 45-48: Those are two strong statements that claimed to be true for every LIC/LMICs, which may or may not be fully true. If there is evidence of this refer to it. Please revise and clarify if it applies to every LIC/LMICs or just some of them.

Lines 56-69: I would suggest moving it up in the section, as it provides directly context of Nepal. It could be the 1st or 2nd paragraph of the introduction.

Lines 83-93: I would remove that as such a detailed context is not very relevant for the remaining parts of the manuscript and would enhance the readability.

Line 98: See my comment regarding Lines 45-48 and include this line there.

Line 100: See my comment regarding line 24 and include those limitations there.

Figure 2: It is irrelevant for the results and discussion of the paper. It would rather go as a suppl. Material.

Lines 107-108: authors mention repeatedly the issue of new guidelines, but they don't include major context about. Are there any plans to produce new guidelines in the country? If yes, included in the context of Nepal. If not, suggesting that it is an abrupt conclusion considering the study is looking at whether the guidelines are used or not. Revise, clarify and correct. Also along the entire text.

Lines 109-110: authors state the aim of the study is to determine effectiveness f the current road cut slope design guidelines in Nepal. This is inconsistent with the abstract that mention "efficacy" which is a different criterion, so please harmonize this along the text. Moreover, the "effectiveness" or "efficacy" to lessen landslides is not evaluated, or at least, it is not very clear how. I would recommend, in overall, to take this part from the scope and aim of the study.

Lines 112-114: This is part of the methodology.

*METHODOLOGY*

Line 121: The descriptors *fair* and *representative* should be removed, as presented in Table 1, they do not seem to be as such. Additional reasons of why employed different qualitative methods can be included here instead. This would enhance the scientific validity.

Line 135: it would be beneficial to include the details of the presentations and respective discussions as supplementary material.

Line 141: information in brackets might be irrelevant for the reader. I would suggest deleting it.

Lines 141-146: It doesn't mention which local government units participated and how it was distributed among the participants. Also, it doesn't mention how people were invited and based on what (e.g., reputation, experience, current work, specialization). The selection approach based on the availability of engineers is a bit weak for a FGD.

Lines 166-169: Improve the description by providing numbers instead of words such as, *majority, some, others* included in the lines.

**QUESTIONS AND RESEARCH OUTCOMES**

Figure 3: Number of records in the figure is not clear. Add numbers on each response to understand better the figure. Also move the figure below the text referencing it.

Figure 4: It is hard to identify the number of responses of percentage. Add labels to understand it better.

**DISCUSSION**

Line 377: The aim of the study here is different from the ones mentioned before in the text. Harmonize it accordingly.

Lines 385-387: these findings are not discussed. Implications of the rule of thumb in the context of landslide most be more elaborated.

Lines 436- 441: This information was not presented in the results section. Adjust either the results or remove this part of the discussion.

Lines 442-446: It is not clear from where these suggestions come from as are not part of the scope and any presented results seem to support enough these recommendations. Suggest removing this.

Lines 466-470: This information seems to be irrelevant for the scope and issues dealt within the study. The suggested Canadian guidelines seem to be artificially integrated. If that is just an example, I would recommend introducing it as such and not as prescriptive as it is read.

**CONCLUSIONS**

Line 521: it refers again to effectiveness, which was not clearly evaluated in the study. Remove it.

**RECOMMENDATIONS:**

Lines 554-560: these two paragraphs are out of the blue, considering the results and discussion. Remove them.

**3. Technical Corrections**

Line 51: Consider providing a brief explanation or example of "rule of thumb" for international readers who may not be familiar with the term.

Line 148: Reference is misplaced.

Line 154: Appendix B is introduced first than Appendix A. I would suggest to swap the nomenclature (Appendix B → Appendix A, and viceversa).

Line 164: reference there is odd. What does it exactly supporting?

Line 165: see comment regarding line 154.

Line 502: remove the word *hugely* as it may be unnecessary.

*Issued*: **16.08.2024**

---

## Author Comment (AC1)

We thank the reviewers for their thorough comments on our manuscript: "A participatory approach to determine the use of road cut slope design guidelines in Nepal to lessen landslides" by Ellen B. Robson, Bhim Kumar Dahal and David G. Toll.

Below we provide detailed responses to these comments and how we will proceed if the editor invites us to submit a revision. The reviewer's comments are shown in black text, while our responses are in blue text. We highlight the added text in bold and strike out the deleted text. We have also included a reference list at the end of added references.

**Reviewer 1 (R1)**

**Overall comments**

(R1-00): This article presents the findings of recent qualitative research into the use of rod cut slope design guidelines in Nepal. It is a topical piece responding to a concerning rise in the number of landslides made possible by road construction in Nepal and more widely across The Himalayas. It is well written, logically structured, and accessible to specialists and non-specialists alike. The methodology is clearly set out and the results well presented. There are a good set of recommendations provided at the end of the piece, and it is clear that this work will support further work on this issue going forward. The manuscript is novel in that it really tries to get to grips with the issues as they arise "on the ground" with people involved in the process of road construction and, arguably, "disaster risk creation". It is of obvious relevance to the case study in question but does well to highlight that the findings and implications could be applied, in a broad sense, to other LIC contexts. The manuscript could be improved if it linked to some wider questions and issues relating to disaster risk management and reduction, and reflected further on the limitations of the largely technical recommendations provided. The manuscript would probably benefit from including a literature/background section rather than what is a lengthy and dense introduction. Overall, whilst this is a natural hazards journal I think there is scope to bring in some perspectives from critical social scientific views of landslides and disasters: not least because this is a qualitative study focussing on the links between infrastructural development and risk.

(Reply to R1-00): We thank R1 for their comments on the topical and novel manuscript. We agree that the manuscript could be better linked to wider questions on disaster risk management and reduction. We thank R1 for the suggestion to add a background section and agree that it will improve the readability of the paper. We will include a background section after the introduction. We will introduce some perspectives from critical social scientific views of landslides and disasters in our background.

Below we respond to each comment in turn.

**Specific comments**

Below I will set out some specific suggestions in line with the points made above and in relation to a few specific points made in the manuscript which need revision and/or further elaboration.

**Introduction > introduction and background section**

- (R1-01) In the introduction, could you include some more numbers/statistics on numbers of landslides (and landslides related to road construction), casualties, economic impacts, etc? Or maybe link to specific events/landslides etc. Something to grab the reader's interest.

(Reply to R1-01) Thank you for this suggestion. We will include the following statistics in the Introduction:

- According to Hearn (2011), 70% of slope failures on mountain roads are these shallow cut slope failures rather than larger 'natural' landslides.
- Over 10% of global rainfall-triggered fatal landslides occur in Nepal, despite having less than 0.4% of the global population (Froude & Petley, 2018)
- It has been estimated that 44% of land in Nepal has moderate-high landslide susceptibility, whilst less than 15% of the population has moderate-high exposure to landsliding (Kincey et al. 2024)

- (R1-02) I think the manuscript would benefit from splitting the current introduction into two sections. For the new introduction section, I would recommend moving the final paragraph of section 1 (109-119) to roughly line 30. This would probably mean you need to slightly rewrite the current final sentence to lead properly into a background section (lines 27-30). The paragraph lines 31-34 seems out of place and does not really add much to the manuscript. I would remove it. What remains would be the new introduction section with the new Background section becoming "In HICs" (line 35) onwards.

(Reply to R1-02) We thank R1 for this suggestion and agree that it will improve the readability of the paper. We will include a background section after the introduction. The final paragraph of the current introduction will be moved to act as the final paragraph of the new introduction. The background section will now begin with a subsection of landslides and road construction in Nepal. The final paragraph of the introduction and start of the background section will now read:

"**The participatory study utilized a range of qualitative data collection methods, including a one-day workshop, semi-structured interviews, unstructured interviews, focus groups, and questionnaires. The research was undertaken with road engineers working for a range of different agency and organisation types. The qualitative research methods are outlined in Sect. 3, and the outcomes of the research are presented in Sect. 4 and discussed further in Sect. 5. Based on the outcomes of this research, we present recommendations for new guideline development in Sect. 6. These recommendations have been shared with the Department of Roads and the Department of Local Infrastructure at the Government of Nepal. The next section presents background on landslide and road construction in Nepal (Sect. 2.1), disaster risk and road slope management (Sect. 2.2), road slope management in Nepal (Sect. 2.3), and the use of participatory studies to improve management (Sect. 2.4).**

**2. Background**

**2.1 Landslides and road construction in Nepal**

**Earthquakes are one of the main triggers of landslides (typically shallow rock avalanches) in Nepal (Owen, 2018). The other major trigger for landslides in Nepal is rainfall. The monsoon season in Nepal runs from June to September during which 80% of Nepal's annual rainfall occurs. 93% of landslides in Nepal occur during this four-month monsoon season (Froude and Petley, 2018). This dominance of landslides occurring during the monsoon was also highlighted by KC et al. (2024) who undertook an analysis of landslides that occurred in Nepal from 2011 to 2020. KC et al. (2024) also found a significant increasing trend in landslide occurrence in Nepal between 2011 and 2020 (landslide density was 0.85 events/1000 km2 in 2011 and had risen to 3.34 events/1000 km2 by 2020).**"

**New Background section**

- (R1-03) You cover most of what is required here to set the context for the rest of the paper but some areas could be tightened and it could flow a little more logically and respond to wider theoretical/policy debates. In short, I think you need to emphasise more the importance of roads to landslide causation and then more explicitly set out how your research responds to this challenge.

(Reply to R1-03) To help improve the flow of the background section and to respond to wider policy debates, we will introduce four subsections in the background section: (1) Landslides and road construction in Nepal; (2) Disaster risk and road slope management; (3) Road slope management in Nepal; and (4) The use of participatory studies to improve management. In the first subsection we will emphasise the importance of roads to landslide causation in Nepal (see reply to R1-04).

More specific suggestions below:

- (R1-04) From around line 56 onwards you review the literature on trends of landslide causation in Nepal. Your overview of the range of physical processes which make Nepal landslide prone is solid. For instance, you cite KC et. al (2024) who find an uptick in landslide occurrence since 2011 and attribute this to changes in rainfall patterns and the 2015 Gorkha Earthquake. You then allude to physical factors not fully explaining landslide causation in Nepal from page 4 onwards. However, and particularly in relation to the KC et. al paper, I think there is scope to expand on the reasons for this recent uptick in landslide activity and the centrality of roads to it. For example, you could cite Rosser et al.'s (2021) scientific study which clearly shows that the 2015 EQ can only be attributed to roughly half of the increase in landslide activity since 2015 (page 11). Instead, they suggest the signing of the 2015 constitution, 2017 elections, and ensuing investments into road infrastructure may well explain the disconnect between the expected number of landslides in their co-seismic modelling and actual landslide numbers. This also correlates with Petley et al.'s (2007) foundational paper on landslide causation in Nepal. Given the focus of your paper, it would seem important to be explicit about the centrality of roads to these issues and the scientific evidence which backs this claim up. At a theoretical level, this also helps link your analysis to the idea that disasters from complex interplays of processes which escape easy categorisations between the geophysical and the geopolitical (Donovan, 2017).

(Reply to R1-04) Thank you for your detailed suggestion. We will add the following information to the first subsection in the background to address this comment:

"Earthquakes are one of the main triggers of landslides (typically shallow rock avalanches) in Nepal (Owen, 2018). The other major trigger for landslides in Nepal is rainfall. The monsoon season in Nepal runs from June to September during which 80% of Nepal's annual rainfall occurs. 93% of landslides in Nepal occur during this four-month monsoon season (Froude and Petley, 2018). This dominance of landslides occurring during the monsoon was also highlighted by KC et al. (2024) who undertook an analysis of landslides that occurred in Nepal from 2011 to 2020. KC et al. (2024) also found a significant increasing trend in landslide occurrence in Nepal between 2011 and 2020 (landslide density was 0.85 events/1000 km2 in 2011 and had risen to 3.34 events/1000 km2 by 2020). The authors suggest that this trend is a consequence of the 2015 Mw 7.8 Gorkha earthquake in Nepal which caused ground cracking resulting in a reduction of the ground material strength and,

thereby, contributing to landslide occurrence. This suggestion was based on the finding that landslide occurrence within the 14 worst-affected districts remains significantly higher than it was before the earthquake. **Rosser et al. (2021), who used satellite images to map landslides in the 14 districts worst affected by the Gorkha earthquake up to post-monsoon 2019, found that in 2019 only roughly half of the mapped landslides in this region were triggered by the Gorkha earthquake. They suggest that the cause of other landslides should be attributed to the wider social and political context with anthropogenic activity that can exacerbate landslide susceptibility. They include the example of the boom in rural road construction activity that is anecdotally associated with the first 2017 local elections. However, Rosser et al. (2021) note that the damage resulting from the Gorkha earthquake will have created challenges in conducting sustainable road construction practices.**

**Constructing a fixed linear structure of a road on Nepal's dynamic landscape is hugely challenging. However, many other scholars have suggested that haphazard practices in road construction in Nepal have aggravated landslide activity (Hearn, 2002; Shakya and Niraula, 2008; Hearn and Shakya, 2017; McAdoo et al., 2018).** Robson et al. (2021) aimed to understand the issues around the coordination and protocol of implementing road slope stabilization that may lead to road cut slope failures by conducting qualitative data collection with stakeholders in road slope stabilization in Nepal. Key findings of this research were that roads were being haphazardly constructed, that there was poor communication between the key stakeholders, and that slope stabilization is not prioritized in road construction projects. **It is argued that these geopolitical factors play a significant role in cut slope instability on Nepal's road network.**

**As highlighted by Robson et al. (2021), Rosser et al. (2021), and KC et al. (2024) landslides in Nepal are caused by a complex interplay of geophysical and geopolitical processes that are challenging to unpick. This contributes to the growing understanding that most disasters worldwide are caused by some interplay of physical and societal processes (Donovan, 2017; McGowran and Donovan, 2021).**"

> o (R1-05) The current paragraph from line 78 onwards disrupts the flow of the argument slightly. Some of the points here are useful but they are not well linked to the overall manuscript. To integrate it more, could you highlight the trade-off between the recent uptick in landslides and the fact that, as you say, "the density of the total road network has more than tripled in the last three decades due to significant national and foreign investments aiming to improve economic and social development in Nepal through road construction". This seems like the kind of point you could use to highlight the policy problems your paper is responding to. It also situates the paper more firmly in the wider literature on disasters and development (Collins, 2009), and disaster risk management (Lavell and Maskrey, 2014; McGowran and Donovan, 2021).

(Reply to R1-05) We agree with your comment. The third subsection within the background will now include this paragraph, titled 'Road slope management in Nepal'. The prior subsection will be on general disaster risk and road slope management which will situate the paper more firmly in the wider literature on disasters and development and will introduce the context for this third subsection. The start of this second section will read:

"**Given that most disasters are rooted in a complex interplay of geophysical and geopolitical processes, they require management that addresses both the geophysical and geopolitical processes in a complimentary way (McGowran and Donovan, 2021; Lavell and Maskrey, 2014). This**

is highlighted by the Sendai Framework for Disaster Risk Reduction 2015-2030 who state that measures to reduce disaster risk should be multisectoral and inclusive (UNDRR, 2015). This report also outlines that *'clear vision, plans, competence, guidance and coordination within and across sectors, as well as participation of relevant stakeholders, are needed to strengthen disaster risk governance'* (UNDRR, 2015).

A crucial component of the management of road-related landslide risk is the design and excavation of road cut slopes (Aydin et al., 2018; McAdoo et al., 2018; Hearn, 2002). This component involves the coordination of many stakeholders, including politicians who generally provide the resources, policymakers who set and enforce the standards, the engineers responsible for designing the cut slope, and the engineers responsible for excavating it."

The start of the third subsection on road slope management in Nepal will then read:

"**The management of road slopes in Nepal is chiefly governed by the Department of Roads (DoR), provincial and local governments, and the Department of Local Infrastructure (DoLI).** The DoR, a department in the central Government of Nepal, is responsible for the management, planning, and maintenance of the strategic road network of Nepal which comprises highways (main trunk road connecting different regions, nationally and internationally) and feeder roads (connecting district roads to the highways)."

- (R1-06) A related point here is that your current framing of HIC/LIC is oversimplified (even beyond the wider question of approaching questions of development through the unit of the nation state (see Horner and Hulme, 2019; Horner, 2020). I understand this is not a paper about "development" as such but given you are adopting this HIC/LIC framing it seems important to acknowledge there is a question of inequality here and the uptake of guidelines is ultimately tied into questions of power and resources. It may also be worth caveating that the uptake of guidelines is not perfect in "HICs" but that at base there is more capacity to accommodate the extra costs adhering to guidelines incurs. An example which springs to mind in terms of the complicated relationship between economic development, disasters, and adherence to building regulations would be Turkey. One or two sentences which acknowledge that these issues are tied into political and economic processes and questions which are beyond the scope of the paper to address in depth would be sufficient. Maybe you could signpost Ed Simpson's 2021 book as an example of a text which engages with these questions more explicitly? Gurung's Geoforum paper is already cited but is indicative of the kinds of questions I think you could reflect more on. Dinesh Paudel's work on disaster reconstruction in Nepal also seems relevant (e.g. Paudel and Le Billon, 2020).

(Reply to R1-06) Thank you for this suggestion. We will edit the third subsection on disaster risk and road slope management to acknowledge the oversimplification of using the HIC/LIC framing and discuss the inequalities further:

"In high income countries (HICs) standard practice for designing road cut slopes involves a detailed site investigation (including in-situ and laboratory testing) to determine the strength of the cut slope and surrounding land, and numerical stability analyses (a process to quantify the stresses of the slope to establish it's stability) of the cut slope to determine the optimal design taking into account the strength of the cut slope geomaterial (soil/rock), as well as spatial and budget constraints. The design will generally incorporate the cut slope geometry (inclination and height), a drainage system (drainage built within, on the surface of, or next to the cut slope to direct water inside or on top of

the slope), and any additional structures (e.g. retaining walls and anchoring systems) implemented to improve stability. Normally all steps in this design process are conducted in accordance with national or international design standards. For example, British Standards are used in the United Kingdom and these outline that the design of road cut slope stabilization should conform to the Eurocode 7 (European codes for Geotechnical Design) (The British Standards Institution, 2023). Design guidelines (e.g. Geotechnical Engineering Office (2011)) and stability charts (e.g. Wyllie (2017); Li et al. (2008)), used to determine stable cut slope inclinations based on ground strength, are often used in the preliminary design stage. **It is important to note that we are generalizing and oversimplifying here by adopting HIC framing, a classification provided by World Bank based on gross national income per capita (The World Bank, 2023). The design and excavation of road cut slopes in HICs is not perfect, and there will be a vast number of examples where the design is substandard with important steps outlined here missing. However, there is generally more political and economic capacity in HICs to accommodate these steps and adhere to standards. The behavior and relationship of development and disaster management is beyond the scope of this paper, but we suggest Simpson (2022) and Horner and Hulme (2019) for more information on this topic.**

In low- and lower-middle-income countries (LIC/LMICs) in mountainous regions, **again note that we are generalizing here by adopting this framing**, the processes involved in road cut slope design and implementation can be hugely variable in technical rigor, due to having more variability in political and economic capacity. Often major roads have relatively large budgets, and design generally follows a process of geotechnical investigation and numerical analysis (similar to that in HICs), directed somewhat by design guidelines. However, for road projects with a smaller budget, cut slopes will often be designed following design guidelines in government and donor agency manuals (e.g. Department of Public Works and Highways (2007); Slope Engineering Branch (2010)), without additional geotechnical investigation nor numerical analyses (Robson et al., 2022; Hearn, 2002; Robson et al., 2021). However, Robson et al. (2022) documented that current guidelines in Nepal, as well as in other LIC/LMICs, lack technical rigor and usability."

- (R1-07) Ultimately, I think this new background section needs to integrate the discussion of landslide causation, road construction, development, and the uptake of guidelines to highlight the importance of your study (which is novel in its focus on those actually involved in the construction process).

(Reply to R1-07) We thank you for this suggestion. We will integrate the suggested discussions into the new subsections of the background section: (1) Landslides and road construction in Nepal (this will highlight landslide causation in Nepal and link to road construction); (2) Disaster risk and road slope management (this will link disaster management to road construction in different development contexts); (3) Road slope management in Nepal (this will contextualise road slope management within disaster management in Nepal, and highlight the importance of new guidelines); and (4) The use of participatory studies to improve management (this will outline why a participatory study is necessary in the development of guidelines).

**Results and discussion**

(R1-08) Assuming the literature review incorporates the above points, the results and discussion could do more to respond to these more fundamental and wide-ranging questions the manuscript raises. Beyond the technical recommendations you make, are there more policy-focussed questions your study raises? For example, would there be scope to tie the guidelines and their implementation into Nepal's rapidly developing and increasingly important commitments and frameworks relating to

Disaster Risk Reduction/ the Sendai Framework, etc? You allude to questions of politics and policy in section 4.2 but I think there is more to say here, maybe in the overall conclusion itself?

(Reply to R1-08) Thank you for your comment. We agree that we can better link our results to the wider questions on disaster management in Nepal, referring to the Sendai Framework for Disaster Risk Reduction where applicable to emphasise the importance of our findings. We will add reference to wider questions throughout the discussion, and improve the conclusion by:

[revised manuscript text omitted]

**Technical comments**

(R1-9) Line 09 – Add a comma after "Normally",

(Reply to R1-09) The manuscript will be updated to address this mistake.

**Reviewer 2 (R2)**

**1. General Comments**

(R2-00) After careful reading the pre-print, it is recommended to revise Manuscript egusphere-20241300 from the NHESS section. The manuscript showcases the application of various participatory methods to evaluate the usability of different road cut slope design guidelines in Nepal in the context of landslides prevention, which is of relevance to the field of natural hazards. The study addresses issues in existing design guidelines in relation to road cut slope failures, which are prevalent in Nepal due to a combination of natural factors and inadequacies. It involves engineers from various governmental levels, consultants, and academics to get a comprehensive understanding of the on-ground challenges and the applicability of the current guidelines. While the article provides valuable insights for the context of Nepal, it would be important to reflect on the discussion how participatory approaches can enhance other existing standards by comparing other similar studies.

(Reply to R2-00) We thank R2 for recognising that the research provides valuable insights for the context of Nepal. We agree that participatory approaches can enhance other existing standards. In the new background section, we will include a section on the use of participatory studies to improve management. Here we will introduce similar studies that use participatory studies, and existing standards that could be enhanced through a participatory study:

"As discussed, the Sendai Framework for Disaster Risk Reduction call for clear guidance to strengthen disaster risk governance (UNDRR, 2015). In addition, they highlight the importance of the participation of relevant stakeholders for the efficient management of disaster risk. Participatory research approaches are effective means to incorporate relevant stakeholders in disaster risk governance and guidance (Ardaya et al., 2019; Folhes et al., 2015). For example, Ardaya et al. (2019) use participatory methods to ease communication between the local populations living in flood risk areas of Rio de Janeiro in Brazil and the authorities to aid flood risk management. In this study, we will use participatory research to aid the development of clear guidance on road slope design to reduce the risk of road landslide, by incorporating the experience and opinions of a range of road engineers that will use the guidance.

Robson et al. (2022) discuss a range of LIC/LMIC's road cut slope design guidelines that lack in technical rigor or usability and, therefore, require an upgrade (e.g. the Philippines: Department of Public Works and Highways (2007); Malaysia: Slope Engineering Branch (2010); and Liberia: Ministry

of Public Works (2019)). **We suggest that a participatory study as presented in this paper should be conducted prior to the design of new guidelines for any of these LIC/LMIC's, to incorporate the experience and opinions of those using the guidelines in their development**."

(R2-01) The manuscript contributes to the understanding of landslide prevention in Nepal by highlighting the gaps in current practices and proposing ways to address them from the participants' perspectives. The scientific quality of the manuscript is fair, with the methods and data collection being well-explained and appropriate for the research questions posed. However, the manuscript could benefit from a more detailed explanation of the thematic analysis of qualitative data, e.g., how themes were derived from the data. A clearer outline of the limitations of the applied participatory methods would strengthen the manuscript too.

(Reply to R2-01) Thank you for this comment. We will edit the description of the thematic analysis of the qualitative data to give a more detailed explanation:

" Thematic analysis is a commonly used method to identify and analyze themes within qualitative data (Braun and Clarke, 2006). We followed five key steps for **thematic analysis**  that were adapted from Nowell et al. (2017): (a) familiarising ourselves with the data; (b) grouping the data into the predetermined themes **(deductive);** (c) identifying initial sub-themes within the main themes **(reductive);** (d) reviewing these themes; and (e) writing up the findings. **We utilised a deductive approach to initially group our data as we had clear objectives for the use of the data (i.e. to inform the development of new guidelines that are suited to Nepali road engineers). Therefore, the data was grouped into predetermined themes that would help us reach this objective. This was done by placing the data into the theme it best represented. The subthemes were then established by grouping the data within these themes based on their specific topics.**"

We will then add a section on the limitations of the methodology:

"**3.6 Limitations of the participatory study:**

**A participatory study of any kind is subject to biases introduced by the involvement of participants, with the data and findings being skewed towards the participants' perspectives (Burgess, 2002). In our study, we tried to include participants working for a range of different organization and agency types with different levels of experience so that the data was not biased towards a particular group of road engineers. In addition, we used a range of data collection methods to ensure that we gathered perspectives that may have been alienated if only one data collection method was chosen. In doing so, we also reduced self-selection bias, the biases introduced when a study solicits participation from people, and those that take part are likely to differ from those that do not (Bermingham, 2020).**

**A key limitation of this study is that the majority of data collection activities (other than the focus group discussions and around half of the questionnaires) took place in Kathmandu. This occurred as Kathmandu acted as a convenient central location for the workshop, with many engineers (particularly central government engineers and consultancies) being based in or near to Kathmandu. This means our findings are bias towards engineers that are based in Kathmandu. However, many of these engineers (the research participants) have experience working in many different regions across Nepal.**"

(R2-02) The presentation quality is acceptable, with the manuscript being susceptive to major improvements in the structured (content and flow), the use of tables and figures, and a better presentation of results.

(Reply to R2-02) We will improve the flow of the paper by:

- Dividing the current introduction into an introduction and background section and adding more context on disaster risk management to the study in the background section.
- Creating a firmer link between the background section and the results and discussion.
- Adding a figure of a flowchart to highlight the method.
- Adding a table to summarise the results of the study.
- Generalising the results and recommendations for other LIC/LMICs with similar problems.
- Improving the readability of existing figures.

(R2-03) The manuscript is generally clear but some sections, particularly those dealing with technical aspects, could be simplified to make them more accessible to a broader audience. This is particularly important given the diverse audience of NHESS.

(Reply to R2-03) We will simplify the descriptions of the technical aspects mentioned throughout the manuscript and add more definitions for technical aspects. For example:

- Line 16/17 - We will improve the definition of a road cut slope from 'a slope adjacent to the road that is often steeper than the surrounding topography' to 'a slope adjacent to the road resulting from excavation that is often steeper than the surrounding topography'.
- Line 24 - We will specify that the guidelines in question, are those used to design the geometry of the cut slope.
- Introduction - We will add context to the importance of the cut slope geometry: 'Design guidelines can be used to establish an optimal cut slope inclination based on different slope characteristics to identify a stable inclination that is not overly conservative. Generally, the lower the inclination of the slope, the more stable the slope. However, a low inclination requires more excavation and space and is, therefore, more costly.'
- Line 33 - We will incorporate a definition for drainage – 'drainage built within, on top of, or next to the cut slope to allow water to drain out of the slope'.
- Line 34 - We will list examples of structure – 'retaining walls and anchoring systems'.
- Line 37 - We will remove the examples of numerical stability analyses and replace these with a definition – a process to quantify the stresses of the slope to establish how stable it is.
- Throughout - We will edit mentions of geomaterial parameters' to 'ground strength characteristics'.
- Line 266 – We will add a sentence to explain the options in the table – 'The multiple choice options listed for this question included well known failure criteria for rocks and soils.  Failure criteria are used to quantify the stresses of a rock or soil mass at failure.'

(R2-04) The description of the methodology could be enhanced by adding a figure of how all these different methods were integrated and results analysed.

(Reply to R2-04) Thank you for this suggestion. We will add a flow chart that depicts the methodological process of the study. This will highlight the key methods of data gathering, how they were integrated, and the data analysis.

(R2-05) Also, a table or figures summarizing clearly the results described in text in the section could significantly enhance the readability of the article.

(Reply to R2-05) We will add a table to summarise the results. This will include a column for each theme, with a sentence in a row to summarise the subtheme.

(R2-06) The manuscript could benefit from a more robust discussion on, for instance, how these findings could be generalized to other low-income countries facing similar challenges, or by reflecting on the relevance of the outcomes to the disaster risk management field (academia, policymaking, or practice), or examining the implications and correlations in the landslide's prevention in Nepal. For this, it might benefit from the inclusion of more recent studies on landslide prevention and road construction in other low-income countries, to provide a broader context for the findings.

(Reply to R2-06) We will divide the current introduction into an introduction and background section, as suggested by R1. In the new background section, we will include a subsection on disaster risk and road slope management to contextualise the relevance of the study in the disaster risk management field. The start of this subsection will read:

"**Given that most disasters are rooted in a complex interplay of geophysical and geopolitical processes, they require management that addresses both the geophysical and geopolitical processes in a complimentary way (McGowran and Donovan, 2021; Lavell and Maskrey, 2014). This is highlighted by the Sendai Framework for Disaster Risk Reduction 2015-2030 who state that measures to reduce disaster risk should be multisectoral and inclusive (UNDRR, 2015). This report also outlines that 'clear vision, plans, competence, guidance and coordination within and across sectors, as well as participation of relevant stakeholders, are needed to strengthen disaster risk governance' (UNDRR, 2015).**

**A crucial component of the management of road-related landslide risk is the design and excavation of road cut slopes (Aydin et al., 2018; McAdoo et al., 2018; Hearn, 2002). This component involves the coordination of many stakeholders, including politicians who generally provide the resources, policymakers who set and enforce the standards, the engineers responsible for designing the cut slope, and the engineers responsible for excavating it.**"

We will also add more detail to the discussion and conclusion to outline how this study can also be generalised in other low-income countries, and how the findings are relevant to the disaster risk management field in general. Please see reply to R1-08 for how we would do this in the conclusions.

(R2-07) The conclusions are substantial and directly tied to the findings of the study. The main conclusion, development of new guidelines, needs however more elaborating and supporting information. Additionally, the key recommendations of the study should be more general recommendations to the field rather than specific recommendations for new guidelines. Likewise, for the next steps, it would be good that the suggested outlook is framed around the implementation of participatory approaches for lessen landslides, or any other scientific relevant gap, rather than next steps for developing specific guidelines.

(Reply to R2-07) Thank you for this comment. To address it we will add a paragraph of general recommendations based on the findings of this paper to the conclusions of the paper (see R1-08). However, since the aim of the paper is to come up with suggestions for the improvements of new guidelines in Nepal, we think it is important to also keep the specific recommendations and next steps for this process.

More specific suggestions are detailed in the section below.

ABSTRACT

(R2-08) Line 12: the suggestion of having a training program with "all" Nepali road engineers might be too ambitious.

(Reply to R2-08) Thank you for point this out. We will edit the sentence to read:

"We suggest that a program of training is conducted with  Nepali  engineers **working on provincial and local roads** with the publication of new guidelines".

INTRODUCTION

(R2-09) Line 16: need to include the context of the construction and widening of roads, since the statement is not true for every topography. For example, saying "in mountainous regions".

(Reply to R2-09) Thank you for this suggestion. We will edit this sentence to read:

"The construction and widening of roads **in mountainous regions** requires excavation of the ground alongside the road, often resulting in a road cut slope (a slope adjacent to the road that is often steeper than the surrounding topography) (Hearn, 2011).

(R2-10) Line 24: when talking about limitations, it would be beneficial for the reader to include some examples of these limitations mentioned by Paudyal et al and Robson et al.

(Reply to R2-10) We will update the manuscript to move Line 100 (limitations) to Line 24. This will now read:

"Paudyal et al. (2023); Robson et al. (2022, 2024) suggest that the extensive cut slope failures on the Nepal road network can be partly blamed on the limitations of current guidelines used by engineers to design the cut slopes in Nepal. **The geometry of a cut slope, specifically its inclination, plays a crucial role in its stability. Generally, the lower the inclination of a slope, the more stable it is. However, a lower inclination requires more excavation and space and is, therefore, more costly. Design guidelines can be used to establish an optimal cut slope inclination based on different slope characteristics to identify a stable inclination that is not overly conservative. The limitations of the Nepal guidelines outlined by Paudyal et al. (2023); Robson et al. (2022, 2024) include not including important ground strength characteristics, the strength characterization being too broad leading to mischaracterization, a lack of suitable descriptions, and often being presented in inaccessible formats**."

(R2-11) Line 25: it is important to enhance the justification and background of participatory studies in relation with the study's scope. Add why and how participatory approaches can help with as referred in the abstract: "… improved road cut slope designs to prevent these failures…. And assess the efficacy of the current guidelines…"

(Reply to R2-11) We agree. We will include an improved justification for the participatory study at this point in the paper:

"~~In this paper, we present the methods and outcomes of a participatory study conducted with Nepali engineers to assess how the current guidelines are used, how effective they are, and how they can be improved. The outcomes of this study are hugely important for the development of new design guidelines that are suited to the needs of Nepali road engineers, and that can be used to lessen road cut slope failures.~~ In this paper, we present the methods and outcomes of a participatory study conducted with Nepali engineers to establish their experience and perspective on the current use of road cut slope design guidelines in Nepal and how they can be improved. The outcomes of this paper will be used to inform the development of new guidelines for road cut slope design in Nepal.**

**We decided to conduct this participatory study (research involving the participation of people affected by the issues being researched, Cornish et al. (2023)) with Nepali road engineers to ensure that these guidelines are tailored towards their needs. These guidelines are to be developed by a collaboration between the Centre for Disaster Studies, Institute of Engineering (IoE) at Tribhuvan University, Nepal, with the Institute of Hazard, Risk and Resilience (IHRR) at Durham University, UK, supported by Mott MacDonald UK, Nepal Geotechnical Society, and the Department for Local Infrastructure (DoLI), and funded by the EPSRC Impact Acceleration Account. It is hoped that these new guidelines, informed by this study, will be used by engineers to design safe cut slopes to lessen road cut slope failures and therefore improve the resilience of the Nepali road network.**"

We will also include a subsection in the new background section on the use of participatory studies to improve management. This is detailed in the reply to R2-00.

(R2-12) Lines 35-44: After reading the article, this paragraph seems irrelevant for the content and discussion, and cuts the flow of the reading. I would suggest removing it.

(Reply to R2-12) Thank you for your suggestion. This paragraph is used to familiarise non-technical readers with protocol for cut slope design in a higher-income country setting and to introduce other cut slope guidelines. Rather than removing the paragraph, we would like to edit it to read:

"In HICs, standard practice for designing  road cut slopes involves a detailed site investigation (including in-situ and laboratory testing) to determine the strength of the  of the cut slope and surrounding land, and numerical stability analyses  of the cut slope to determine the optimal **cut slope** design taking into account the strength of the cut slope geomaterial, as well as spatial and budget constraints. Normally all steps in this design process are conducted in accordance with national or international design standards. For example, British Standards are used in the United Kingdom and these outline that the design of road cut slope stabilization should conform to the Eurocode 7 (European codes for Geotechnical Design) (The British Standards Institution, 2023). Design guidelines (e.g. Geotechnical Engineering Office (2011)) and stability charts (e.g. Wyllie (2017); Li et al. (2008)), used to determine stable cut slope  inclinations based on **ground strength** , are often used in the preliminary design stage.

(R2-13) Lines 45-48: Those are two strong statements that claimed to be true for every LIC/LMICs, which may or may not be fully true. If there is evidence of this refer to it. Please revise and clarify if it applies to every LIC/LMICs or just some of them.

(Reply to R2-13) Thank you for pointing this out. We will revise these sentences as follows:

"In LIC/LMICs i**n mountainous regions**, the processes involved in road cut slope design and implementation **can be**  hugely variable in technical rigor, due to variability in budget. **Often**  major roads **have relatively**  large budgets, **and** design generally follows a process of geotechnical investigation and numerical analysis (similar to that in HICs), directed somewhat by design guidelines. However, for road projects with a smaller budget, cut slopes will often be designed following design guidelines in government and donor agency manuals (e.g. Department of Public Works and Highways (2007); Slope Engineering Branch (2010)), without additional geotechnical investigation nor numerical analyses (Robson et al., 2022; Hearn, 2002; Robson et al., 2021)."

(R2-14) Lines 56-69: I would suggest moving it up in the section, as it provides directly context of Nepal. It could be the 1st or 2nd paragraph of the introduction.

(Reply to R2-14) Thank you for this suggestion. The first paragraph of the introduction will now read:

**"Over 10% of global rainfall-triggered fatal landslides occur in Nepal, despite having less than 0.4% of the global population (Froude and Petley, 2018). Landslides are widespread throughout Nepal due to a complex interplay of natural and anthropogenic processes (McAdoo et al., 2018; KC et al., 2024). The most significant natural triggers of slope instability in Nepal are the interactions of the mountainous and hilly topography with tectonic activity (resulting in rock movement and deformation, and earthquakes), and with the annual monsoon season (resulting in mass erosion and the reduced strength of rocks and soils) (Shakya and Niraula, 2008; Hearn and Shakya, 2017). While the most well-documented anthropogenic activity contributing to slope instability in Nepal is the rapid and haphazard construction of roads (Hearn, 2002; Shakya and Niraula, 2008; Hearn and Shakya, 2017; McAdoo et al., 2018)."**

(R2-15) Lines 83-93: I would remove that as such a detailed context is not very relevant for the remaining parts of the manuscript and would enhance the readability.

(Reply to R2-15) We will create a background section, and this paragraph will be included in here to improve readability.

(R2-16) Line 98: See my comment regarding Lines 45-48 and include this line there.

(Reply to R2-16)  We will move line 98 to lines 45-48. Line 45 will now read:

"In LIC/LMICs in mountainous regions, the processes involved in road cut slope design and implementation can be hugely variable in technical rigor, due to variability in budget. **Robson et al., (2022) documented that current guidelines in Nepal, as well as in other LIC/LMICs, lack technical rigor and usability. Often major roads have relatively large budgets, and design generally follows a process of geotechnical investigation and numerical analysis (similar to that in HICs), directed somewhat by design guidelines.**"

(R2-17) Line 100: See my comment regarding line 24 and include those limitations there.

(Reply to R2-17) See reply to R2-10.

(R2-18) Figure 2: It is irrelevant for the results and discussion of the paper. It would rather go as a suppl. Material.

(Reply to R2-18) We will move this figure into the appendices.

(R2-19) Lines 107-108: authors mention repeatedly the issue of new guidelines, but they don't include major context about. Are there any plans to produce new guidelines in the country? If yes, included in the context of Nepal. If not, suggesting that it is an abrupt conclusion considering the study is looking at whether the guidelines are used or not. Revise, clarify and correct. Also along the entire text.

(Reply to R2-19) Thank you for your comment. The outcomes of the paper will be used to inform the development of new guidelines. We will make this clear throughout the manuscript. We will update the mentioned lines to read:

**"The outcomes of this paper will be used to inform the development of new guidelines for road cut slope design tailored towards the needs of engineers in Nepal. These guidelines are to be developed by a collaboration between the Centre for Disaster Studies, Institute of Engineering (IoE) at Tribhuvan University, Nepal, with the Institute of Hazard, Risk and Resilience (IHRR) at Durham University, UK, supported by Mott MacDonald UK, Nepal Geotechnical Society, and the**

**Department for Local Infrastructure (DoLI), and funded by the EPSRC Impact Acceleration Account. It is hoped that these new guidelines, informed by this study, will be used by engineers to design safe cut slopes to lessen road cut slope failures and therefore improve the resilience of the Nepali road network.**"

(R2-20) Lines 109-110: authors state the aim of the study is to determine effectiveness f the current road cut slope design guidelines in Nepal. This is inconsistent with the abstract that mention "efficacy" which is a different criterion, so please harmonize this along the text. Moreover, the "effectiveness" or "efficacy" to lessen landslides is not evaluated, or at least, it is not very clear how. I would recommend, in overall, to take this part from the scope and aim of the study.

(Reply to R2-20) Thank you for pointing this out. We will harmonise all mentions of the aim of the study throughout the manuscript. We will remove mentioned of evaluating the effectiveness and efficacy. Towards the end of the introduction, we will state:

" **In this paper, we present the methods and outcomes of a participatory study conducted with Nepali engineers to establish the current use of road cut slope design guidelines in Nepal and how they can be improved. The outcomes of this paper will be used to inform the development of new guidelines tailored towards the needs of engineers in Nepal."**

(R2-21) Lines 112-114: This is part of the methodology.

(Reply to R2-21) Thank you for this comment. We think it is helpful to the reader to very briefly introduce the methodology in the introduction. We will reduce this to:

" The  participatory study utilized a range of qualitative data collection methods, including a one-day workshop , semi-structured interviews, unstructured interviews, focus groups, and questionnaires."

METHODOLOGY

(R2-22) Line 121: The descriptors fair and representative should be removed, as presented in Table 1, they do not seem to be as such. Additional reasons of why employed different qualitative methods can be included here instead. This would enhance the scientific validity.

(Reply to R2-22) Thank you for your comment. We will edit the manuscript as such:

"We employed a range of qualitative data collection methods **to enhance the validity and reliability of our findings, capturing the complexities of participants' experiences and perspectives**  research."

(R2-23) Line 135: it would be beneficial to include the details of the presentations and respective discussions as supplementary material.

(Reply to R2-23) Thank you for this suggestion. We will include a paragraph in the appendices on the details of the presentations and discussions.

(R2-24) Line 141: information in brackets might be irrelevant for the reader. I would suggest deleting it.

(Reply to R2-24) We will remove the information in brackets here:

"Two focus groups  took place, both with groups of local government engineers at two different local government units of Nepal . **There are 753 local government units in Nepal, and these are the main offices for local governments in Nepal**."

(R2-25) Lines 141-146: It doesn't mention which local government units participated and how it was distributed among the participants. Also, it doesn't mention how people were invited and based on what (e.g., reputation, experience, current work, specialization). The selection approach based on the availability of engineers is a bit weak for a FGD.

(Reply to R2-25) For ethical reasons, we do not want to state which local government units participated in our study. We will strengthen the methodology on the selection approach by stating:

"**The focus groups were conducted to gather rich insights into the experiences of local government engineers.** One focus group had 5 participants, and the other had 11 (Gill et al. (2008) recommends a maximum focus group size of 14).  **We invited all engineers that specialised in road design and construction at that local government unit that were available to attend. Participants had a range of experience and were employed at different levels within the local government at that unit**.  The focus groups were about an hour long and were conducted mainly in Nepali (handwritten minutes from the meeting were subsequently translated into English)."

(R2-26) Lines 166-169: Improve the description by providing numbers instead of words such as, majority, some, others included in the lines.

(Reply to R2-26) Thank you for this comment. We will edit the manuscript by adding the following information (note that exact numbers are not given on the questionnaire location as they were anonymised):

"The questions were written by the authors of this paper and designed to address the predetermined themes. **14 out of 17**  of **the** questions allowed the respondent to include an answer that was not offered on the form. **Around half of the**  questionnaires were completed at a conference in Kathmandu (conference attendees were selected at random), whilst **the other half**  were completed at provincial and local government offices."

QUESTIONS AND RESEARCH OUTCOMES

(R2-27) Figure 3: Number of records in the figure is not clear. Add numbers on each response to understand better the figure. Also move the figure below the text referencing it.

(Reply to R2-27) We will improve the figure so that the number of records is clearer and move the figure below the first mention of it.

(R2-28) Figure 4: It is hard to identify the number of responses of percentage. Add labels to understand it better.

(Reply to R2-28) We will improve this figure by adding the percentages to the labels.

DISCUSSION

(R2-29) Line 377: The aim of the study here is different from the ones mentioned before in the text. Harmonize it accordingly.

(Reply to R2-29) Thank you for pointing this out. We believe the aim stated here is most accurate, and therefore, we have harmonised other mentions with this.

(R2-30) Lines 385-387: these findings are not discussed. Implications of the rule of thumb in the context of landslide most be more elaborated.

(Reply to R2-30) Thank you for this suggested, we will add the following detail:

"Provincial and local government engineers often use a rule of thumb approach to design road cut slopes (based on questionnaire responses, as well as discussions at the focus groups). **As previously discussed, the rule of thumb approach refers to the engineer designing the cut slope based on their experience on what has worked in the surrounding area**. **The area surrounding a cut slope can certainly give important clues to the stability of that cut slope (e.g. the geological and hydrological context), however, this information alone is insufficient to determine the design of a cut slope. In addition, engineers' experience can differ substantially, meaning a rule of thumb approach is highly subjective and can result in the design of unstable slopes**."

(R2-31) Lines 436- 441: This information was not presented in the results section. Adjust either the results or remove this part of the discussion.

(Reply to R2-31) Thank you for this comment. We refer to this the Government prioritising lengthening the road over the quality of roads in line 280 of the results. We refer to politicians building roads for political gain in line 290 of the results. We will refer to the section that these are mentioned to help readability.

(R2-32) Lines 442-446: It is not clear from where these suggestions come from as are not part of the scope and any presented results seem to support enough these recommendations. Suggest removing this.

(Reply to R2-32) This recommendation for further research is also based on the results discussed in the reply to R2-33, specifically that politicians use roads as a political bargaining tool.

(R2-33) Lines 466-470: This information seems to be irrelevant for the scope and issues dealt within the study. The suggested Canadian guidelines seem to be artificially integrated. If that is just an example, I would recommend introducing it as such and not as prescriptive as it is read.

(Reply to R2-33) Thank you for this comment. We will edit the manuscript to introduce this as an example:

"This can be dealt with by conducting quality assurance checks. Therefore, protocol for quality assurance checks needs to be clarified by the DoR and highlighted in new guidelines, as well as during a training program. **This protocol could be written into policy as a local law. The SFDRR outline that necessary mechanisms and incentives should be employed to ensure high levels of compliance with the existing safety-enhancing provisions of sectoral laws and regulations (UNDRR, 2015). For example, policy makers in British Columbia, Canada, have enforced byelaws**  for the direct supervision of landslide assessments, as well as internal and external peer review of the landslide assessment (APEGBC, 2010). They state that direct supervision can 'typically take the form of specific instructions on what to observe, check, confirm, test, record and report back to the Qualified Professional' (p.

30). They discuss that the internal review should be carried out by another qualified professional in the same firm, and the external review is carried out by some who is independent."

CONCLUSIONS

(R2-34) Line 521: it refers again to effectiveness, which was not clearly evaluated in the study. Remove it.

(Reply to R2-34) Thank you for pointing this out. We will edit this to read:

"In order to develop new guidelines fit for engineers in Nepal, there needed to be a better understanding of how guidelines are currently used in Nepal, their effectiveness, and how they can best be improved **from the perspective of Nepali engineers**."

RECOMMENDATIONS:

(R2-35) Lines 554-560: these two paragraphs are out of the blue, considering the results and discussion. Remove them.

(Reply to R2-35) We will remove the first paragraph. We would like to move the second paragraph into the background section of the paper in the context of discussing how this method may be used in other mountainous LIC/LMICs settings.

**3. Technical Corrections**

(R2-36) Line 51: Consider providing a brief explanation or example of "rule of thumb" for international readers who may not be familiar with the term.

(Reply to R2-36) Thank you for this suggestion. We will include the following definition for a rule of thumb approach:

**"**Sometimes the design of road cut slopes will be based only on a rule of thumb (based on experience designing cut slopes in the local area). **'Rule of thumb' refers to the engineer designing the cut slope based on their experience on what has worked in the surrounding area, and not following any guidelines nor conducting any investigation nor analysis**."

(R2-37) Line 148: Reference is misplaced.

(Reply to R2-37) Thank you for highlighting this. We will edit the manuscript to improve its readability:

"**An interview is a qualitative research method used to gather an in-depth account of the participants' experiences (Gill et al., 2008; Flick, 2018).** We conducted interviews **with Nepali road engineers to gather in-depth accounts of their experiences of road slope stability in** Nepal to gather an in-depth account of the participants' experiences (Gill et al., 2008; Flick, 2018).

(R2-38) Line 154: Appendix B is introduced first than Appendix A. I would suggest to swap the nomenclature (Appendix B → Appendix A, and viceversa).

(Reply to R2-38) We will rename the Appendices in order of reference as suggested.

(R2-39) Line 164: reference there is odd. What does it exactly supporting?

(Reply to R2-39) Thank you for pointing this out. The reference is supporting the explanation for questionnaires. We will edit the manuscript to state:

"**Questionnaires incorporate a list of multiple-choice questions and are designed to efficiently collect information (Slattery et al., 2011)**. Nineteen questionnaires were conducted **in this study**: nine with provincial and local government engineers and 10 with central government engineers."

(R2-40) Line 165: see comment regarding line 154.  Line 502: remove the word hugely as it may be unnecessary.

(Reply to R2-40) We will remove the word hugely as suggested.

**Additional references:**

Ardaya, A. B., Evers, M., and Ribbe, L.: Participatory approaches for disaster risk governance? Exploring participatory mechanisms and mapping to close the communication gap between population living in flood risk areas and authorities in Nova Friburgo Municipality, RJ, Brazil, Land Use Policy, 88, 104 103, 2019.

Cornish, F., Breton, N., Moreno-Tabarez, U., Delgado, J., Rua, M., de Graft Aikins, A., and Hodgetts, D.: Participatory action research, Nature Reviews Methods Primers, 3, 34, 2023.

Donovan, A.: Geopower: Reflections on the critical geography of disasters, Progress in Human Geography, 41, 44–67, 2017.

Folhes, R. T., de Aguiar, A. P. D., Stoll, E., Dalla-Nora, E. L., Araújo, R., Coelho, A., and do Canto, O.: Multi-scale participatory scenario methods and territorial planning in the Brazilian Amazon, Futures, 73, 86–99, 2015.

Lavell, A. and Maskrey, A.: The future of disaster risk management, Environmental Hazards, 13, 267–280, 2014

McGowran, P. and Donovan, A.: Assemblage theory and disaster risk management, Progress in Human Geography, 45, 1601–1624, 2021.

Rosser, N., Kincey, M., Oven, K., Densmore, A., Robinson, T., Pujara, D. S., Shrestha, R., Smutny, J., Gurung, K., Lama, S., et al.: Changing significance of landslide Hazard and risk after the 2015 Mw 7.8 Gorkha, Nepal Earthquake, Progress in Disaster Science, 10, 100 159, 2021

Simpson, E.: Highways to the end of the world: Roads, roadmen and power in south Asia, Hurst Publishers, 2022

The World Bank: The World Bank by Income and Region, https://datatopics.worldbank.org/world-development-indicators/the-world-by-income-and-region.html, acessed: 16/09/2024, 2023.880

UNDRR: Sendai Framework for Disaster Risk Reduction 2015-2030, https://www.undrr.org/publication/sendai-framework-disaster-risk-reduction-2015-2030, acessed: 16/09/2024, 2015

---

## Author Response (AR1)

We thank the reviewers for their thorough comments on our manuscript: "A participatory approach to determine the use of road cut slope design guidelines in Nepal to lessen landslides" by Ellen B. Robson, Bhim Kumar Dahal and David G. Toll.

Below we provide detailed responses to these comments and how we have addressed them in the revised manuscript. The reviewer's comments are shown in black text, while our responses are in blue text. We highlight the added text in bold and strike out the deleted text. We have also included a reference list at the end of added references.

**Reviewer 1 (R1)**

**Overall comments**

(R1-00): This article presents the findings of recent qualitative research into the use of rod cut slope design guidelines in Nepal. It is a topical piece responding to a concerning rise in the number of landslides made possible by road construction in Nepal and more widely across The Himalayas. It is well written, logically structured, and accessible to specialists and non-specialists alike. The methodology is clearly set out and the results well presented. There are a good set of recommendations provided at the end of the piece, and it is clear that this work will support further work on this issue going forward. The manuscript is novel in that it really tries to get to grips with the issues as they arise "on the ground" with people involved in the process of road construction and, arguably, "disaster risk creation". It is of obvious relevance to the case study in question but does well to highlight that the findings and implications could be applied, in a broad sense, to other LIC contexts. The manuscript could be improved if it linked to some wider questions and issues relating to disaster risk management and reduction, and reflected further on the limitations of the largely technical recommendations provided. The manuscript would probably benefit from including a literature/background section rather than what is a lengthy and dense introduction. Overall, whilst this is a natural hazards journal I think there is scope to bring in some perspectives from critical social scientific views of landslides and disasters: not least because this is a qualitative study focussing on the links between infrastructural development and risk.

(Reply to R1-00): We thank R1 for their comments on the topical and novel manuscript. We agree that the manuscript could be better linked to wider questions on disaster risk management and reduction. We thank R1 for the suggestion to add a background section and agree that it will improve the readability of the paper. We have included a background section after the introduction. We have introduced some perspectives from critical social scientific views of landslides and disasters in our background.

Below we respond to each comment in turn.

**Specific comments**

Below I will set out some specific suggestions in line with the points made above and in relation to a few specific points made in the manuscript which need revision and/or further elaboration.

**Introduction > introduction and background section**

- (R1-01) In the introduction, could you include some more numbers/statistics on numbers of landslides (and landslides related to road construction), casualties, economic impacts, etc? Or maybe link to specific events/landslides etc. Something to grab the reader's interest.

(Reply to R1-01) Thank you for this suggestion. We have included the following statistics in the Introduction:

- According to Hearn (2011), 70% of slope failures on mountain roads are these shallow cut slope failures rather than larger 'natural' landslides.
- Over 10% of global rainfall-triggered fatal landslides occur in Nepal, despite having less than 0.4% of the global population (Froude & Petley, 2018)

- (R1-02) I think the manuscript would benefit from splitting the current introduction into two sections. For the new introduction section, I would recommend moving the final paragraph of section 1 (109-119) to roughly line 30. This would probably mean you need to slightly rewrite the current final sentence to lead properly into a background section (lines 27-30). The paragraph lines 31-34 seems out of place and does not really add much to the manuscript. I would remove it. What remains would be the new introduction section with the new Background section becoming "In HICs" (line 35) onwards.

(Reply to R1-02) We thank R1 for this suggestion and agree that it will improve the readability of the paper. We have included a background section after the introduction. The final paragraph of the old introduction has been moved to act as the final paragraph of the new introduction. The background section now begins with a subsection of landslides and road construction in Nepal. The final paragraph of the introduction and start of the background section now reads:

"**The participatory study utilized a range of qualitative data collection methods, including a one-day workshop, semi-structured interviews, unstructured interviews, focus groups, and questionnaires. The research was undertaken with road engineers working for a range of different agency and organisation types. The qualitative research methods are outlined in Sect. 3, and the outcomes of the research are presented in Sect. 4 and discussed further in Sect. 5. Based on the outcomes of this research, we present recommendations for new guideline development in Sect. 6. These recommendations have been shared with the Department of Roads and the Department of Local Infrastructure at the Government of Nepal. The next section presents background on landslide and road construction in Nepal (Sect. 2.1), disaster risk and road slope management (Sect. 2.2), road slope management in Nepal (Sect. 2.3), and the use of participatory studies to improve management (Sect. 2.4).**

**2. Background**

**2.1 Landslides and road construction in Nepal**

**Earthquakes are one of the main triggers of landslides (typically shallow rock avalanches) in Nepal (Owen, 2018). The other major trigger for landslides in Nepal is rainfall. The monsoon season in Nepal runs from June to September during which 80% of Nepal's annual rainfall occurs. 93% of landslides in Nepal occur during this four-month monsoon season (Froude and Petley, 2018). This dominance of landslides occurring during the monsoon was also highlighted by KC et al. (2024) who undertook an analysis of landslides that occurred in Nepal from 2011 to 2020. KC et al. (2024) also found a significant increasing trend in landslide occurrence in Nepal between 2011 and 2020 (landslide density was 0.85 events/1000 km2 in 2011 and had risen to 3.34 events/1000 km2 by 2020).**"

**New Background section**

- (R1-03) You cover most of what is required here to set the context for the rest of the paper but some areas could be tightened and it could flow a little more logically and respond to wider theoretical/policy debates. In short, I think you need to emphasise more the importance of roads to landslide causation and then more explicitly set out how your research responds to this challenge.

(Reply to R1-03) To help improve the flow of the background section and to respond to wider policy debates, we have introduced four subsections in the background section: (1) Landslides and road construction in Nepal; (2) Disaster risk and road slope management; (3) Road slope management in Nepal; and (4) The use of participatory studies to improve management. In the first subsection we emphasise the importance of roads to landslide causation in Nepal (see reply to R1-04).

More specific suggestions below:

- (R1-04) From around line 56 onwards you review the literature on trends of landslide causation in Nepal. Your overview of the range of physical processes which make Nepal landslide prone is solid. For instance, you cite KC et. al (2024) who find an uptick in landslide occurrence since 2011 and attribute this to changes in rainfall patterns and the 2015 Gorkha Earthquake. You then allude to physical factors not fully explaining landslide causation in Nepal from page 4 onwards. However, and particularly in relation to the KC et. al paper, I think there is scope to expand on the reasons for this recent uptick in landslide activity and the centrality of roads to it. For example, you could cite Rosser et al.'s (2021) scientific study which clearly shows that the 2015 EQ can only be attributed to roughly half of the increase in landslide activity since 2015 (page 11). Instead, they suggest the signing of the 2015 constitution, 2017 elections, and ensuing investments into road infrastructure may well explain the disconnect between the expected number of landslides in their co-seismic modelling and actual landslide numbers. This also correlates with Petley et al.'s (2007) foundational paper on landslide causation in Nepal. Given the focus of your paper, it would seem important to be explicit about the centrality of roads to these issues and the scientific evidence which backs this claim up. At a theoretical level, this also helps link your analysis to the idea that disasters from complex interplays of processes which escape easy categorisations between the geophysical and the geopolitical (Donovan, 2017).

(Reply to R1-04) Thank you for your detailed suggestion. We have added the following information to the first subsection in the background to address this comment:

"Earthquakes are one of the main triggers of landslides (typically shallow rock avalanches) in Nepal (Owen, 2018). The other major trigger for landslides in Nepal is rainfall. The monsoon season in Nepal runs from June to September during which 80% of Nepal's annual rainfall occurs. 93% of landslides in Nepal occur during this four-month monsoon season (Froude and Petley, 2018). This dominance of landslides occurring during the monsoon was also highlighted by KC et al. (2024) who undertook an analysis of landslides that occurred in Nepal from 2011 to 2020. KC et al. (2024) also found a significant increasing trend in landslide occurrence in Nepal between 2011 and 2020 (landslide density was 0.85 events/1000 km2 in 2011 and had risen to 3.34 events/1000 km2 by 2020). The authors suggest that this trend is a consequence of the 2015 Mw 7.8 Gorkha earthquake in Nepal which caused ground cracking resulting in a reduction of the ground material strength and, thereby, contributing to landslide occurrence. This suggestion was based on the finding that landslide occurrence within the 14 worst-affected districts remains significantly higher than it was before the

earthquake. **Rosser et al. (2021), who used satellite images to map landslides in the 14 districts worst affected by the Gorkha earthquake up to post-monsoon 2019, found that in 2019 only roughly half of the mapped landslides in this region were triggered by the Gorkha earthquake. They suggest that the cause of other landslides should be attributed to the wider social and political context with anthropogenic activity that can exacerbate landslide susceptibility. They include the example of the boom in rural road construction activity that is anecdotally associated with the first 2017 local elections. However, Rosser et al. (2021) note that the damage resulting from the Gorkha earthquake will have created challenges in conducting sustainable road construction practices.**

**The density of Nepal's total road network has more than tripled in the last three decades due to significant national and foreign investments aiming to improve economic and social development in Nepal through road construction (Government of Nepal, 2017; Gurung, 2021). Constructing a fixed linear structure of a road on Nepal's dynamic landscape is hugely challenging. However, many other scholars have suggested that haphazard practices in road construction in Nepal have aggravated landslide activity (Hearn, 2002; Shakya and Niraula, 2008; Hearn and Shakya, 2017; McAdoo et al., 2018).** Robson et al. (2021) aimed to understand the issues around the coordination and protocol of implementing road slope stabilization that may lead to road cut slope failures by conducting qualitative data collection with stakeholders in road slope stabilization in Nepal. Key findings of this research were that roads were being haphazardly constructed, that there was poor communication between the key stakeholders, and that slope stabilization is not prioritized in road construction projects. **It is argued that these geopolitical factors play a significant role in cut slope instability on Nepal's road network.**

**As highlighted by Robson et al. (2021), Rosser et al. (2021), and KC et al. (2024) landslides in Nepal are caused by a complex interplay of geophysical and geopolitical processes that are challenging to unpick. This contributes to the growing understanding that most disasters worldwide are caused by some interplay of physical and societal processes (Donovan, 2017; McGowran and Donovan, 2021)."**

- o (R1-05) The current paragraph from line 78 onwards disrupts the flow of the argument slightly. Some of the points here are useful but they are not well linked to the overall manuscript. To integrate it more, could you highlight the trade-off between the recent uptick in landslides and the fact that, as you say, "the density of the total road network has more than tripled in the last three decades due to significant national and foreign investments aiming to improve economic and social development in Nepal through road construction". This seems like the kind of point you could use to highlight the policy problems your paper is responding to. It also situates the paper more firmly in the wider literature on disasters and development (Collins, 2009), and disaster risk management (Lavell and Maskrey, 2014; McGowran and Donovan, 2021).

(Reply to R1-05) We agree with your comment. The third subsection within the background now includes this paragraph, titled 'Road slope management in Nepal'. The prior subsection is on general disaster risk and road slope management which will situate the paper more firmly in the wider literature on disasters and development and will introduce the context for this third subsection. The start of this second section reads:

"**Given that most disasters are rooted in a complex interplay of geophysical and geopolitical processes, they require management that addresses both the geophysical and geopolitical**

**processes in a complimentary way (McGowran and Donovan, 2021; Lavell and Maskrey, 2014). This is highlighted by the Sendai Framework for Disaster Risk Reduction 2015-2030 who state that measures to reduce disaster risk should be multisectoral and inclusive (UNDRR, 2015). This report also outlines that '*clear vision, plans, competence, guidance and coordination within and across sectors, as well as participation of relevant stakeholders, are needed to strengthen disaster risk governance*' (UNDRR, 2015).**

**A crucial component of the management of road-related landslide risk is the design and excavation of road cut slopes (Aydin et al., 2018; McAdoo et al., 2018; Hearn, 2002). This component involves the coordination of many stakeholders, including politicians who generally provide the resources, policymakers who set and enforce the standards, the engineers responsible for designing the cut slope, and the engineers responsible for excavating it.**"

The start of the third subsection on road slope management in Nepal then reads:

"**The management of road slopes in Nepal is chiefly governed by the Department of Roads (DoR), local governments, and the Department of Local Infrastructure (DoLI).** The DoR, a department in the federal Government of Nepal, is responsible for the management, planning, and maintenance of the strategic road network of Nepal which comprises highways (main trunk road connecting different regions, nationally and internationally) and feeder roads (connecting district roads to the highways)."

- (R1-06) A related point here is that your current framing of HIC/LIC is oversimplified (even beyond the wider question of approaching questions of development through the unit of the nation state (see Horner and Hulme, 2019; Horner, 2020). I understand this is not a paper about "development" as such but given you are adopting this HIC/LIC framing it seems important to acknowledge there is a question of inequality here and the uptake of guidelines is ultimately tied into questions of power and resources. It may also be worth caveating that the uptake of guidelines is not perfect in "HICs" but that at base there is more capacity to accommodate the extra costs adhering to guidelines incurs. An example which springs to mind in terms of the complicated relationship between economic development, disasters, and adherence to building regulations would be Turkey. One or two sentences which acknowledge that these issues are tied into political and economic processes and questions which are beyond the scope of the paper to address in depth would be sufficient. Maybe you could signpost Ed Simpson's 2021 book as an example of a text which engages with these questions more explicitly? Gurung's Geoforum paper is already cited but is indicative of the kinds of questions I think you could reflect more on. Dinesh Paudel's work on disaster reconstruction in Nepal also seems relevant (e.g. Paudel and Le Billon, 2020).

(Reply to R1-06) Thank you for this suggestion. We have edited the third background subsection on disaster risk and road slope management to acknowledge the oversimplification of using the HIC/LIC framing and discuss the inequalities further:

"In high income countries (HICs) standard practice for designing road cut slopes involves a detailed site investigation (including in-situ and laboratory testing) to determine the strength of the cut slope and surrounding land, and numerical stability analyses (a process to quantify the stresses of the slope to establish it's stability) of the cut slope to determine the optimal design taking into account the strength of the cut slope geomaterial (soil/rock), as well as spatial and budget constraints. The design will generally incorporate the cut slope geometry (inclination and height), a drainage system (drainage built within, on the surface of, or next to the cut slope to direct water inside or on top of

the slope), and any additional structures (e.g. retaining walls and anchoring systems) implemented to improve stability. Normally all steps in this design process are conducted in accordance with national or international design standards. For example, British Standards are used in the United Kingdom and these outline that the design of road cut slope stabilization should conform to the Eurocode 7 (European codes for Geotechnical Design) (The British Standards Institution, 2023). Design guidelines (e.g. Geotechnical Engineering Office (2011)) and stability charts (e.g. Wyllie (2017); Li et al. (2008)), used to determine stable cut slope inclinations based on ground strength, are often used in the preliminary design stage. **It is important to note that we are generalizing and oversimplifying here by adopting HIC framing, a classification provided by World Bank based on gross national income per capita (The World Bank, 2023). The design and excavation of road cut slopes in HICs is not perfect, and there will be a vast number of examples where the design is substandard with important steps outlined here missing. However, there is generally more political and economic capacity in HICs to accommodate these steps and adhere to standards. The behavior and relationship of development and disaster management is beyond the scope of this paper, but we suggest Simpson (2022) and Horner and Hulme (2019) for more information on this topic.**

In low- and lower-middle-income countries (LIC/LMICs) in mountainous regions, **again note that we are generalizing here by adopting this framing**, the processes involved in road cut slope design and implementation can be hugely variable in technical rigor, due to having more variability in political and economic capacity. Often major roads have relatively large budgets, and design generally follows a process of geotechnical investigation and numerical analysis (similar to that in HICs), directed somewhat by design guidelines. However, for road projects with a smaller budget, cut slopes will often be designed following design guidelines in government and donor agency manuals (e.g. Department of Public Works and Highways (2007); Slope Engineering Branch (2010)), without additional geotechnical investigation nor numerical analyses (Robson et al., 2022; Hearn, 2002; Robson et al., 2021). However, Robson et al. (2022) documented that current guidelines in Nepal, as well as in other LIC/LMICs, lack technical rigor and usability."

- (R1-07) Ultimately, I think this new background section needs to integrate the discussion of landslide causation, road construction, development, and the uptake of guidelines to highlight the importance of your study (which is novel in its focus on those actually involved in the construction process).

(Reply to R1-07) We thank you for this suggestion. We integrated the suggested discussions into the new subsections of the background section: (1) Landslides and road construction in Nepal (highlighting landslide causation in Nepal and link to road construction); (2) Disaster risk and road slope management (linking disaster management to road construction in different development contexts); (3) Road slope management in Nepal (contextualising road slope management within disaster management in Nepal, and highlight the importance of new guidelines); and (4) The use of participatory studies to improve management (outlining why a participatory study is necessary in the development of guidelines).

**Results and discussion**

(R1-08) Assuming the literature review incorporates the above points, the results and discussion could do more to respond to these more fundamental and wide-ranging questions the manuscript raises. Beyond the technical recommendations you make, are there more policy-focussed questions your study raises? For example, would there be scope to tie the guidelines and their implementation into Nepal's rapidly developing and increasingly important commitments and frameworks relating to

Disaster Risk Reduction/ the Sendai Framework, etc? You allude to questions of politics and policy in section 4.2 but I think there is more to say here, maybe in the overall conclusion itself?

(Reply to R1-08) Thank you for your comment. We agree and we have better linked our results to the wider questions on disaster management in Nepal, referring to the Sendai Framework for Disaster Risk Reduction where applicable to emphasise the importance of our findings. We have added reference to wider questions throughout the discussion, and improved the conclusion by:

[revised manuscript text omitted]

**Technical comments**

(R1-9) Line 09 – Add a comma after "Normally",

(Reply to R1-09) The manuscript has been updated to address this mistake.

**Reviewer 2 (R2)**

**1. General Comments**

(R2-00) After careful reading the pre-print, it is recommended to revise Manuscript egusphere-20241300 from the NHESS section. The manuscript showcases the application of various participatory methods to evaluate the usability of different road cut slope design guidelines in Nepal in the context of landslides prevention, which is of relevance to the field of natural hazards. The study addresses issues in existing design guidelines in relation to road cut slope failures, which are prevalent in Nepal due to a combination of natural factors and inadequacies. It involves engineers from various governmental levels, consultants, and academics to get a comprehensive understanding of the on-ground challenges and the applicability of the current guidelines. While the article provides valuable insights for the context of Nepal, it would be important to reflect on the discussion how participatory approaches can enhance other existing standards by comparing other similar studies.

(Reply to R2-00) We thank R2 for recognising that the research provides valuable insights for the context of Nepal. We agree that participatory approaches can enhance other existing standards. In the new background section, we have included a section on the use of participatory studies to improve management. Here we have introduced similar studies that use participatory studies, and existing standards that could be enhanced through a participatory study:

**"As discussed, the Sendai Framework for Disaster Risk Reduction call for clear guidance to strengthen disaster risk governance (UNDRR, 2015). In addition, they highlight the importance of the participation of relevant stakeholders for the efficient management of disaster risk. Participatory research approaches are effective means to incorporate relevant stakeholders in disaster risk governance and guidance (Ardaya et al., 2019; Folhes et al., 2015). For example, Ardaya et al. (2019) use participatory methods to ease communication between the local populations living in flood risk areas of Rio de Janeiro in Brazil and the authorities to aid flood risk management. In this study, we will use participatory research to aid the development of clear guidance on road slope design to reduce the risk of road landslide, by incorporating the experience and opinions of a range of road engineers that will use the guidance.**

Robson et al. (2022) discuss a range of LIC/LMIC's road cut slope design guidelines that lack in technical rigor or usability and, therefore, require an upgrade (e.g. the Philippines: Department of Public Works and Highways (2007); Malaysia: Slope Engineering Branch (2010); and Liberia: Ministry of Public Works (2019)). **We suggest that a participatory study as presented in this paper should be**

**conducted prior to the design of new guidelines for any of these LIC/LMIC's, to incorporate the experience and opinions of those using the guidelines in their development**."

(R2-01) The manuscript contributes to the understanding of landslide prevention in Nepal by highlighting the gaps in current practices and proposing ways to address them from the participants' perspectives. The scientific quality of the manuscript is fair, with the methods and data collection being well-explained and appropriate for the research questions posed. However, the manuscript could benefit from a more detailed explanation of the thematic analysis of qualitative data, e.g., how themes were derived from the data. A clearer outline of the limitations of the applied participatory methods would strengthen the manuscript too.

(Reply to R2-01) Thank you for this comment. We have edited the description of the thematic analysis of the qualitative data to give a more detailed explanation:

" Thematic analysis is a commonly used method to identify and analyze themes within qualitative data (Braun and Clarke, 2006). We followed five key steps for **thematic analysis**  that were adapted from Nowell et al. (2017): (a) familiarising ourselves with the data; (b) grouping the data into the predetermined themes **(deductive);** (c) identifying initial sub-themes within the main themes **(reductive);** (d) reviewing these themes; and (e) writing up the findings. **We utilised a deductive approach to initially group our data as we had clear objectives for the use of the data (i.e. to inform the development of new guidelines that are suited to Nepali road engineers). Therefore, the data was grouped into predetermined themes that would help us reach this objective. This was done by placing the data into the theme it best represented. The subthemes were then established by grouping the data within these themes based on their specific topics.**"

We have then added a section on the limitations of the methodology:

"**3.6 Limitations of the participatory study:**

**A participatory study of any kind is subject to biases introduced by the involvement of participants, with the data and findings being skewed towards the participants' perspectives (Burgess, 2002). In our study, we tried to include participants working for a range of different organization and agency types with different levels of experience so that the data was not biased towards a particular group of road engineers. In addition, we used a range of data collection methods to ensure that we gathered perspectives that may have been alienated if only one data collection method was chosen. In doing so, we also reduced self-selection bias, the biases introduced when a study solicits participation from people, and those that take part are likely to differ from those that do not (Bermingham, 2020).**

**A key limitation of this study is that the majority of data collection activities (other than the focus group discussions and around half of the questionnaires) took place in Kathmandu. This occurred as Kathmandu acted as a convenient central location for the workshop, with many engineers (particularly federal government engineers and consultancies) being based in or near to Kathmandu. This means our findings are bias towards engineers that are based in Kathmandu. However, many of these engineers (the research participants) have experience working in many different regions across Nepal.**"

(R2-02) The presentation quality is acceptable, with the manuscript being susceptive to major improvements in the structured (content and flow), the use of tables and figures, and a better presentation of results.

(Reply to R2-02) We have improved the flow of the paper by:

- Dividing the current introduction into an introduction and background section and adding more context on disaster risk management to the study in the background section.
- Creating a firmer link between the background section and the results and discussion.
- Adding a figure of a flowchart to highlight the method.
- Adding a table to summarise the results of the study.
- Generalising the results and recommendations for other LIC/LMICs with similar problems.
- Improving the readability of existing figures.

(R2-03) The manuscript is generally clear but some sections, particularly those dealing with technical aspects, could be simplified to make them more accessible to a broader audience. This is particularly important given the diverse audience of NHESS.

(Reply to R2-03) We have simplified the descriptions of the technical aspects mentioned throughout the manuscript and add more definitions for technical aspects. Line numbers refer to the new manuscript. For example:

- Line 25 - We have improved the definition of a road cut slope from 'a slope adjacent to the road that is often steeper than the surrounding topography' to 'a slope that has been excavated adjacent to a road that is often steeper than the surrounding topography'.
- Line 35 - We have specified that the guidelines in question, are those used to design the geometry of the cut slope.
- Introduction - We have added context to the importance of the cut slope geometry: 'Design guidelines can be used to establish an optimal cut slope inclination based on different slope characteristics to identify a stable inclination that is not overly conservative. Generally, the lower the inclination of the slope, the more stable the slope. However, a low inclination requires more excavation and space and is, therefore, more costly.'
- Line 109 - We have incorporated a definition for drainage – 'drainage built within, on top of, or next to the cut slope to allow water to drain out of the slope'.
- Line 110 - We have listed examples of structure – 'supporting walls and anchoring systems'.
- Line 107 - We have removed the examples of numerical stability analyses and replaced these with a definition – a process to quantify the stresses of the slope to establish how stable it is.
- Line 367 – We have added a sentence to explain the options in the table – 'The multiple-choice options listed for this question included well known failure criteria for rocks and soils. Failure criteria are used to quantify the stresses of a rock or soil mass at failure.'

(R2-04) The description of the methodology could be enhanced by adding a figure of how all these different methods were integrated and results analysed.

(Reply to R2-04) Thank you for this suggestion. We have added a flow chart that presents the methodological process of the study. This highlights the key methods of data collection and the data analysis.

(R2-05) Also, a table or figures summarizing clearly the results described in text in the section could significantly enhance the readability of the article.

(Reply to R2-05) We have added a table to summarise the results. This includes a column for each theme, with a sentence in a row to summarise each main finding.

(R2-06) The manuscript could benefit from a more robust discussion on, for instance, how these findings could be generalized to other low-income countries facing similar challenges, or by reflecting on the relevance of the outcomes to the disaster risk management field (academia, policymaking, or practice), or examining the implications and correlations in the landslide's prevention in Nepal. For this, it might benefit from the inclusion of more recent studies on landslide prevention and road construction in other low-income countries, to provide a broader context for the findings.

(Reply to R2-06) We have divided the current introduction into an introduction and background section, as suggested by R1. In the new background section, we include a subsection on disaster risk and road slope management to contextualise the relevance of the study in the disaster risk management field. The start of this subsection reads:

**"Given that most disasters are rooted in a complex interplay of geophysical and geopolitical processes, they require management that addresses both the geophysical and geopolitical processes in a complimentary way (McGowran and Donovan, 2021; Lavell and Maskrey, 2014). This is highlighted by the Sendai Framework for Disaster Risk Reduction 2015-2030 who state that measures to reduce disaster risk should be multisectoral and inclusive (UNDRR, 2015). This report also outlines that 'clear vision, plans, competence, guidance and coordination within and across sectors, as well as participation of relevant stakeholders, are needed to strengthen disaster risk governance' (UNDRR, 2015).**

**A crucial component of the management of road-related landslide risk is the design and excavation of road cut slopes (Aydin et al., 2018; McAdoo et al., 2018; Hearn, 2002). This component involves the coordination of many stakeholders, including politicians who generally provide the resources, policymakers who set and enforce the standards, the engineers responsible for designing the cut slope, and the engineers responsible for excavating it."**

We have also added more detail to the discussion and conclusion to outline how this study can also be generalised in other low-income countries, and how the findings are relevant to the disaster risk management field in general. Please see reply to R1-08 for how we do this in the conclusions.

(R2-07) The conclusions are substantial and directly tied to the findings of the study. The main conclusion, development of new guidelines, needs however more elaborating and supporting information. Additionally, the key recommendations of the study should be more general recommendations to the field rather than specific recommendations for new guidelines. Likewise, for the next steps, it would be good that the suggested outlook is framed around the implementation of participatory approaches for lessen landslides, or any other scientific relevant gap, rather than next steps for developing specific guidelines.

(Reply to R2-07) Thank you for this comment. To address it we have added a paragraph of general recommendations based on the findings of this paper to the conclusions of the paper (see R1-08). However, since the aim of the paper is to come up with suggestions for the improvements of new guidelines in Nepal, we think it is important to also keep the specific recommendations and next steps for this process.

More specific suggestions are detailed in the section below.

ABSTRACT

(R2-08) Line 12: the suggestion of having a training program with "all" Nepali road engineers might be too ambitious.

(Reply to R2-08) Thank you for pointing this out. We have removed this sentence.

INTRODUCTION

(R2-09) Line 16: need to include the context of the construction and widening of roads, since the statement is not true for every topography. For example, saying "in mountainous regions".

(Reply to R2-09) Thank you for this suggestion. We have edited this sentence (Line 25 – new manuscript) to read:

"**A road cut slope is a slope that has been excavated adjacent to a road that is often steeper than the surrounding topography.** The construction and widening of roads **in mountainous regions** often results in road cut slopes (Hearn, 2011).

(R2-10) Line 24: when talking about limitations, it would be beneficial for the reader to include some examples of these limitations mentioned by Paudyal et al and Robson et al.

(Reply to R2-10) We have updated the manuscript to move Line 100 (old manuscript) to Line 36 (new manuscript). This now reads:

"Paudyal et al. (2023); Robson et al. (2022, 2024) suggest that the extensive cut slope failures on the Nepal road network can be partly blamed on the limitations of current guidelines used by engineers to design the cut slopes in Nepal. **The geometry of a cut slope, specifically its inclination, plays a crucial role in its stability. Generally, the shallower the inclination of a slope, the more stable it is. However, a lower inclination requires more excavation and space and is, therefore, more costly. Design guidelines can be used to establish an optimal cut slope inclination based on different slope characteristics to identify a stable inclination that is not overly conservative. The limitations of the Nepal guidelines outlined by Paudyal et al. (2023); Robson et al. (2022, 2024) include not including important ground strength characteristics, the strength characterization being too broad leading to mischaracterization, a lack of suitable descriptions, and often being presented in inaccessible formats**."

(R2-11) Line 25: it is important to enhance the justification and background of participatory studies in relation with the study's scope. Add why and how participatory approaches can help with as referred in the abstract: "… improved road cut slope designs to prevent these failures…. And assess the efficacy of the current guidelines…"

(Reply to R2-11) We agree. We have included an improved justification for the participatory study (Line 40):

"~~In this paper, we present the methods and outcomes of a participatory study conducted with Nepali engineers to assess how the current guidelines are used, how effective they are, and how they can be improved. The outcomes of this study are hugely important for the development of new design guidelines that are suited to the needs of Nepali road engineers, and that can be used to lessen road cut slope failures.~~ In this paper, we present the methods and outcomes of a participatory study conducted with Nepali engineers to establish their experience and perspective on the current use of road cut slope design guidelines in Nepal and how they can be improved. The outcomes of this paper will be used to inform the development of new guidelines for road cut slope design tailored towards the needs of engineers in Nepal. We decided to conduct this participatory study (research involving the participation of people affected by the issues being researched, Cornish et al. (2023))**

**with Nepali road engineers to ensure that these guidelines are tailored towards their needs. The new guidelines are to be developed as a collaboration between the Centre for Disaster Studies, Institute of Engineering (IoE) at Tribhuvan University, Nepal, with the Institute of Hazard, Risk and Resilience (IHRR) at Durham University, UK, supported by Mott MacDonald UK, Nepal Geotechnical Society, and the Department for Local Infrastructure (DoLI), and funded by the EPSRC Impact Acceleration Account. It is hoped that the new guidelines, informed by this study, will be used by engineers to design safe cut slopes to lessen road cut slope failures and, therefore, improve the resilience of the Nepali road network. This will contribute to the Sendai Framework for Disaster Risk Reduction 2015-2030 (SFDRR) priority to strengthen disaster risk governance to manage disaster risk through the use of clear guidelines (UNDRR, 2015).**"

We have also included a subsection in the new background section on the use of participatory studies to improve management. This is detailed in the reply to R2-00.

(R2-12) Lines 35-44: After reading the article, this paragraph seems irrelevant for the content and discussion, and cuts the flow of the reading. I would suggest removing it.

(Reply to R2-12) Thank you for your suggestion. This paragraph is used to familiarise non-technical readers with protocol for cut slope design in a higher-income country setting and to introduce other cut slope guidelines. This now sits in the new background section on 'Disaster risk and road slope management' and has been edited to read:

"In high income countries (HICs), standard practice for designing  road cut slopes involves a detailed site investigation (including in-situ and laboratory testing) to determine the strength of the  of the cut slope and surrounding land, and numerical stability analyses **(a process to quantify the stresses of the slope to establish it's stability)**  of the cut slope to determine the optimal design taking into account the strength of the cut slope geomaterial **(soil/rock)**, as well as spatial and budget constraints. **The design will generally incorporate the cut slope geometry (inclination and height), a drainage system (drainage built within, on top of, or next to the cut slope to direct water inside or on top of the slope), and any additional structures (e.g. retaining walls and anchoring systems) implemented to improve stability.** Normally all steps in this design process are conducted in accordance with national or international design standards. For example, British Standards are used in the United Kingdom and these outline that the design of road cut slope stabilization should conform to the Eurocode 7 (European codes for Geotechnical Design) (The British Standards Institution, 2023). Design guidelines (e.g. Geotechnical Engineering Office (2011)) and stability charts (e.g. Wyllie (2017); Li et al. (2008)), used to determine stable cut slope  inclinations based on **ground strength** , are often used in the preliminary design stage.

(R2-13) Lines 45-48: Those are two strong statements that claimed to be true for every LIC/LMICs, which may or may not be fully true. If there is evidence of this refer to it. Please revise and clarify if it applies to every LIC/LMICs or just some of them.

(Reply to R2-13) Thank you for pointing this out. We have revised these sentences as follows (line 123 – new manuscript):

"In **low- and lower-middle-income countries** (LIC/LMICs) i**n mountainous regions, again note that we are generalizing here by adopting this framing**, the processes involved in road cut slope design and implementation **can be**  hugely variable in technical rigor, **due to having more variability in political and economic capacity** . **Often**  major roads **have relatively**

 large budgets, **and** design generally follows a process of geotechnical investigation and numerical analysis (similar to that in HICs), directed somewhat by design guidelines. However, for road projects with a smaller budget, cut slopes will often be designed following design guidelines in government and donor agency manuals (e.g. Department of Public Works and Highways (2007); Slope Engineering Branch (2010)), without additional geotechnical investigation nor numerical analyses (Robson et al., 2022; Hearn, 2002; Robson et al., 2021)."

(R2-14) Lines 56-69: I would suggest moving it up in the section, as it provides directly context of Nepal. It could be the 1st or 2nd paragraph of the introduction.

(Reply to R2-14) Thank you for this suggestion. The first paragraph of the introduction now reads:

"**Over 10% of global rainfall-triggered fatal landslides occur in Nepal, despite having less than 0.4% of the global population (Froude and Petley, 2018). Landslides are widespread throughout Nepal due to a complex interplay of natural and anthropogenic processes (McAdoo et al., 2018; KC et al., 2024). The most significant natural triggers of slope instability in Nepal are tectonic activity (through rock movement and earthquakes), and rainfall (resulting in mass erosion and the reduced strength of rocks and soils) (Shakya and Niraula, 2008; Hearn and Shakya, 2017). While the most well-documented anthropogenic activity contributing to slope instability in Nepal is the rapid and haphazard construction of roads (Hearn, 2002; Shakya and Niraula, 2008; Hearn and Shakya, 2017; McAdoo et al., 2018).**"

(R2-15) Lines 83-93: I would remove that as such a detailed context is not very relevant for the remaining parts of the manuscript and would enhance the readability.

(Reply to R2-15) We have created a background section, and this paragraph is included in here to improve readability.

(R2-16) Line 98: See my comment regarding Lines 45-48 and include this line there.

(Reply to R2-16)  We have moved line 98 (old manuscript) to lines 130 (new manuscript). Line 130 will now read:

"In **low- and lower-middle-income countries** (LIC/LMICs) i**n mountainous regions, again note that we are generalizing here by adopting this framing**, the processes involved in road cut slope design and implementation **can be**  hugely variable in technical rigor, **due to having more variability in political and economic capacity** . **Often**  major roads **have relatively**  large budgets, **and** design generally follows a process of geotechnical investigation and numerical analysis (similar to that in HICs), directed somewhat by design guidelines. However, for road projects with a smaller budget, cut slopes will often be designed following design guidelines in government and donor agency manuals (e.g. Department of Public Works and Highways (2007); Slope Engineering Branch (2010)), without additional geotechnical investigation nor numerical analyses (Robson et al., 2022; Hearn, 2002; Robson et al., 2021). **However, Robson et al. (2022) documented that current guidelines in Nepal, as well as in other LIC/LMICs, lack technical rigor and usability. Sometimes the design of road cut slopes will be based only on a rule of thumb. 'Rule of thumb' refers to the engineer designing the cut slope based on their experience of what has worked in the surrounding area, and not following any guidelines nor conducting any investigation nor analysis. Hearn and Massey (2009) conducted geotechnical assessments of case studies in Bhutan and Ethiopia and found that very limited geotechnical assessments were carried out before road construction for low-cost roads in Bhutan and Ethiopia. Robson et al. (2021) conducted a**

**series of interviews and discussions with key stakeholders in road slope stability in Nepal and found that local roads in Nepal are often excavated using a bulldozer with no prior slope design**."

(R2-17) Line 100: See my comment regarding line 24 and include those limitations there.

(Reply to R2-17) See reply to R2-10.

(R2-18) Figure 2: It is irrelevant for the results and discussion of the paper. It would rather go as a suppl. Material.

(Reply to R2-18) We have moved this figure into the appendices.

(R2-19) Lines 107-108: authors mention repeatedly the issue of new guidelines, but they don't include major context about. Are there any plans to produce new guidelines in the country? If yes, included in the context of Nepal. If not, suggesting that it is an abrupt conclusion considering the study is looking at whether the guidelines are used or not. Revise, clarify and correct. Also along the entire text.

(Reply to R2-19) Thank you for your comment. The outcomes of the paper will be used to inform the development of new guidelines. We have now made this clear throughout the manuscript. We have updated the mentioned lines to read (Line 42 – new manuscript):

"**The outcomes of this paper will be used to inform the development of new guidelines for road cut slope design tailored towards the needs of engineers in Nepal. We decided to conduct this participatory study (research involving the participation of people affected by the issues being researched, Cornish et al. (2023)) with Nepali road engineers to ensure that these guidelines are tailored towards their needs. The new guidelines are to be developed as a collaboration between the Centre for Disaster Studies, Institute of Engineering (IoE) at Tribhuvan University, Nepal, with the Institute of Hazard, Risk and Resilience (IHRR) at Durham University, UK, supported by Mott MacDonald UK, Nepal Geotechnical Society, and the Department for Local Infrastructure (DoLI), and funded by the EPSRC Impact Acceleration Account. It is hoped that the new guidelines, informed by this study, will be used by engineers to design safe cut slopes to lessen road cut slope failures and, therefore, improve the resilience of the Nepali road network. This will contribute to the Sendai Framework for Disaster Risk Reduction 2015-2030 (SFDRR) priority to strengthen disaster risk governance to manage disaster risk through the use of clear guidelines (UNDRR, 2015).**

(R2-20) Lines 109-110: authors state the aim of the study is to determine effectiveness f the current road cut slope design guidelines in Nepal. This is inconsistent with the abstract that mention "efficacy" which is a different criterion, so please harmonize this along the text. Moreover, the "effectiveness" or "efficacy" to lessen landslides is not evaluated, or at least, it is not very clear how. I would recommend, in overall, to take this part from the scope and aim of the study.

(Reply to R2-20) Thank you for pointing this out. We have harmonised all mentions of the aim of the study throughout the manuscript. We have removed mentions of evaluating the effectiveness and efficacy. Towards the end of the introduction, we have stated:

" **In this paper, we present the methods and outcomes of a participatory study conducted with Nepali**

**engineers to establish the current use of road cut slope design guidelines in Nepal and how they can be improved. The outcomes of this paper will be used to inform the development of new guidelines tailored towards the needs of engineers in Nepal."**

(R2-21) Lines 112-114: This is part of the methodology.

(Reply to R2-21) Thank you for this comment. We think it is helpful to the reader to very briefly introduce the methodology in the introduction. We have reduced this to (Line 53 – new manuscript):

" The  participatory study utilized a range of qualitative data collection methods, including a one-day workshop  semi-structured interviews, unstructured interviews, focus groups, and questionnaires. The research was undertaken with road engineers working for a range of different agencies and organization types."

METHODOLOGY

(R2-22) Line 121: The descriptors fair and representative should be removed, as presented in Table 1, they do not seem to be as such. Additional reasons of why employed different qualitative methods can be included here instead. This would enhance the scientific validity.

(Reply to R2-22) Thank you for your comment. We have edited the manuscript as such (Line 186-new manuscript):

"We employed a range of qualitative data collection methods **to enhance the validity and reliability of our findings, capturing the complexities of participants' experiences and perspectives**  research."

(R2-23) Line 135: it would be beneficial to include the details of the presentations and respective discussions as supplementary material.

(Reply to R2-23) Thank you for this suggestion. We have included a section in the appendices outlining the details of the presentations and discussions.

(R2-24) Line 141: information in brackets might be irrelevant for the reader. I would suggest deleting it.

(Reply to R2-24) We have removed the information in brackets here (line 206 – new manuscript):

"Two focus groups  took place, both with groups of local government engineers at two different local government units of Nepal . **There are 753 local government units in Nepal, and these are the main offices for local governments in Nepal**."

(R2-25) Lines 141-146: It doesn't mention which local government units participated and how it was distributed among the participants. Also, it doesn't mention how people were invited and based on what (e.g., reputation, experience, current work, specialization). The selection approach based on the availability of engineers is a bit weak for a FGD.

(Reply to R2-25) For ethical reasons, we do not want to state which local government units participated in our study. We have strengthened the methodology on the selection approach by stating (line 207-new manuscript):

"**The focus groups were conducted to gather rich insights into the experiences of local government engineers.** One focus group had 5 participants, and the other had 11 (Gill et al. (2008) recommends a maximum focus group size of 14).  **We invited all engineers that specialised in road design and construction at that local government unit that were available to attend. Participants had a range of experience and were employed at different levels within the local government at that unit**.  The focus groups were about an hour long and were conducted mainly in Nepali (handwritten minutes from the meeting were subsequently translated into English)."

(R2-26) Lines 166-169: Improve the description by providing numbers instead of words such as, majority, some, others included in the lines.

(Reply to R2-26) Thank you for this comment. We have edited the manuscript by adding the following information (note that exact numbers are not given on the questionnaire location as they were anonymised) (line 233-new manuscript):

"The questions were written by the authors of this paper and designed to address the predetermined themes. **14 out of 17**  questions allowed the respondent to include an answer  not offered on the form. **Around half of the**  questionnaires were completed at a conference in Kathmandu (conference attendees were selected at random), whilst **the other half**  were completed at local government offices."

QUESTIONS AND RESEARCH OUTCOMES

(R2-27) Figure 3: Number of records in the figure is not clear. Add numbers on each response to understand better the figure. Also move the figure below the text referencing it.

(Reply to R2-27) We have improved the figure so that the percentage of responses is outlined and moved the figure below the first mention of it.

(R2-28) Figure 4: It is hard to identify the number of responses of percentage. Add labels to understand it better.

(Reply to R2-28) We have improved this figure by making it clearer and adding the percentages to the labels.

DISCUSSION

(R2-29) Line 377: The aim of the study here is different from the ones mentioned before in the text. Harmonize it accordingly.

(Reply to R2-29) Thank you for pointing this out. We believe the aim stated here is most accurate, and therefore, we have harmonised other mentions with this.

(R2-30) Lines 385-387: these findings are not discussed. Implications of the rule of thumb in the context of landslide most be more elaborated.

(Reply to R2-30) Thank you for this suggested, we have added the following detail (line 475-new manuscript):

"Local government engineers often use a rule of thumb approach to design road cut slopes (based on questionnaire responses, as well as discussions at the focus groups). **As previously discussed, the rule of thumb approach refers to the engineer designing the cut slope based on their experience**

**on what has worked in the surrounding area**. **The area surrounding a cut slope can certainly give important clues to the stability of that cut slope (e.g. the geological and hydrological context), however, this information alone is insufficient to determine the design of a cut slope. In addition, engineers' experience can differ substantially, meaning a rule of thumb approach is highly subjective and can result in the design of unstable slopes**."

(R2-31) Lines 436- 441: This information was not presented in the results section. Adjust either the results or remove this part of the discussion.

(Reply to R2-31) Thank you for this comment. We refer to the Government prioritising lengthening the road over the quality of roads in line 383 (new manuscript) of the results. We refer to politicians building roads for political gain in line 536 (new manuscript) of the results. We have now added reference to the section that these are mentioned to help readability.

(R2-32) Lines 442-446: It is not clear from where these suggestions come from as are not part of the scope and any presented results seem to support enough these recommendations. Suggest removing this.

(Reply to R2-32) This recommendation for further research is also based on the results discussed in the reply to R2-31, specifically that politicians use roads as a political bargaining tool.

(R2-33) Lines 466-470: This information seems to be irrelevant for the scope and issues dealt within the study. The suggested Canadian guidelines seem to be artificially integrated. If that is just an example, I would recommend introducing it as such and not as prescriptive as it is read.

(Reply to R2-33) Thank you for this comment. We have edited the manuscript to introduce this as an example:

"This can be dealt with by conducting quality assurance checks. Therefore, protocol for quality assurance checks needs to be clarified by the DoR and highlighted in new guidelines, as well as during a training program. **The SFDRR outline that necessary mechanisms and incentives should be employed to ensure high levels of compliance with the existing safety-enhancing provisions of sectoral laws and regulations (UNDRR, 2015). This protocol could be written into policy as a local law. For example, policy makers in British Columbia, Canada, have enforced byelaws**  for the direct supervision of landslide assessments, as well as internal and external peer review of the landslide assessment (APEGBC, 2010). They state that direct supervision can 'typically take the form of specific instructions on what to observe, check, confirm, test, record and report back to the Qualified Professional' (p. 30). They discuss that the internal review should be carried out by another qualified professional in the same firm, and the external review is carried out by some who is independent."

CONCLUSIONS

(R2-34) Line 521: it refers again to effectiveness, which was not clearly evaluated in the study. Remove it.

(Reply to R2-34) Thank you for pointing this out. We have edited this to read:

"In order to ensure these guidelines respond to the needs of the stakeholders using them, we conducted a participatory study to gather the experiences and perspectives of Nepali road engineers on the use of the current guidelines and how they can be improved

."

RECOMMENDATIONS:

(R2-35) Lines 554-560: these two paragraphs are out of the blue, considering the results and discussion. Remove them.

(Reply to R2-35) We have removed the first paragraph. We have moved the second paragraph into the background section (sect. 2.4) of the paper in the context of discussing how this method may be used in other mountainous LIC/LMICs settings.

**3. Technical Corrections**

(R2-36) Line 51: Consider providing a brief explanation or example of "rule of thumb" for international readers who may not be familiar with the term.

(Reply to R2-36) Thank you for this suggestion. We have included the following definition for a rule of thumb approach:

"Sometimes the design of road cut slopes will be based only on a rule of thumb . **'Rule of thumb' refers to the engineer designing the cut slope based on their experience on what has worked in the surrounding area, and not following any guidelines nor conducting any investigation nor analysis**."

(R2-37) Line 148: Reference is misplaced.

(Reply to R2-37) Thank you for highlighting this. We have edited the manuscript to improve its readability:

"**An interview is a qualitative research method used to gather an in-depth account of the participants' experiences (Gill et al., 2008; Flick, 2018).** We conducted interviews **with Nepali road engineers to gather in-depth accounts of their experiences of road slope stability in** .

(R2-38) Line 154: Appendix B is introduced first than Appendix A. I would suggest to swap the nomenclature (Appendix B → Appendix A, and viceversa).

(Reply to R2-38) We have renamed the Appendices in order of reference as suggested.

(R2-39) Line 164: reference there is odd. What does it exactly supporting?

(Reply to R2-39) Thank you for pointing this out. The reference is supporting the explanation for questionnaires. We have edited the manuscript to state (line 230-new manuscript):

"**Questionnaires incorporate a list of multiple-choice questions and are designed to efficiently collect information (Slattery et al., 2011)**. Nineteen questionnaires were conducted **in this study**: nine with local government engineers and 10 with federal government engineers."

(R2-40) Line 165: see comment regarding line 154.

(Reply to R2-40) We have ordered the appendices as the appear.

(R2-41) Line 502: remove the word hugely as it may be unnecessary.

(Reply to R2-41) We have removed the word hugely as suggested.

**Additional references:**

Ardaya, A. B., Evers, M., and Ribbe, L.: Participatory approaches for disaster risk governance? Exploring participatory mechanisms and mapping to close the communication gap between population living in flood risk areas and authorities in Nova Friburgo Municipality, RJ, Brazil, Land Use Policy, 88, 104 103, 2019.

Cornish, F., Breton, N., Moreno-Tabarez, U., Delgado, J., Rua, M., de Graft Aikins, A., and Hodgetts, D.: Participatory action research, Nature Reviews Methods Primers, 3, 34, 2023.

Donovan, A.: Geopower: Reflections on the critical geography of disasters, Progress in Human Geography, 41, 44–67, 2017.

Folhes, R. T., de Aguiar, A. P. D., Stoll, E., Dalla-Nora, E. L., Araújo, R., Coelho, A., and do Canto, O.: Multi-scale participatory scenario methods and territorial planning in the Brazilian Amazon, Futures, 73, 86–99, 2015.

Lavell, A. and Maskrey, A.: The future of disaster risk management, Environmental Hazards, 13, 267–280, 2014

McGowran, P. and Donovan, A.: Assemblage theory and disaster risk management, Progress in Human Geography, 45, 1601–1624, 2021.

Rosser, N., Kincey, M., Oven, K., Densmore, A., Robinson, T., Pujara, D. S., Shrestha, R., Smutny, J., Gurung, K., Lama, S., et al.: Changing significance of landslide Hazard and risk after the 2015 Mw 7.8 Gorkha, Nepal Earthquake, Progress in Disaster Science, 10, 100 159, 2021

Simpson, E.: Highways to the end of the world: Roads, roadmen and power in south Asia, Hurst Publishers, 2022

The World Bank: The World Bank by Income and Region, https://datatopics.worldbank.org/world-development-indicators/the-world-by-income-and-region.html, acessed: 16/09/2024, 2023.880

UNDRR: Sendai Framework for Disaster Risk Reduction 2015-2030, https://www.undrr.org/publication/sendai-framework-disaster-risk-reduction-2015-2030, acessed: 16/09/2024, 2015

---

## Referee Report (RR1)

**Peer review report 2 on "A participatory approach to determine the use of road cut slope design guidelines in Nepal to lessen landslides"**

The revised Manuscript egusphere-2024-1300 from the NHESS section has significantly improved and authors have addressed thoroughly the comments from the first revision. It showcases the value of applying participatory approaches to evaluate and improve the road cut slope design guidelines in Nepal. The paper provides a well-reasoned and detailed analysis of the issues surrounding road cut slope design guidelines in Nepal, with practical and actionable recommendations to prevent landslide occurrence. Thus, it is of relevance to the field of natural hazards and the Journal.

However, it needs to strengthen the discussion and conclusion with the following revisions:

- Discussion: The use of road construction as a political bargain is a recurring theme in the text; however, the paper doesn't explore recommendations to tackle this issue.
- Conclusions:
  - While the study offers valuable insights for Nepal, especially regarding technical recommendations, it is essential to expand on the broader implications and applicability of these conclusions to other regions facing similar challenges. This particularly pertains to weighing the advantages and limitations of participatory approaches rather than providing specific technical recommendations for road cut slope design in Nepal, which are already covered in the Results and Discussion.
  - The conclusions section is lengthy, content overloaded in duplicated content from previous sections, particularly in terms of technical recommendations. A concise summary of both the gaps/challenges and the recommendations is necessary to enhance clarity and impact.
  - Next steps should prioritize future research rather than detailing the requirements for new guidelines. Any text currently in that section should be relocated to the discussion to maintain relevance, while a clear research outlook should be articulated: "Future research should focus on…" in alignment with emerging/remaining knowledge gaps and/or methodological limitations encountered.

*Issued*: **25.11.2024**

---

## Referee Report (RR2)

**Reviewer comments**

**Technical comments:**

Line 20 – using a new sentence starter instead of "whilst". Either needs to be a conjunction or a different word to start the sentence. Reads awkwardly as is.

Line 90 – the transition from line 89 to the new para on line 89 does not flow well. Not sure it needs to be a new paragraph. You could remove the sentence beginning "It is argued that these geopolitical…" and go straight from "… projects." Into "Thus, and as argued by Rosser et al and KC et al (maybe remove ref to Robson et al 2021 which is used quite a lot throughout), landslides in Nepal tend to be caused by a complex interplay of geophysical and geopolitical processes that are challenging to unpick and address (also see later comment re. importance of speaking across disciplines etc).

Line 157 – 166 – It feels like there's quite a bit of repetition in this paragraph. Consider paring down so that you only reference Paudyal and Robson refs only one or twice.

Line 208 – Note typo on Gill reference.

Line 279. Feels like the sentence starting "However" should come before Fig 3.

Line 515 – As I read this, I find myself thinking there is something to say here about the challenges of interdisciplinary working for DRR (e.g. Donovan et. al, 2023), which also links nicely to your conceptual section where you outline that landslides/disasters are characterised by complex interactions between the geophysical and geopolitical (consider adding specific reference to interdisciplinarity in the last bit of section 2.1?). In short, is there scope to mention this as an area for future research? I.e., mention that there is a need to develop vocabularies and best-practices for interdisciplinary research and action on the links between roads, risks, and resilience? Whether this comes around this section or nearer the end would be up to you.

**Reference(s)**

Donovan, A., Morin, J., & Walshe, R. (2023). Interdisciplinary research in hazards and disaster risk. *Progress in Environmental Geography*, *2*(3), 202-222.

---

## Author Response (AR2)

20/12/2024

**Reply to second round of reviewer comments on NHESS-2024-1300**

We thank the reviewers for their further comments on our manuscript: "A participatory approach to determine the use of road cut slope design guidelines in Nepal to lessen landslides" by Ellen B. Robson, Bhim Kumar Dahal and David G. Toll.

Below we provide detailed responses to these comments and the edits made to the manuscript. The reviewer's comments are shown in black text, while our responses are in blue text. We highlight the added text in bold and strike out the deleted text.

**Reviewer 1 (R1)**

Discussion:

(R1-00) The use of road construction as a political bargain is a recurring theme in the text; however, the paper doesn't explore recommendations to tackle this issue.

(Reply to R1-00) Thank you for this comment. We believe that we have adequately addressed this in the general recommendations section where we state:

'We found that politicians can have a negative impact on landslide risk by prioritizing rapidly expanding road lengths (and widths) to gain popularity, instead of constructing well-designed roads with safe road cut slopes. We suggest that politicians can improve their priorities in road construction by coordinating more effectively with other stakeholders in road slope management and road users and recognizing road slope management as a key component in their disaster risk management protocol to commit to the SFDRR.'

In the newly added section on 'Future research recommendations' we have now added: **'This study points out that politicians in Nepal use roads as a political bargaining tool. We suggest that further research should be conducted to investigate how political influence in road construction can contribute to landslide risk. We have two suggestions for the main lines of investigation into this topic: (1) research conducted to understand how road construction varies over time within an election cycle, so that the impacts following an election can be anticipated; and (2) how the link between political concerns, road construction, and road failure varies across different parts of the country. As a starting point, we need to better understand the distribution of roads, road construction, and road cut slope failures in space and time.'**

Conclusions:

(R1-01) While the study offers valuable insights for Nepal, especially regarding technical recommendations, it is essential to expand on the broader implications and applicability of these conclusions to other regions facing similar challenges. This particularly pertains to weighing the advantages and limitations of participatory

approaches rather than providing specific technical recommendations for road cut slope design in Nepal, which are already covered in the Results and Discussion.

(Reply to R1-01) Thank you for your comment. We believe that we have already expanded on the broader implications and applicability of this study in the general recommendations section of the conclusions. We have now edited these into a bulleted list to make readability clearer:

'This study highlights the roles and responsibilities that key stakeholders have in road slope management and improvements that these stakeholder groups can make to reduce the risk of road-related landslides. These improvements are relevant to other LIC/LMICs that need to improve the management of road-related landslide risk in line with the SFDRR (e.g. Bhutan and Ethiopia - Hearn and Massey (2009), the mountainous regions of India - Sana et al. (2024), Indonesia - Diara et al. (2022), Malaysia - Rahman and Mapjabil (2017)), and include:

- Policymakers need to set standards and laws for road slope management processes, and 'encourage the establishment of necessary mechanisms and incentives to ensure high levels of compliance' with these standards and laws (UNDRR, 2015, p. 17).
- Policymakers need to define and clarify a protocol for land acquisition and compensation, quality assurance checks, and spoil disposal, and provide incentives to encourage compliance with this protocol.
- We also suggest that they need to define the protocol and provide incentives for the uptake of clear guidelines.
- We found that politicians can have a negative impact on landslide risk by prioritizing rapidly expanding road lengths (and widths) to gain popularity, instead of constructing well-designed roads with safe road cut slopes. We suggest that politicians can improve their priorities in road construction by coordinating more effectively with other stakeholders in road slope management and road users and recognize road slope management as a key component in their disaster risk management protocol to commit to the SFDRR.
- Engineers and technical specialists have a crucial responsibility in designing and excavating road slopes so that they do not contribute to landslide risk. The responsibility that they have in disaster risk reduction should be conveyed to them more clearly in their training.

The coordination of these key stakeholder groups is crucial to ensure that road slope management is effective in reducing the risk of road-related landslides.'

To address your suggestion on weighing the advantages and limitations of participatory approaches, we have added the following to the section on 'Future research recommendations':

20/12/2024

'Finally, we believe that this participatory study has successfully gathered the experiences and perspectives of Nepali road engineers on the use of the current guidelines and how they can be improved. However, as stated in the limitations section, this study (and participatory study of any kind) is subject to biases introduced by the involvement of participants. Despite this, we recommend that a participatory study of this kind can be replicated in other LIC/LMICs that need to improve the management of road-related landslide risk, to ensure that improvements are made in line with the needs of road management stakeholders.'

(R1-02) The conclusions section is lengthy, content overloaded in duplicated content from previous sections, particularly in terms of technical recommendations. A concise summary of both the gaps/challenges and the recommendations is necessary to enhance clarity and impact.
(Reply to R1-02) Thank you for highlighting this. We have now cut down the text in the conclusions section to remove repetition, particularly in the technical recommendations section.

(R1-03) Next steps should prioritize future research rather than detailing the requirements for new guidelines. Any text currently in that section should be relocated to the discussion to maintain relevance, while a clear research outlook should be articulated. The conventional statement of: "Future research should focus on…" in alignment with emerging/remaining knowledge gaps and/or methodological limitations encountered.
(Reply to R1-03) Thank you for this suggestion. We have now moved the original text in the next steps subsection into a subsection at the end of the discussion titled 'Next steps for guideline development'. We have then added a new subsection at the end of the conclusions titled 'Future research recommendations. This includes the following text:

'This study points out that politicians in Nepal use roads as a political bargaining tool. We suggest that further research should be conducted to investigate how political influence in road construction can contribute to landslide risk. We have two suggestions for the main lines of investigation into this topic: (1) research conducted to understand how road construction varies over time within an election cycle, so that the impacts following an election can be anticipated; and (2) how the link between political concerns, road construction, and road failure varies across different parts of the country. As a starting point, we need to better understand the distribution of roads, road construction, and road cut slope failures in space and time.

20/12/2024

**We suggest that further research is needed on effective coordination and communication between stakeholders in road slope management.**

**This study also underscores the challenges of interdisciplinary work within disaster risk reduction (Donovan et al., 2023). We suggest that there is a need to develop vocabularies and best practices for interdisciplinary research and action at the intersection of roads, risks, and resilience.**

**Finally, we believe that this participatory study has successfully gathered the experiences and perspectives of Nepali road engineers on the use of the current guidelines and how they can be improved. However, as stated in the limitations section, this study (and participatory study of any kind) is subject to biases introduced by the involvement of participants. Despite this, we recommend that a participatory study of this kind can be replicated in other LIC/LMICs that need to improve the management of road-related landslide risk, to ensure that improvements are made in line with the needs of road management stakeholders.'**

**Reviewer 2 (R2)**

Technical comments:

(R2-00) Line 20 – using a new sentence starter instead of "whilst". Either needs to be a conjunction or a different word to start the sentence. Reads awkwardly as is.

(Reply to R2-00)
Thank you for recognising this. We have revised the manuscript to remove 'While' so it now starts with 'The'.

(R2-01) Line 90 – the transition from line 89 to the new para on line 89 does not flow well. Not sure it needs to be a new paragraph. You could remove the sentence beginning "It is argued that these geopolitical..." and go straight from "... projects." Into "Thus, and as argued by Rosser et al and KC et al (maybe remove ref to Robson et al 2021 which is used quite a lot throughout), landslides in Nepal tend to be caused by a complex interplay of geophysical and geopolitical processes that are challenging to unpick and address (also see later comment re. importance of speaking across disciplines etc).

(Reply to R2-01) Thank you for this suggestion. The manuscript has been updated as per your suggested revised sentence structure:

'Key findings of this research were that roads were being haphazardly constructed, that there is poor communication between the key stakeholders, and that slope stabilization is not prioritized in road construction projects**. Thus, and as argued** by Rosser et al. (2021) and KC et al. (2024), landslides in Nepal are caused by a complex interplay of geophysical and geopolitical processes that are challenging to unpick and address.'

20/12/2024

(R2-02) Line 157 – 166 – It feels like there's quite a bit of repetition in this paragraph. Consider paring down so that you only reference Paudyal and Robson refs only one or twice.

(Reply to R2-02) Thank you for highlighting this. We have updated the paragraph to read to remove repetition of citations.

(R2-03) Line 208 – Note typo on Gill reference.

(Reply to R2-03) Thank you for highlighting this. We have updated the manuscript to fix this error.

(R2-04) Line 279. Feels like the sentence starting "However" should come before Fig 3.

(Reply to R2-04) We have updated the manuscript as per your suggestion.

(R2-05) Line 515 – As I read this, I find myself thinking there is something to say here about the challenges of interdisciplinary working for DRR (e.g. Donovan et. al, 2023), which also links nicely to your conceptual section where you outline that landslides/disasters are characterised by complex interactions between the geophysical and geopolitical (consider adding specific reference to interdisciplinarity in the last bit of section 2.1?). In short, is there scope to mention this as an area for future research? I.e., mention that there is a need to develop vocabularies and best-practices for interdisciplinary research and action on the links between roads, risks, and resilience? Whether this comes around this section or nearer the end would be up to you.

(Reply to R2-05) Thank you for this suggestion. We have added the following paragraph to the section on 'Future research recommendations:

**'This study also underscores the challenges of interdisciplinary work within disaster risk reduction (Donovan et al.,2023). We suggest that there is a need to develop vocabularies and best practices for interdisciplinary research and action at the intersection of roads, risks, and resilience.'**

Reference(s)
Donovan, A., Morin, J., & Walshe, R. (2023). Interdisciplinary research in hazards and disaster risk. Progress in Environmental Geography, 2(3), 202-222.